# AutoGeTS: Knowledge-based Automated Generation of Text Synthetics for Improving Text Classification

## Abstract

When developing text classification models for real-world applications, one major challenge is the difficulty of collecting sufficient data for all text classes. In this work, we address this challenge by utilizing large language models (LLMs) to generate synthetic data and using such data to improve the performance of the models without waiting for more real data to be collected and labeled. As an LLM generates different synthetic data in response to different input examples, we formulate an automated workflow, which searches for input examples that lead to more "effective" synthetic data for improving the model concerned. We study three search strategies with an extensive set of experiments, and use experiment results to inform an ensemble algorithm that selects a search strategy according to the characteristics of a class. Our further experiments demonstrate that this ensemble approach is more effective than each individual strategy in our automated workflow for improving classification models using LLMs. The source code of the main software developed for this work is made available at `https://anonymous.4open.science/r/AutoGeTS-2B0D`.

## 1 Introduction

In industrial applications of text classification, the classes are typically defined according to semantic grouping as well as organizational function. Critical impediments to developing robust text classification models for such applications include (i) noticeably imbalanced class sizes, data scarcity in some classes, and (ii) changes of the categorization scheme due to organizational changes. One group of examples is automated ticketing systems in different companies and organizations for processing users' messages and distributing them to different services, e.g., IT issues, building problems, operational incidents, and service requests (Al-Hawari & Barham, 2021).

As illustrated in Figure 1, a model is initially trained on a set of labeled tickets. Classification errors necessitate manual classification and redistribution, incurring delays and costs (Li et al., 2022). With the gradual changes within each organization, messages of certain semantics may become less frequent, while those with other semantics (including new semantics) become more frequent. Over time, the model performance degrades (Gandla et al., 2024). On the one hand, such a model needs to be improved regularly. On the other hand, identifying worsened performance of a model is usually not accompanied by adequate training data for improving the model.

Synthetic data has been used to overcome the limitations of real-world data, addressing data scarcity, sensitivity, or collection cost (Lu et al., 2023; Patki et al., 2016) in many fields, such as computer vision and NLP (Mumuni et al., 2024), medical imaging (Frid-Adar et al., 2018b), autonomous driving systems (Song et al., 2023), finance (Chalé & Bastian, 2022), and cybersecurity (Potluru et al., 2023). In this work, we focus on providing a novel and cost-effective solution to improve models for classifying short messages in ticketing systems.

In particular, we conducted systematic machine learning experiments to build a knowledge base on the effectiveness of different example message selection strategies and optimization metrics for LLMs to generate synthetic data to improve models. Building on these findings, we developed a knowledge-based ensemble algorithm to control the model improvement workflow. The algorithm distributes the computational resources

(e.g., for searching examples, generating synthetic data, retraining models, and testing models) according to the knowledge gained from the systematic exploration of the algorithmic space for identifying effective message examples. We applied the algorithm to an industrial ticketing system with severe class imbalance and observed clear performance gains. We also validated our findings using other datasets and data augmentation methods (EDA (Wei & Zou, 2019) & AEDA (Karimi et al., 2021)).

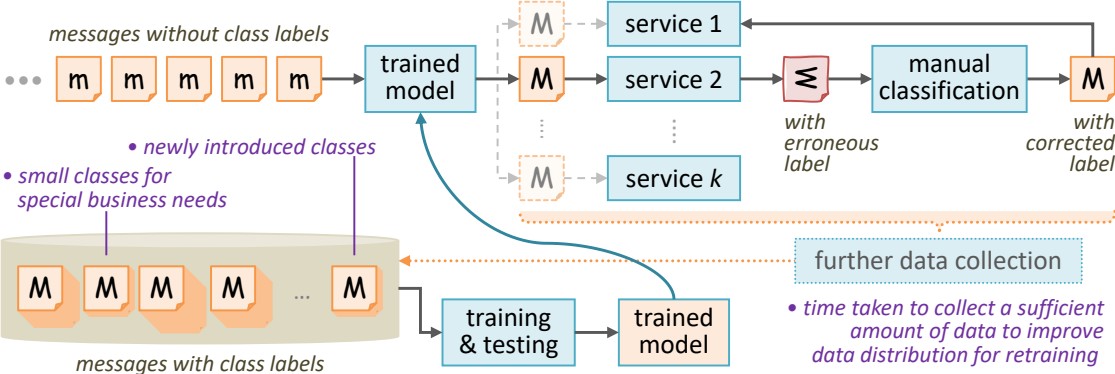

Figure 1: The workflow for developing and deploying a classification model in an industrial ticketing system, and the main obstacles impacting the performance of the model.

## 2   Related Work

Synthetic data has increasingly been used to assist in various data science tasks across many domains (Meier et al., 1988; Bersano et al., 1997). Bootstrapping (Efron, 1992; Breiman, 1996) is one of the early *data synthesis* methods. It resamples original data to simulate desired distributions and improve model performance (Sutton, 2005). To address class imbalance, synthetic data was used in conjunction with the common method to over-sample the minority classes and under-sample the majority classes (Chawla et al., 2002). Today, *data augmentation* encompasses a family of techniques that transform existing data in a dataset to generate synthetic data with desired properties, such as increasing diversity, changing distributions, filling in missing data, and so on (Jaderberg et al., 2014). In Natural Language Processing (NLP), for example, these techniques include rule-based methods that generate new training examples by randomly editing words (e.g., synonym replacement, swap, deletion) or inserting punctuation marks (Wei & Zou, 2019; Karimi et al., 2021).

Machine learning models, such as Generative Adversarial Networks (GANs) (Goodfellow et al., 2014), were used to generate synthetic data with a high level of realism and complexity. Studies have shown that models trained on GAN-generated data often can perform comparably to those trained on real data (Zhang et al., 2017; Cortés et al., 2020). For example, Frid-Adar et al. (2018a) enhanced liver lesion diagnosis using GAN-generated images. Yale et al. (2020) demonstrated comparable performance using GAN-generated electronic health records for ICU patient predictions. Croce et al. (2020) demonstrated their effectiveness in generating realistic text for NLP tasks. He et al. (2022) explored task-specific text generation. However, GAN-generated data for text classification often lacks semantic coherence and relevance to specific tasks (Torres, 2018).

Advancements in large language models (LLMs), such as GPT-2 (Radford et al., 2019), provide new approaches to overcome these limitations. LLMs excel in few-shot and zero-shot learning (Brown et al., 2020; Wang et al., 2021), adapting to unseen tasks and generating contextually relevant data that improves model robustness. GPT-3Mix by Yoo et al. (2021) demonstrated LLMs' capability to generate diverse, effective synthetic data for text classification through careful *prompt engineering*. Prompt optimization strategies have shown that carefully crafting input prompts can significantly impact the quality of generated data (Wang et al., 2023). Automated search techniques for identifying optimal prompts, such as those used in AutoPrompt (Shin et al., 2020) and PromptAgent (Wang et al., 2023), and techniques for improving prompt

diversity, such as knowledge-infused prompting (Xu et al., 2024), offer a potential solution to improve synthetic data generation.

One aspect of prompt-based augmentation is to select existing data objects as input prompts (examples). The effectiveness of such example data objects becomes a crucial factor in synthetic data generation. Several methods have been proposed to enable the selection of effective input examples, ranging from uniform distribution (Li et al., 2023) to human selection with the aid of visualization (Anonymous Authors, 2024). There are also automated strategy- or policy-based augmentation algorithms in text, image, and time series domains, such as Data Boost (Liu et al., 2020), Text AutoAugment (Ren et al., 2021), Randaugment (Cubuk et al., 2020), and TSAA (Nochumsohn & Azencot, 2025). Motivated by industrial deployment requirements, this work aims to provide an automated technique that can perform example selection much faster than the human selection approach (Anonymous Authors, 2024), while simulating some behaviors in human selection processes.

To address per-instance variability, researchers have explored ensemble methods that combine multiple strategies for synthetic data generation. Zhu et al. (2023) proposed a method that selects different ensembles of machine learning (ML) pipelines for different test instances based on their local characteristics. Pecher et al. (2024) proposed an automatic framework that combines multiple sample selection strategies to identify training examples with complementary properties for few-shot learning. Xu et al. (2022) proposed AdaDEM, which ensembles multiple convolutional networks at different granularity levels to optimize synthetic data selection for text classification. Similarly, Zhou et al. (2021) introduced MetaAugment, an ensemble framework based on reinforcement learning that dynamically selects augmentation strategies per class, ensuring adaptive augmentation. Agbesi et al. (2024) developed MuTCELM, integrating multiple sub-classifiers to capture distinct linguistic features in an ensemble framework. In traffic forecasting, Angarita-Zapata et al. (2023) used meta-learned knowledge from similar datasets to warm-start AutoML ensemble construction. In recommendation systems, Li et al. (2025) used knowledge distillation and mutual learning to stabilize click-through rate prediction ensembles when scaling up. Using the concept of ensembling but differing from these approaches, our work focuses on ensembling methods for data augmentation with the goal of improving text classification models. We use a knowledge-based approach, with which a system stores the knowledge gained from previous instances of data augmentation and uses it to guide future selection of a method for data augmentation in a model improvement workflow.

## 3 Methods

As the prior work (Anonymous Authors, 2024) has already confirmed that synthetic data generated using LLMs can improve classification models used in automated ticketing systems, this work focuses on three research questions as shown in the upper part of Figure 2. Firstly, we conducted a large number of structured experiments to understand the effect of different variants of the algorithm for finding suitable message examples on the performance of model improvement. Ideally, we could discover a superior algorithm. We will describe our experimental method in this section and report the results and analysis in Section 4. Our analysis showed the absence of such a superior algorithm. This led to the second and third research questions, which will be discussed in Section 5.

**Objectives for Model Optimization.** Ticketing systems deployed in specific organizational environments often face different, sometimes conflicting, requirements. Typical business requirements and related performance metrics include:

R1. The accuracy of every class should be as high as possible and above a certain threshold. One may optimize a model with a performance metric such as class-based *balanced accuracy* or *F1-score* as the objective function, with each threshold value as a constraint.

R2. The overall classification accuracy of a model should be as high as possible and above a certain threshold because misclassified messages lead to undesirable consequences. One may optimize a model with a global performance metric, such as overall *balanced accuracy* and overall *F1-score.*

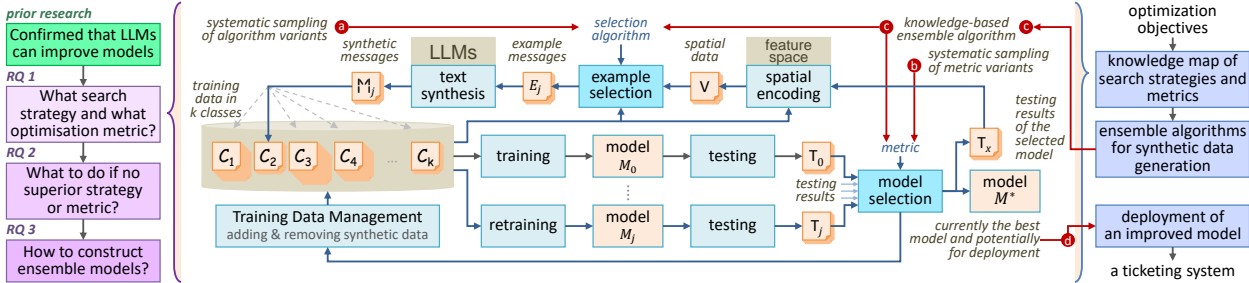

Figure 2: AutoGeTS architecture: (left) the research questions (RQs) instigated its design; (middle) its experimental workflow used for answering research questions, creating a knowledge map, and enabling a model-improvement operation; (right) its operational pipeline for improving a model. It can support the development and maintenance of multiple ticketing systems.

    R3. The recall for some specific classes (e.g., important) should be as high as possible and above a certain threshold in order to minimize the delay due to the messages in such a class being sent to other services. Class-based *recall* is the obvious metric for this requirement.

These requirements inform the definition of objective functions and constraints for model optimization. However, as the use of LLMs to generate synthetic data to aid ML is a recent approach (Anonymous Authors, 2024), it is necessary to understand how different example selection algorithms for LLMs may impact the performance of optimization with different metrics. This led to **research question 1** in Figure 2.

**AutoGeTS Architecture and Workflow.** The AutoGeTS architecture includes an experimental environment and an operational pipeline, which are illustrated in the middle and right parts of Figure 2. The experimental environment was initially used to answer **research question 1**, and later to support the operational pipeline, which will be detailed in Section 5.

Given a model $M_0$ to be improved and a set of improvement requirements (overall or class-specific), the improvement process is an optimization workflow as shown in the middle of Figure 2. To answer **research question 1**, we systematically sample different variants of the selection algorithm, run the workflow for each algorithmic sample, collect a large volume of results, and analyze the performance of these variants in obtaining the best model.

As reported in Anonymous Authors (2024), using class-based visualization plots similar to those shown in Figure 3, humans can select example messages from the training data and provide LLMs with these examples to generate synthetic data. To conduct experiments at a large scale, we replaced humans in the process. For a selected class $C$, visualization plots are automatically generated for every class. All training and testing data are mapped onto data points in an $n$-D feature space (in our work, $n = 20$). Each plot depicts data points in two feature dimensions, with blue for correct testing data points in $C$, red for incorrect ones in $C$, light green for training data points in $C$, and gray for all data points in other classes. Note that an algorithm selects example messages only from the green data points.

An algorithm for selecting example messages can have many variations, and their effectiveness depends on (1) the training datasets, (2) the structures and training parameters of models, (3) optimization metrics, and (4) the LLMs used and their control parameters. While it is not feasible to sample all algorithmic variations and their related conditions finely, our experiments covered a broad scope:

    0. We compared three search strategies for example selection, namely *Sliding Window* (SW), *Hierarchical Sliding Window* (HSW), and *Generic Algorithm* (GA). We conducted a pilot study to ensure that the parameters (e.g., window size, crossover method, etc.) of each strategy are fairly optimized. This pilot study is presented in Appendix D.

1. In addition to a real-world dataset collected in an industrial ticketing system, we also experimented with two other datasets, TREC-6 (Li & Roth, 2002) and Amazon Reviews 2023 (Hou et al., 2024).

2. We optimized the structure and training parameters for our baseline model $M_0$, and used the same structure and training parameters consistently throughout experiments for a training dataset. The hyperparameter study for $M_0$ is presented in Appendix C.

3. We compared optimization performance using a number of metrics, including *balanced accuracy*, *F1-score*, and *recall* for each of the 15 classes as well as for the overall dataset.

4. We used the GPT-3.5's API with a zero-shot prompt template (shown in Appendix D.1). We conducted a pilot study to ensure that its control parameters were fairly optimized for examples selected using all three algorithms (i.e., SW, HSW, and GA). We compared the data generated using the GPT-3.5's API and the Easy Data Augmentation (EDA) tool (Wei & Zou, 2019) & AEDA tool (Karimi et al., 2021).

**Strategies for Example Selection.** Given $m$ text messages in a training dataset, there are a total of $2^m$ different combinations for selecting $1 \leq k \leq m$ messages as input examples for LLMs to generate synthetic data. Probabilistically, synthetic data generated using any of these $2^m$ combinations might help improve a model. However, testing all $2^n$ combinations falls into the NP category.

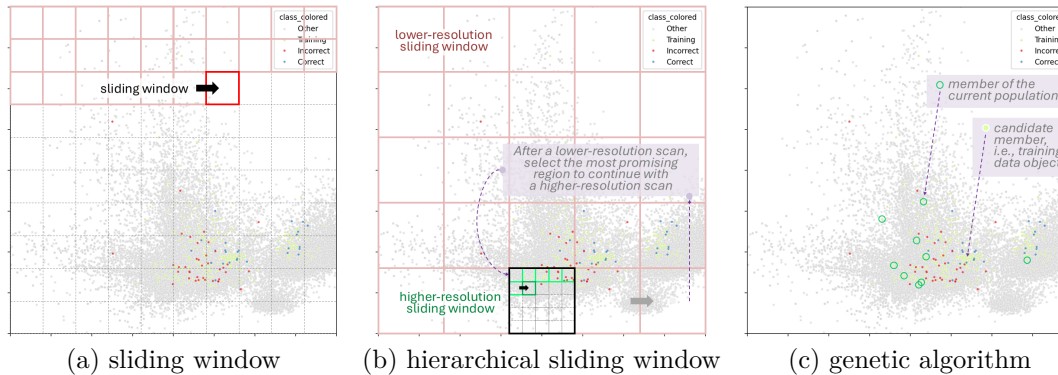

(a) sliding window      (b) hierarchical sliding window      (c) genetic algorithm

Figure 3: The three strategies experimented with in this work for selecting example messages to be used by LLMs for generating synthetic data. The examples are only selected from the training data of a particular class (light green dots), while blue, red, and gray dots illustrate contextual information, including correct and incorrect testing results and the data of other classes.

The previous non-automatic work (Anonymous Authors, 2024) has found uses visual clusters of negative testing results (red dots) in different 2D feature spaces to guide the selection of examples from training data (light green dots) in an individual class. They found that the approach was relatively effective, though some attempts would fail to result in an improved model. Based on this finding, we consider three strategies for example selection:

▶ *Sliding Window* (SW) — As illustrated in Figure 3(a), this simple automated strategy scans a 2D feature space square by square, and in each attempt, selects $k$ training data points randomly as examples. The total number of *attempts* is more than the number of windows.

▶ *Hierarchical Sliding Window* (HSW) — In the non-automatic work (Anonymous Authors, 2024), ML developers make intuitive decisions about the sizes of regions for selecting examples according to the visual patterns they are observing. In order to provide more flexibility than the SW approach, this automated strategy has a larger sliding window at level 1, and when encountering a promising or interesting window, it hierarchically makes more attempts with reduced window sizes as illustrated in Figure 3(b). The specification of $k$ is the same as SW. The total number of *attempts* is the sum of the attempts in every window examined at every level.

► *Genetic Algorithm* (GA) — In the non-automatic work (Anonymous Authors, 2024), ML developers do not just rely on simple visual clusters of erroneous testing data points, and they sometimes select examples from a few separated regions. To enable more flexibility than the SW and HSW approaches, this automated strategy allows examples to be selected from any training data in a 2D feature space. Given $m_i$ training data points in a particular class $C_i$, the GA maintains a population of $r$ chromosomes, each of which has $1 \leq s \leq m_i$ genes that are switched on, representing $s$ training data points selected as examples. In every iteration, GA evolves the population while making attempts to improve the model with chromosomes that have not been previously tested. The total number of *attempts* is the sum of the attempts in every iteration.

**Objective Functions for Optimization.** As illustrated in Figure 2, the above three strategies provide alternative algorithms to the **example selection** process, with which the workflow in the middle of the figure makes numerous *attempts* to find different subsets of training data points of a particular class as examples for LLMs to generate synthetic data. The synthetic data is then added to the training data, and the classification is retrained and tested. Therefore, the workflow is essentially an optimization process to find a set of examples $E^*$ that results in a model $M^*$ that is considered to be the best model, i.e., given $\mathbb{E}$ consisting of all sets of examples attempted in a multi-iteration workflow, the best model $M^*$ is:

$$\text{Objective:} \quad \arg\max\big(f(M_0), \max_{E_j \in \mathbb{E}} f(M_j)\big) \quad j = 1, 2, \ldots, t$$

$$\text{Subject to:} \quad M_j \leftarrow \text{TRAIN}\big(M^{\square}, D_{\text{tn}}, \text{LLM}(E_j)\big) \quad j = 1, 2, \ldots, t$$

$$f(M_j) \leftarrow \phi\big(\text{TEST}(M_j, D_{\text{tt}})\big) \quad j = 1, 2, \ldots, t$$

where $D_{\text{tn}}$ and $D_{\text{tt}}$ are the original training and testing data respectively, $M_0$ is the baseline model to be improved, $t = \|\mathbb{E}\|$ is the total number of attempts (i.e., sets of example data points selected from $D_{\text{tn}}$), $M_j(j \in [1, t])$ are sampled models in the optimization process, TRAIN() and TEST() are the processes for training and testing a classification model $M_j$, $f$ is the measurement of the testing results with a statistical measure $\phi$, $M^{\square}$ is the structure of a classification model, and LLM() is the process for LLMs to generate synthetic data using a set of examples $E_j$.

In ticketing systems, *recall* for an individual class $\phi^{\text{cr}}$ is a measure that client organizations pay attention to because improving $\phi^{\text{cr}}$ reduces the amount of manual classification in Figure 1. On the other hand, for a specific class, when we consider both TP and FP (true and false positive) messages, for a small class, the FP total (#FP) can easily overwhelm the TP total (#TP). Measures, such as *accuracy* and *precision*, can be overly biased by #FP. For this reason, we focus on *recall* $\phi^{\text{cr}}$ and *balanced accuracy* $\phi^{\text{cba}}$ to measure the improvement made in the context of a specific class $C_i$. Meanwhile, to measure the overall performance of a model, we use overall *balanced accuracy* $\phi^{\text{oba}}$ and overall F1-score $\phi^{\text{of1}}$. In summary:

$$\phi_c^{\text{cr}} = \frac{\#\text{TP}_c}{\#\text{TP}_c + \#\text{FN}_c} \quad \phi_c^{\text{cba}} = \frac{1}{2}\Big(\frac{\#\text{TP}_c}{\#\text{TP}_c + \#\text{FN}_c} + \frac{\#\text{TN}_c}{\#\text{TN}_c + \#\text{FP}_c}\Big)$$

$$\phi^{\text{of1}} = \frac{2\#\text{TP}}{2\#\text{TP} + \#\text{FP} + \#\text{FN}} \quad \phi^{\text{oba}} = \frac{1}{2}\Big(\frac{\#\text{TP}}{\#\text{TP} + \#\text{FN}} + \frac{\#\text{TN}}{\#\text{TN} + \#\text{FP}}\Big)$$

where the subscript $_c$ indicated the total values in a class $C$.

## 4 Single-Phase Experiments and Results

In this work, we first conducted single-phase experiments to answer ***research question 1*** as discussed in Section 3. Here "single-phase" means that a baseline model $M_0$ is improved by running the workflow using $\mathbb{E}_a$, i.e., examples selected from the training data of a single class $C_a$. As indicated by (a) and (b) in Figure 2, we systematically sampled three algorithmic variants and four metric variants, together with the variations of classes where example messages were selected. In Section 5, we will report that the systematic testing can provide an answer to ***research questions 2 & 3***, while supporting multi-phase optimization, with which $M_0$ is improved by running the workflows in multiple phases, using examples from different classes, i.e., $\mathbb{E}_a$, $\mathbb{E}_b$, etc.

For ***research question 1***, we sought answers in several aspects:

- How does each of the three strategies (SW, HSW, GA) perform under different testing conditions (e.g., performance metric, time allowed for finding $M^*$, etc.).

- Is there a superior strategy for each metric $\phi$ as an objective function?

- How does optimization using one metric $\phi$ affect the model performance measured with other metrics?

- How does improvement made using examples from one class affect the overall performance of the model and its performance in other classes?

- Is synthetic data generated by LLMs comparable with traditional data augmentation methods? For this, we also conducted experiments with data generated using the Easy Data Augmentation (EDA) tool (Wei & Zou, 2019) and the AEDA tool (Karimi et al., 2021).

- Are the findings obtained in our experiments data-dependent? For this, we also conducted experiments on two public datasets, TREC-6 (Li & Roth, 2002) and Amazon Reviews 2023 (Hou et al., 2024).

**Experiment Setup.** Our main experiments were conducted using a real-world dataset collected in an industrial ticketing system. It has 39,100 pieces of messages as training and testing data objects. Their labels fall into 15 classes for different services (Figure 1). As shown in Table 1, the dataset is highly imbalanced, with some classes having less than 1% of the total data objects. Because of the imbalance, we split the dataset into 60% for training, 20% for optimization testing, and 20% for performance testing outside the optimization process. Developing models with imbalanced data is a common phenomenon among almost all ticketing systems in different organizations.

Table 1: The performance of the original CatBoost model $M_0$

| Class | Class Size | Balanced Accuracy | Recall | F1–Score |
|---|---|---|---|---|
| T1 | 8529 | 0.986 | 0.979 | 0.977 |
| T2 | 11350 | 0.950 | 0.941 | 0.921 |
| T3 | 4719 | 0.952 | 0.914 | 0.922 |
| T4 | 1387 | 0.899 | 0.801 | 0.859 |
| T5 | 2755 | 0.889 | 0.794 | 0.794 |
| T6 | 1888 | 0.821 | 0.665 | 0.623 |
| T7 | 1963 | 0.883 | 0.780 | 0.766 |
| T8 | 1028 | 0.828 | 0.665 | 0.672 |
| T9 | 1466 | 0.861 | 0.747 | 0.680 |
| T10 | 1699 | 0.761 | 0.540 | 0.554 |
| T11 | 471 | 0.973 | 0.947 | 0.967 |
| T12 | 358 | 0.742 | 0.484 | 0.608 |
| T13 | 180 | 0.666 | 0.333 | 0.469 |
| T14 | 764 | 0.772 | 0.548 | 0.607 |
| T15 | 543 | 0.726 | 0.452 | 0.596 |
| Overall | 39100 | 0.923 | 0.856 | 0.856 |

Referring to the workflow in the middle of Figure 2, we used GPT-3.5 (version: 2023-03-15-preview) to generate synthetic text with main parameters *temperature* = 0.7, *max tokens* = 550, *top p* = 0.5, *frequency penalty* = 0.3, and *presence penalty* = 0.0. The original model ($M_0$) was developed in an industrial application using CatBoost. To ensure this research is relevant to the industrial application, we trained all models using CatBoost consistently with fixed hyperparameters, i.e., *number of iterations* = 300, *learning rate* = 0.2, *depth* = 8, *L2 leaf regularization* = 1.

**Comparison of Strategies and Metrics.** We systematically tested the combinations of three strategies (SW, HSW, GA) and four objective metrics ($\phi^{\mathrm{cr}}, \phi^{\mathrm{cba}}, \phi^{\mathrm{oba}}, \phi^{\mathrm{of1}}$). When these combinations were applied to each of the 15 classes where example messages were selected, there were a total of 180 combinations. Table

Table 2: (a) The results of systematic testing of 180 combinations of 3 strategies, 4 objective metrics, and 15 classes in the ticketing dataset. The bars in cells depict the improvement in the range $(0\%, 50\%]$, while red texts indicate worsened performance. (b) The results can be summarized as a knowledge map showing the best strategy or strategies for each metric-class combination. For each map region, the strategies within 0.03% difference from the best strategy are also selected.

(a) the results of systematic testing

| Class | ΔCR% SW | ΔCR% HSW | ΔCR% GA | ΔCBA% SW | ΔCBA% HSW | ΔCBA% GA | ΔOBA% SW | ΔOBA% HSW | ΔOBA% GA | ΔOF1% SW | ΔOF1% HSW | ΔOF1% GA |
|---|---|---|---|---|---|---|---|---|---|---|---|---|
| T1 | 0.23 | 0.35 | 0.35 | 0.05 | 0.05 | -0.12 | 0.30 | 0.31 | 0.20 | 0.61 | 0.63 | 0.40 |
| T2 | 0.71 | 0.85 | 0.80 | 0.53 | 0.50 | -0.11 | 0.33 | 0.36 | 0.10 | 0.66 | 0.73 | 0.19 |
| T3 | 2.01 | 2.01 | 2.01 | 0.61 | 0.65 | 0.73 | 0.33 | 0.30 | 0.31 | 0.66 | 0.61 | 0.63 |
| T4 | 8.00 | 9.78 | 8.44 | 3.39 | 4.10 | 3.59 | 0.32 | 0.36 | 0.39 | 0.64 | 0.72 | 0.78 |
| T5 | 5.67 | 4.26 | 2.84 | 2.12 | 1.57 | 0.66 | 0.34 | 0.36 | 0.11 | 0.69 | 0.73 | 0.22 |
| T6 | 6.85 | 6.45 | 4.84 | 2.32 | 2.33 | 1.22 | 0.38 | 0.33 | 0.16 | 0.76 | 0.66 | 0.33 |
| T7 | 5.84 | 5.84 | -0.32 | 2.58 | 2.56 | -0.30 | 0.39 | 0.36 | 0.13 | 0.78 | 0.73 | 0.25 |
| T8 | 10.19 | 11.46 | 5.10 | 3.87 | 4.33 | 1.71 | 0.33 | 0.32 | 0.22 | 0.66 | 0.64 | 0.43 |
| T9 | 4.39 | 5.37 | 2.93 | 1.70 | 2.22 | 0.89 | 0.39 | 0.40 | 0.28 | 0.79 | 0.60 | 0.57 |
| T10 | 12.15 | 11.05 | 12.08 | 3.69 | 3.25 | 3.68 | 0.37 | 0.48 | 0.29 | 0.75 | 0.96 | 0.58 |
| T11 | 1.12 | 1.12 | 1.12 | 0.55 | 0.55 | 0.55 | 0.33 | 0.33 | 0.42 | 0.66 | 0.66 | 0.85 |
| T12 | 29.03 | 32.26 | 32.26 | 9.43 | 10.41 | 10.45 | 0.40 | 0.35 | 0.39 | 0.81 | 0.70 | 0.79 |
| T13 | 26.67 | 33.33 | 33.33 | 6.65 | 8.22 | 8.23 | 0.33 | 0.40 | 0.37 | 0.66 | 0.81 | 0.75 |
| T14 | 13.51 | 16.22 | 16.22 | 4.23 | 5.12 | 5.13 | 0.25 | 0.31 | 0.21 | 0.51 | 0.63 | 0.42 |
| T15 | 20.75 | 20.75 | 24.53 | 6.28 | 6.14 | 7.34 | 0.36 | 0.36 | 0.40 | 0.72 | 0.73 | 0.81 |

(b) a summary of the best strategy-metric combinations

| Class | Measured Performance Metric CR | CBA | OBA | OF1 |
|---|---|---|---|---|
| T1 | HSW/GA | SW/HSW | SW/HSW | SW/HSW |
| T2 | HSW | SW/HSW | SW/HSW | HSW |
| T3 | SW/HSW/GA | GA | SW/HSW/GA | SW/GA |
| T4 | HSW | HSW | HSW/GA | GA |
| T5 | SW | SW | SW/HSW | HSW |
| T6 | SW | SW/HSW | SW | SW |
| T7 | SW/HSW | SW/HSW | SW/HSW | SW |
| T8 | HSW | HSW | SW/HSW | SW/HSW |
| T9 | HSW | HSW | SW | SW |
| T10 | SW | SW/GA | HSW | HSW |
| T11 | SW/HSW/GA | SW/HSW/GA | GA | GA |
| T12 | | | SW/GA | SW/GA |
| T13 | HSW/GA | HSW/GA | HSW/GA | HSW |
| T14 | | | HSW | HSW |
| T15 | GA | GA | GA | GA |

Table 3: When a model is improved with the SW strategy and objective metric $\phi^{\mathrm{cr}}$ (class-based recall) for classes $T1, T2, \dots, T15$ in the ticketing dataset, the direct improvement of the target classes can be seen in the yellow cells on the left part of the table. Meanwhile, there are positive and negative impacts on other classes and other performance metrics. The green bars in cells depict the improvement in the range $(0\%, 50\%]$, while red texts indicate a negative impact. The last row shows the baseline performance. There are 12 such tables for three strategies (SW, HSW, GA) and four objective metrics ($\phi_{\mathrm{cr}}, \phi_{\mathrm{cba}}, \phi_{\mathrm{oba}}, \phi_{\mathrm{of1}}$).

| | | Optimization: SW-CR-Class[row]; Performance Metric: ΔOF1%, ΔCR% | | | | | | | | | | | | | | | | Optimization: SW-CR-Class[row]; Performance Metric: ΔOBA%, ΔCBA% | | | | | | | | | | | | | | | |
| | ΔOF1 | ΔCR% | | | | | | | | | | | | | | | ΔOBA | ΔCBA% | | | | | | | | | | | | | | |
| Class | % | T1 | T2 | T3 | T4 | T5 | T6 | T7 | T8 | T9 | T10 | T11 | T12 | T13 | T14 | T15 | % | T1 | T2 | T3 | T4 | T5 | T6 | T7 | T8 | T9 | T10 | T11 | T12 | T13 | T14 | T15 |
|---|---|---|---|---|---|---|---|---|---|---|---|---|---|---|---|---|---|---|---|---|---|---|---|---|---|---|---|---|---|---|---|---|
| T1 | 0.5 | 0.2 | -0.1 | 0.5 | -0.4 | 0.5 | 0.8 | 1.9 | 3.8 | -1.5 | 8.8 | 0.0 | 6.5 | 0.0 | -1.4 | -5.7 | 0.2 | 0.0 | 0.2 | 0.2 | -0.2 | 0.2 | 0.3 | 1.0 | 1.6 | -0.5 | 3.1 | 0.0 | 2.1 | 0.0 | -0.5 | -1.8 |
| T2 | 0.6 | -0.2 | 0.7 | -0.1 | 1.8 | 0.5 | 0.8 | 3.2 | 0.0 | 2.0 | 5.5 | 0.0 | 16.1 | -13.3 | -1.4 | -7.5 | 0.3 | -0.1 | 0.5 | -0.1 | 0.8 | 0.2 | 0.3 | 1.5 | 0.1 | 0.9 | 2.0 | 0.0 | 5.2 | -3.3 | -0.5 | -2.4 |
| T3 | -0.1 | -0.2 | -0.8 | 2.0 | 1.3 | -1.2 | 2.0 | -2.9 | 2.5 | 0.5 | 0.0 | 0.0 | 3.2 | 0.0 | 1.4 | -3.8 | 0.0 | -0.1 | -0.2 | 0.6 | 0.6 | -0.5 | 0.8 | -1.3 | 1.2 | 0.3 | 0.0 | 0.0 | 1.0 | 0.0 | 0.5 | -1.2 |
| T4 | 0.1 | -0.4 | -0.4 | 0.0 | 8.0 | -0.9 | 1.2 | 2.9 | 1.3 | -2.0 | 2.8 | 0.0 | -3.2 | 6.7 | -1.4 | -9.4 | 0.1 | -0.2 | -0.1 | 0.0 | 3.4 | -0.4 | 0.5 | 1.3 | 0.6 | -0.8 | 0.9 | 0.0 | -1.0 | 1.7 | -0.5 | -3.0 |
| T5 | 0.6 | -0.2 | 0.3 | 0.2 | 0.0 | 5.7 | 5.2 | 0.0 | 3.8 | 0.0 | 0.6 | 0.0 | -3.2 | 0.0 | -2.7 | -7.5 | 0.3 | -0.1 | 0.4 | 0.1 | 0.0 | 2.1 | 2.3 | 0.1 | 1.6 | 0.1 | 0.3 | 0.0 | -1.1 | 0.0 | -1.0 | -2.4 |
| T6 | 0.2 | -0.1 | -0.5 | 0.1 | 1.8 | 1.4 | 6.9 | 1.6 | -5.1 | 0.5 | 4.4 | 0.0 | -3.2 | -6.7 | -2.7 | -9.4 | 0.1 | -0.1 | 0.0 | 0.0 | 0.8 | 0.7 | 2.3 | 0.8 | -1.9 | 0.3 | 1.7 | 0.0 | -1.0 | -1.7 | -0.9 | -3.0 |
| T7 | 0.8 | -0.1 | 0.6 | 0.0 | 1.3 | 0.9 | 0.0 | 5.8 | 3.8 | 0.5 | 3.3 | 0.0 | 16.1 | 0.0 | -1.4 | -1.9 | 0.4 | 0.0 | 0.5 | 0.0 | 0.6 | 0.5 | 0.1 | 2.5 | 1.6 | 0.3 | 1.2 | 0.0 | 5.3 | 0.0 | -0.5 | -0.6 |
| T8 | -0.1 | -0.1 | -0.7 | 0.1 | 0.4 | -1.9 | -1.2 | 1.0 | 10.2 | 0.5 | 2.2 | 0.0 | 0.0 | 6.7 | -4.1 | -5.7 | 0.0 | 0.0 | -0.2 | 0.1 | 0.2 | -0.8 | -0.4 | 0.4 | 3.9 | 0.1 | 0.8 | 0.0 | 0.0 | 1.7 | -1.4 | -1.8 |
| T9 | 0.2 | -0.5 | 0.4 | 0.4 | 1.8 | 0.5 | -0.4 | 1.6 | -1.3 | 4.4 | 2.2 | 0.0 | -3.2 | -6.7 | -4.1 | -5.7 | 0.1 | -0.2 | 0.3 | 0.1 | 0.8 | 0.2 | -0.1 | 0.8 | -0.5 | 1.7 | 0.9 | 0.0 | -1.0 | -1.7 | -1.5 | -1.8 |
| T10 | -0.1 | -0.4 | -0.5 | 0.5 | -0.4 | -0.9 | -0.8 | 1.9 | -3.2 | -2.9 | 12.2 | 0.0 | -6.5 | 6.7 | 1.4 | -5.7 | 0.0 | -0.2 | 0.0 | 0.2 | -0.2 | -0.3 | -0.2 | 0.8 | -1.2 | -1.1 | 3.4 | 0.0 | -2.1 | 1.6 | 0.5 | -1.8 |
| T11 | -0.1 | -0.2 | 0.0 | 0.5 | 0.4 | -0.9 | -3.2 | 1.3 | 0.6 | -1.0 | 2.2 | 1.1 | -3.2 | -6.7 | 1.4 | -3.8 | 0.0 | -0.1 | 0.1 | 0.2 | 0.2 | -0.4 | -1.3 | 0.5 | 0.3 | -0.4 | 0.8 | 0.6 | -1.1 | -1.7 | 0.5 | -1.2 |
| T12 | 0.4 | -0.2 | 0.5 | 0.2 | 0.4 | 0.7 | -1.2 | 1.6 | 4.5 | -2.4 | 4.4 | 0.0 | 29.0 | 0.0 | -4.1 | -7.5 | 0.2 | -0.1 | 0.4 | 0.1 | 0.2 | 0.3 | -0.4 | 0.7 | 1.8 | -1.0 | 1.6 | 0.0 | 9.4 | 0.0 | -1.5 | -2.4 |
| T13 | 0.2 | -0.5 | 0.0 | 0.0 | 1.8 | 0.2 | 0.8 | 3.9 | 0.0 | -2.0 | 4.4 | 0.0 | -6.5 | 26.7 | -1.4 | -9.4 | 0.1 | -0.2 | 0.1 | 0.0 | 0.8 | 0.1 | 0.4 | 1.7 | 0.0 | -0.9 | 1.6 | 0.0 | -2.1 | 6.6 | -0.5 | -3.0 |
| T14 | -0.1 | -0.2 | -0.1 | 0.0 | -0.4 | 1.2 | -1.6 | 0.0 | -1.3 | -1.0 | -1.1 | 0.0 | 3.2 | 0.0 | 13.5 | -5.7 | 0.0 | -0.1 | 0.2 | 0.0 | -0.2 | 0.5 | -0.5 | 0.0 | -0.4 | -0.4 | -0.4 | 0.0 | 1.1 | 0.0 | 4.2 | -1.8 |
| T15 | 0.1 | 0.0 | 0.0 | -0.4 | 1.8 | 0.2 | -1.2 | -1.0 | 1.9 | -2.0 | 3.9 | 0.0 | -3.2 | 0.0 | -2.7 | 20.8 | 0.1 | 0.0 | 0.1 | -0.2 | 0.8 | 0.1 | -0.4 | -0.3 | 0.8 | -0.8 | 1.3 | 0.0 | -1.0 | 0.0 | -1.0 | 6.3 |
| M0 | 0.86 | 0.98 | 0.94 | 0.91 | 0.80 | 0.79 | 0.66 | 0.78 | 0.67 | 0.74 | 0.54 | 0.95 | 0.48 | 0.33 | 0.55 | 0.45 | 0.92 | 0.99 | 0.95 | 0.95 | 0.90 | 0.89 | 0.82 | 0.88 | 0.83 | 0.86 | 0.76 | 0.97 | 0.74 | 0.67 | 0.77 | 0.73 |

2(a) shows the results of these 180 combinations. All results were obtained from fixed-time experiments (1 GPU hour).

From the table, we can make several observations:

- Class-based objective metrics ($\phi^{\mathrm{cr}}$ and $\phi^{\mathrm{cba}}$) mostly delivered improvement for the target classes with a few exceptions. For example, when $\phi^{\mathrm{cr}}$ is applied to class T1, strategies SW, HSW, and GA improved the recall of T1 by 0.23, 0.35, and 0.35, respectively.

- Overall objective metrics ($\phi^{\mathrm{oba}}$ and $\phi^{\mathrm{of1}}$) were consistently improved for every class where example messages were selected from.

- Smaller classes, e.g., T12 ∼ T15 were noticeably improved.

- Table 2(b) summarizes the strategies that were considered to be the best for each class-metric combination. While each strategy may appear in several neighboring cells that form a color block,

there is no overwhelmingly superior strategy. This observation leads to **research question 2** in Figure 2.

**Performance Beyond an Objective Metric.** When a strategy-metric combination was applied to a class in an experiment, we also measured the overall performance of the model and its performance on other classes. Table 3 shows such measurements when the combination (SW, $\phi^{\mathrm{cr}}$) was applied to each of the 15 classes.

For example, in the first row T1, the model retrained with synthetic data generated using examples from the training data of T1 not only improves the recall of T1 (i.e., 0.2% in the yellow cell), but also improves the two overall measurements $\phi^{\mathrm{of1}}$ and $\phi^{\mathrm{oba}}$ by 0.5% and 0.2% respectively as well as the class-based measurements $\phi^{\mathrm{cr}}$ and $\phi^{\mathrm{cba}}$ for many other classes. Noticeably, $\phi^{\mathrm{cr}}$ for class T10 was improved by 8.8%.

From the table, we can also observe that some classes, such as T10, T12, and T13, have often benefited from workflows that were intended to improve other classes. Meanwhile, there were many cells with red numbers indicating worsened performance. Note that the results are representative only for single-phase experiments.

With the 12 combinations of three strategies and four objective metrics, there are 11 other tables similar to Table 3. The results of these experiments can be found in Appendix B. Hence, if one wishes to improve $\phi^{\mathrm{cr}}$ for class T1, one can search the column $\Delta$CR%-T1 in all 12 tables for the best performance, and thereby the best combination strategy-metric-class. Any of the 180 cells may potentially offer the best setting for improving $\phi^{\mathrm{cr}}$ for class T1.

The 12 tables of experimental results also inform us that we should not make simple assumptions, such as "*to improve $\phi^{cba}$ of class C, model improvement using metric $\phi^{cba}$ and example messages from class C training data is always the best approach.*" This leads to a solution for **research question 3** (see Section 5).

**Further Experiments.** To ensure what observed in the aforementioned experiments described is not an isolated instance, we conducted further experiments using the same experimental environment with different datasets (TREC-6 (Li & Roth, 2002) and Amazon Reviews 2023 (Hou et al., 2024)) and two alternative methods for generating synthetic data (Easy Data Augmentation (EDA) tool (Wei & Zou, 2019) and AEDA (Karimi et al., 2021)). The results of these experiments can be found in Appendix A.

In general, we can observe a common phenomenon that there is no superior strategy, and we cannot assume a superior objective metric. Meanwhile, the amount of improvement through retraining with synthetic data generated using EDA is normally noticeably lower than with synthetic data generated using GPT-3.5.

## 5 Ensemble Strategy and Further Experiments

The results of our large-scale experiments indicated that there was no superior search strategy or objective metric that could address the typical business requirements outlined at the beginning of Section 3. This led us to **research question 2**. We noticed that the results were highly agreeable when we repeated these single-phase experiments. Although the performance of each combination of [SW, HSW, GA]$\times[\phi^{\mathrm{cr}}, \phi^{\mathrm{cba}}, \phi^{\mathrm{oba}}, \phi^{\mathrm{of1}}]$ depends on many factors (e.g., training data, class size, feature specification, LLMs, and model structure $M^{\square}$), these factors do not change much in the process for improving a model deployed in a specific ticketing system. Therefore, single-phase experiments (such as those reported in Section 4) collect knowledge as to what search strategy and what metric may work for each class. In other words, tables (such as the one in Table 3) are essentially quantitative knowledge maps generated by ML training and testing processes. Once these tables become available, the model improvement process can use them to identify more effective strategies, metrics, and classes for a performance metric, and the process can become more cost-effective.

**Multi-Phase Model Improvement.** Given a model $M_0$ to be improved, the systematic sampling of $[C_1, C_2, \ldots, C_k]\times[\mathrm{SW, HSW, GA}]\times[\phi^{\mathrm{cr}}, \phi^{\mathrm{cba}}, \phi^{\mathrm{oba}}, \phi^{\mathrm{of1}}]$ produces a knowledge map. At the end of this process, likely, it also delivers the most improved model $M_1^*$. Because in this systematic sampling phase, each attempt involves example messages from the training data of only one individual class. Naturally, we can initiate a new process for improving $M_1^*$ obtained in the first phase. As exemplified in Table 3, we can also use examples selected from another class to achieve a requirement for improvement. With 12 such tables,

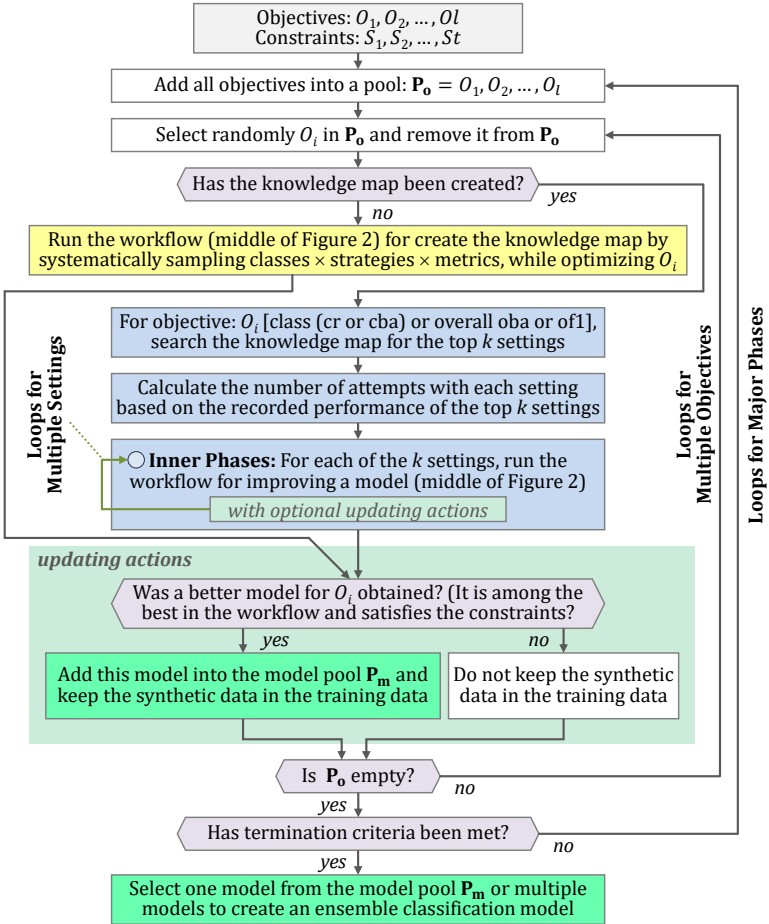

Figure 4: A knowledge-based ensemble algorithm for multi-phase and multi-objective model improvement.

we can also use a different strategy and a different objective metric. Given a single business requirement, we can invoke multiple phases of model improvement with different combinations of strategy, objective metric, and classes for providing example messages.

**Multi-Objective Model Improvement.** Recall the three typical business requirements in Section 3, a model improvement process often has to deal with more than one requirement, e.g., improving the recall of class $X$ and $Y$, while improving the overall accuracy. The experiments reported in Section 4 also indicate that such requirements can conflict with each other if one makes attempts with just one approach (e.g., red numbers in Table 3), while many other combinations of strategy, objective metric, and classes can provide alternative approaches. Once tables similar to Table 3) become available, the model improvement process can use the knowledge stored in these tables to explore different cost-effective approaches to meet multi-objective requirements.

The performance measurements obtained in an early phase can therefore facilitate a knowledge-based algorithm, which provides an answer to **research question 3**.

**Knowledge-based Ensemble Algorithm.** Figure 4 shows the flowchart of an ensemble algorithm for multi-phase and multi-objective model improvement. Given a set of objectives $O_1, O_2, \ldots, O_l$ and a set of constraints $S_1, S_2, \ldots, S_l$, the algorithm begins with the creation of a pool of objectives $\mathbf{P_o}$. The algorithm randomly selects one objective $O_i$ from $\mathbf{P_o}$ each time, aiming to improve the currently-best candidate model with $O_i$ as the focus. The randomness also provides the outer loop of the algorithm with further opportunities to improve the currently-best candidate model.

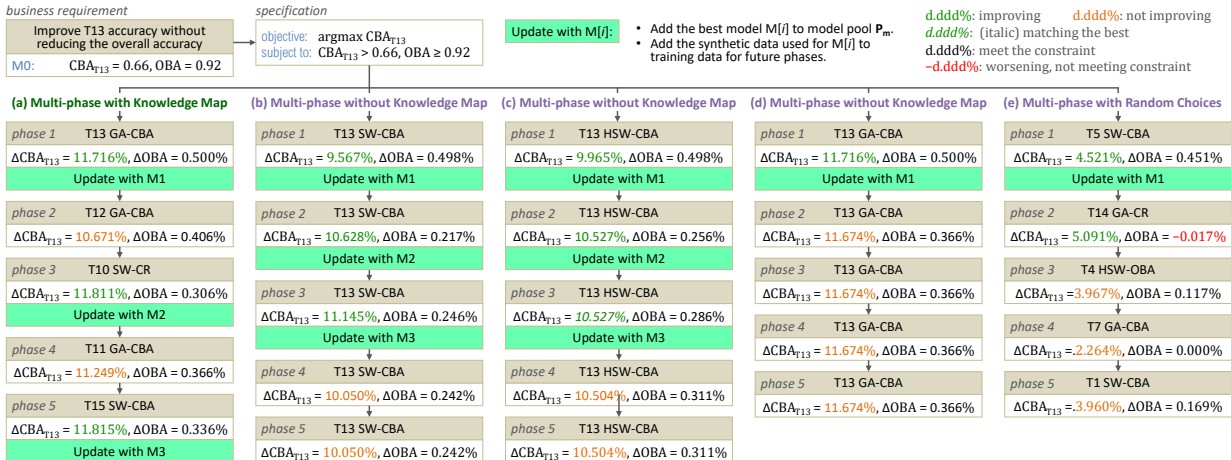

Figure 5: Examples of multi-phase model improvement with and without using a knowledge map.

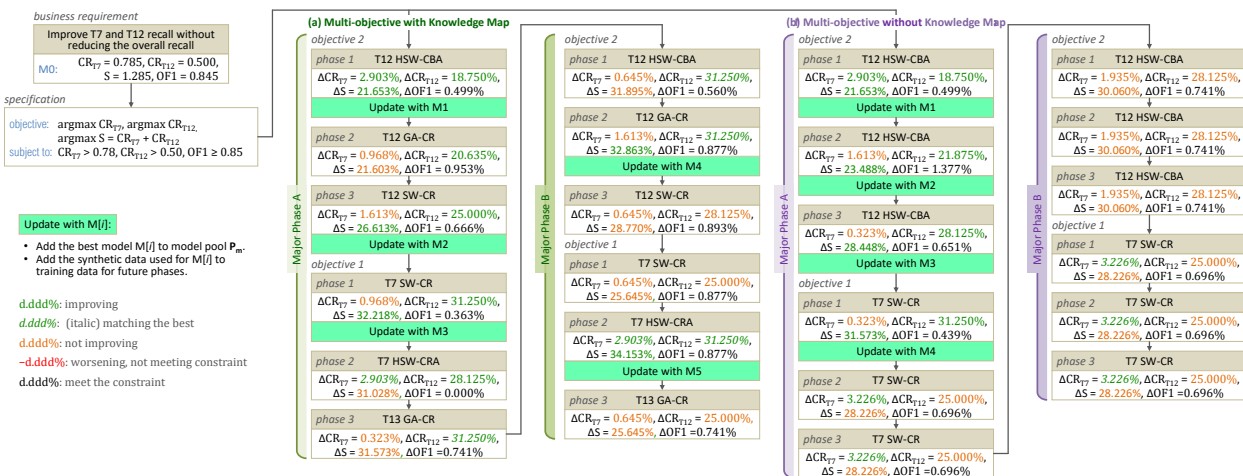

Figure 6: Examples of multi-objective model improvement with and without using a knowledge map.

If the algorithm detects the absence of a knowledge map, it activates the process for creating such a map, while optimizing $O_i$ (i.e., the yellow block). As described in Sections 3 and 4, the process is the middle part of Figure 2 with systematic sampling of all combinations of classes, search strategies, and objective metrics. For example, for the ticketing example, the process may run 180 workflows for 15 classes × 3 strategies × 4 metrics.

If the knowledge map is already there, the algorithm moves directly to the knowledge-based approach (i.e., the three blue blocks). The algorithm selects the top $k > 0$ experimental settings that will likely benefit $O_i$ most according to the knowledge map. Note that the selected settings do not have to use the objective metric $\phi$ defined for $O_i$. For example, if $O_i$ is to improve $\phi_{C11}^{cr}$ (the recall of class $C_{11}$), as long as the knowledge map shows that $\phi_{C13}^{cr}$, $\phi_{C1}^{cba}$, and $\phi_{C7}^{of1}$ can also improve the performance measure of $\phi_{C11}^{cr}$, they can be among the candidate objective metrics to be selected.

The algorithm also distributes computational resources according to the knowledge map. Consider the knowledge map shows that the top $k$ settings previously improved $O_i$ by $x_1\%, x_2\%, \ldots, x_k\%$, the algorithm distributes the computational resources proportionally to the $i$-th setting as $(x_i / \sum x_j)\%$ of the total resources available.

The algorithm adds the best model in each loop to a model pool $\mathbf{P_m}$. At the end of the process, the best model can be selected from $\mathbf{P_m}$ based on a predefined numerical criterion (e.g., a weighted sum of

different objective measurements) or by human experts who brought additional operational knowledge into the decision process. Alternatively, one can select a few best models to create an ensemble classification model.

**Further Experiments and Ablation Studies.** We further conducted ablation studies comparing our knowledge-based ensemble algorithm with baselines that do not use the knowledge base, under the scenarios of multi-phase model improvement for a single objective and for multiple objectives. Figure 5 illustrates the process and results of a set of experiments with a single objective. Here, we use class T13 as an example, which was considered to be important, and the business requires its accuracy to be improved. After the business requirement is translated to an optimization specification, five different workflows were invoked for comparison: (a) knowledge-based ensemble approach, (b,c,d) three brute-force approaches targeting T13 directly with three different strategies (SW, HSW, or GA) repeatedly, and (e) randomly picks a strategy-metric-class combination at each phase. Each approach makes attempts in five phases. For a single objective, one can select the best model according to the highest performance measurement $\Delta \text{CBA}_{\text{T13}}$, i.e., $11.815\%$ resulted from the knowledge-based approach. We conducted similar experiments with different types of single objectives, and the knowledge-based approach performed better in the majority of the experiments.

Figure 6 illustrates the process and results of a set of experiments with two objectives. The business requirements are to improve the recall of classes T7 and T12. The requirements were first translated to maximization of $\phi_{\text{T7}}^{\text{cr}}$ and $\phi_{\text{T12}}^{\text{cr}}$, together with a multi-objective assessment function. For this group of experiments, we define a simple function $S = \phi_{\text{T7}}^{\text{cr}} + \phi_{\text{T12}}^{\text{cr}}$, which will trigger actions of adding a model into the model pool $\mathbf{P_m}$ and adding the corresponding synthetic data to the training data for subsequent phases. In practice, one can define a more complex assessment function and trigger actions, e.g., giving some objectives a high priority, or updating $\mathbf{P_m}$ more frequently than the training data. In the brute-force approach (b) on the right, the first four phases made improvement gradually, and then further improvement became difficult. The knowledge-based approach (a) invoked three different settings for each objective, and achieved improvement in a more distributed manner, reaching the highest $S$ value in the $11^{\text{th}}$ inner phase. We conducted similar experiments with different types of multi-objective settings, and the knowledge-based approach performed better in the majority of the experiments.

## 6 Conclusions

In this work, we developed an automated workflow, AutoGeTS, for improving text classification models, and an approach for using the knowledge discovered in the workflow to guide subsequent model improvement processes. While the work was motivated by challenges in an industrial application, we followed three *research questions*, conducted experiments at a large scale, and validated our findings using public datasets (TREC-6 (Li & Roth, 2002) and Amazon Reviews 2023 (Hou et al., 2024)) in addition to the ticketing data, and tested the AutoGeTS with non-LLM, traditional data augmentation methods such as EDA (Wei & Zou, 2019) and AEDA (Karimi et al., 2021).

In many machine learning workflows, it is common that none of the known individual techniques is superior to others, though most of us are inspired to find the very best solution. A knowledge map is a divide-and-conquer map showing the best solution in each context. Hence, a knowledge-based ensemble is a piecewise optimization of using the best solution(s) in each context. We envisage that this approach can be adapted to many other practical applications.

The work also confirmed that using LLMs to generate synthetic data can address a common challenge in the industrial workflow for improving models for ticketing systems deployed in different organizations. As LLMs are not part of these classification models, they do not add extra burden on the models. As LLMs improve through version updates, they provide sustainable technical support to the AutoGeTS workflow.

In future work, we plan to conduct more large-scale experiments and analyze the results to gain a deep understanding of different factors that make some example messages more effective than others, and we hope the use such analysis as useful knowledge for guiding the development of more intelligent and effective techniques to select examples for generating synthetic data.

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

# APPENDICES

## AutoGeTS: Knowledge-based Automated Generation of Text Synthetics for Improving Text Classification

In the following appendices, we provide further experiment results through visualization plots. The experimental data will be made available on GitHub after the double-blind review process. These appendices include:

- Appendix A provides further experimental results with two additional datasets and with a traditional data augmentation tool. The results confirmed the findings about **research question 1** presented in Section 4.

- Appendix B provides further experimental results in 11 tables. In total, there are 12 tables, only one table (Table 3) was given in the main body of paper. The results in these 12 tables helped us answer **research question 2** and **research question 3**, and provided a knowledge map for the ensemble algorithm presented in Section 5.

- Appendix C provides the hyperparameter study for the original CatBoost model $M_0$ for the industrial ticketing system. The optimal parameter set found in these experiments are then fixed in all other experiments for this dataset shown in Sections 4 and 5.

- Appendix D provides the a pilot study of the *Sliding Window* (SW), *Hierarchical Sliding Window* (HSW), and *Generic Algorithm* (GA) parameters to ensure that these parameters (e.g., window size, crossover method, etc.) of each strategy is fairly optimized. The optimal parameter sets found in these experiments are then fixed in all other experiments for this dataset shown in Sections 4 and 5.

## A  Validation with the EDA & AEDA Tools and TREC-6 and Amazon Datasets

In this appendix, we report further experiments for validating observation about the lack of a superior search strategy from the experiments with the ticketing data (Table 2). We conducted these further experiments with two public datasets: TREC-6 Li & Roth (2002) and Amazon Reviews 2023 Hou et al. (2024), in conjunction with the GPT-3.5's API, the Easy Data Augmentation (EDA) tool Wei & Zou (2019), and the AEDA tool Karimi et al. (2021).

### A.1  TREC-6 Dataset with EDA, AEDA, and GPT-3.5

Table 4: The performance of the original CatBoost model $M_0$ trained on the TREC-6 dataset (without synthetic data).

| Class | Class Size | Balanced Accuracy | Recall | F1-Score |
|-------|-----------|-------------------|--------|----------|
| ENTY  | 1250 | 0.861 | 0.825 | 0.757 |
| HUM   | 1223 | 0.903 | 0.850 | 0.846 |
| DESC  | 1162 | 0.881 | 0.802 | 0.820 |
| NUM   | 896  | 0.908 | 0.836 | 0.866 |
| LOC   | 835  | 0.882 | 0.789 | 0.819 |
| ABBR  | 86   | 0.761 | 0.522 | 0.686 |
| Overall | 5542 | 0.889 | 0.816 | 0.816 |

The TREC-6 dataset Li & Roth (2002) comprises 5542 fact-based questions categorized into six semantic classes with varying class sizes from 86 to 1250 questions. The performance of the original CatBoost classification model trained on TREC-6 (without synthetic data) is shown in Table 4.

Similar to the ticketing data, we used the EDA tool, the AEDA tool, and the GPT-3.5 to generate synthetic data, and ran experiments with all combinations of three strategies, four objective metrics, and six classes

where example questions were selected as prompts. The EDA results were shown in Table 5, the AEDA results were shown in Table 6, and the GPT 3.5 results were shown in Table 7.

Table 5: Systematic experimentation with the TREC-6 dataset and synthetic data generated by the EDA tool. (a) The results of systematic testing of 72 combinations of 3 strategies, 4 objective metrics, and 6 classes in the ticketing dataset. The bars in cells depict the improvement in the range $(0\%, 50\%]$, while red texts indicate worsened performance. (b) The results can be summarized as a knowledge map showing the best strategy or strategies for each metric-class combination. For each map region, the strategies within $0.03\%$ difference from the best strategy are also selected.

(a) the results of systematic testing

| Class | ΔCR% SW | HSW | GA | ΔCBA% SW | HSW | GA | ΔOBA% SW | HSW | GA | ΔOF1% SW | HSW | GA |
|---|---|---|---|---|---|---|---|---|---|---|---|---|
| ENTY | 3.62 | 3.62 | 1.95 | 0.97 | 0.96 | 0.52 | 0.51 | 0.19 | 0.19 | 0.93 | 0.34 | 0.34 |
| HUM | 4.31 | 8.33 | 4.31 | 1.50 | 2.37 | 1.64 | 0.63 | 0.49 | 0.37 | 1.14 | 0.90 | 0.67 |
| DESC | 9.34 | 14.84 | 8.79 | 2.15 | 2.97 | 2.29 | 0.62 | 0.49 | 0.06 | 1.12 | 0.90 | 0.11 |
| NUM | 8.86 | 9.49 | 8.23 | 4.02 | 4.99 | 3.60 | 0.88 | 0.68 | 0.25 | 1.59 | 1.24 | 0.45 |
| LOC | 20.10 | 18.62 | 8.40 | 5.12 | 5.56 | 3.02 | 0.63 | 0.43 | 0.06 | 1.14 | 0.79 | 0.11 |
| ABBR | 20.50 | 20.73 | 25.84 | 8.34 | 8.57 | 8.57 | 0.64 | 0.66 | 0.62 | 1.16 | 1.20 | 1.12 |

(b) a summary of the best strategy-metric combinations

| Class | Measured Performance Metric CR | CBA | OBA | OF1 |
|---|---|---|---|---|
| ENTY | SW/HSW | SW | | |
| HUM | | | SW | SW |
| DESC | HSW | HSW | SW | SW |
| NUM | | HSW | SW | SW |
| LOC | SW | | SW | SW |
| ABBR | GA | SW/GA | HSW | HSW |

Table 6: Systematic experimentation with the TREC-6 dataset and synthetic data generated by the AEDA tool. (a) The results of systematic testing of 72 combinations of 3 strategies, 4 objective metrics, and 6 classes in the ticketing dataset. The bars in cells depict the improvement in the range $(0\%, 50\%]$, while red texts indicate worsened performance. (b) The results can be summarized as a knowledge map showing the best strategy or strategies for each metric-class combination. For each map region, the strategies within $0.03\%$ difference from the best strategy are also selected.

(a) the results of systematic testing

| Class | ΔCR% SW | HSW | GA | ΔCBA% SW | HSW | GA | ΔOBA% SW | HSW | GA | ΔOF1% SW | HSW | GA |
|---|---|---|---|---|---|---|---|---|---|---|---|---|
| ENTY | 4.50 | 4.43 | -2.96 | 1.51 | 0.17 | -1.69 | 0.62 | 0.37 | 0.37 | 1.12 | 0.67 | 0.67 |
| HUM | 1.47 | 12.94 | 0.00 | 0.82 | 0.44 | -0.46 | 0.87 | 0.37 | -0.12 | 1.57 | 0.67 | -0.22 |
| DESC | 2.75 | 21.59 | 2.20 | 1.18 | 1.33 | 0.25 | 0.62 | 0.43 | 0.12 | 1.12 | 0.79 | 0.22 |
| NUM | 6.96 | 15.48 | 5.70 | 3.02 | 3.07 | 2.68 | 0.68 | 0.37 | 0.19 | 1.24 | 0.67 | 0.34 |
| LOC | 7.63 | 21.67 | 0.76 | 3.48 | 2.79 | 0.46 | 0.68 | 0.31 | 0.68 | 1.24 | 0.56 | 1.24 |
| ABBR | 25.00 | 25.64 | 25.00 | 8.57 | 8.57 | 8.57 | 0.68 | 0.62 | 0.87 | 1.24 | 1.12 | 1.57 |

(b) a summary of the best strategy-metric combinations

| Class | Measured Performance Metric CR | CBA | OBA | OF1 |
|---|---|---|---|---|
| ENTY | SW | SW | SW | SW |
| HUM | HSW | SW | SW | SW |
| DESC | HSW | HSW | SW | SW |
| NUM | HSW | HSW | SW | SW |
| LOC | HSW | SW | SW/GA | SW/GA |
| ABBR | HSW | SW/HSW/GA | GA | GA |

Table 7: Systematic experimentation with the TREC-6 dataset and synthetic data generated by the GPT-3.5 API. (a) The results of systematic testing of 72 combinations of 3 strategies, 4 objective metrics, and 6 classes in the ticketing dataset. The bars in cells depict the improvement in the range $(0\%, 50\%]$, while red texts indicate worsened performance. (b) The results can be summarized as a knowledge map showing the best strategy or strategies for each metric-class combination. For each map region, the strategies within $0.03\%$ difference from the best strategy are also selected.

(a) the results of systematic testing

| Class | ΔCR% SW | HSW | GA | ΔCBA% SW | HSW | GA | ΔOBA% SW | HSW | GA | ΔOF1% SW | HSW | GA |
|---|---|---|---|---|---|---|---|---|---|---|---|---|
| ENTY | 7.39 | 5.91 | 3.16 | 2.40 | 1.45 | 0.65 | 0.91 | 0.37 | 0.25 | 1.64 | 0.67 | 0.45 |
| HUM | 14.10 | 13.59 | 7.35 | 3.20 | 3.14 | 2.26 | 0.97 | 0.65 | 0.75 | 1.76 | 1.17 | 1.37 |
| DESC | 25.62 | 21.47 | 11.23 | 4.79 | 4.01 | 2.95 | 0.91 | 0.66 | 0.65 | 1.64 | 1.20 | 1.19 |
| NUM | 15.70 | 15.70 | 15.70 | 5.85 | 5.64 | 5.64 | 1.09 | 0.78 | 0.80 | 1.98 | 1.42 | 1.46 |
| LOC | 18.60 | 20.67 | 14.82 | 5.93 | 6.17 | 4.43 | 1.05 | 0.93 | 0.62 | 1.91 | 1.69 | 1.12 |
| ABBR | 32.38 | 39.40 | 30.08 | 10.31 | 10.31 | 10.31 | 0.91 | 0.99 | 0.99 | 1.64 | 1.80 | 1.79 |

(b) a summary of the best strategy-metric combinations

| Class | Measured Performance Metric CR | CBA | OBA | OF1 |
|---|---|---|---|---|
| ENTY | SW | SW | SW | SW |
| HUM | SW | SW | SW | SW |
| DESC | SW | SW | SW | SW |
| NUM | SW/HSW/GA | SW | SW | SW |
| LOC | HSW | HSW | SW | SW |
| ABBR | HSW | SW/HSW/GA | HSW/GA | HSW/GA |

From these three tables, we can observe:

- The amount of improvement in Table 7 is in general higher than that in Tables 5 and 6, confirming the advantage of using LLMs to generate synthetic data.

- With the synthetic data generated using the EDA tool, the strategy HSW worked better for several classes when classed-measures were used as the objective measures. The strategy SW worked better in most cases when overall measures were used as the objective measures.

- With the synthetic data generated using the AEDA tool, the strategy HSW worked better for most classes when classed-based recall was used as the objective measure. The strategy SW worked better in many cases when overall measures were used as the objective measures. The strategy GA worked better for the ABBR class.

- With the synthetic data generated using the GPT 3.5 API, the strategy SW worked better in most cases when across the whole table. However, there are five cells where SW is not the preferred strategy.

- All three tables confirmed that there is no single superior strategy/metric. So, the knowledge-based approach is needed to enable the selection of more effective strategies for individual classes and objective metrics.

### A.2   Amazon Subset with EDA, AEDA, and GPT-3.5

With the Amazon Reviews'2023 dataset Hou et al. (2024), a subset of 10000 reviews were randomly selected from the Gift Cards subject. The subset has five rating classes with different class sizes ranging from 121 to 8389 reviews. The performance of the original CatBoost classification model trained on this Amazon Reviews subset (without synthetic data) is shown in Table 8.

Table 8: The performance of the original CatBoost model $M_0$ trained on the Amazon Review subset (without synthetic data).

| Class | Class Size | Balanced Accuracy | Recall | F1-Score |
|---|---|---|---|---|
| R1 | 807 | 0.834 | 0.705 | 0.650 |
| R2 | 121 | 0.499 | 0 | 0 |
| R3 | 206 | 0.524 | 0.049 | 0.089 |
| R4 | 477 | 0.586 | 0.174 | 0.288 |
| R5 | 8389 | 0.758 | 0.977 | 0.950 |
| Overall | 10000 | 0.934 | 0.894 | 0.894 |

We can notice that for class R2, there was no true positive results, and the recall and F1-score are both 0. For R3 and R4, the measure of the recall and F1-score measures are very low. Because these are small classes in comparison with R5, misleadingly, the overall measures appear fairly reasonable.

Similar to the ticketing data and TREC-6 data, we used the EDA tool, the AEDA tool, and the GPT-3.5 to generate synthetic data, and ran experiments with all combinations of three strategies, four objective metrics, and five classes where example reviews were selected as prompts. The direct testing results of the experiments with the EDA tool were shown in Table 9, the results with the AEDA tool were shown in Table 10, and the results with the GPT 3.5 API were shown in Table 11. Note that in all three tables, the recall values (CR) for class R2 are now above zero. Clearly synthetic data helped.

Table 9: Amazon Reviews Subset EDA-Based Results

| Class | CR | | | CBA | | | OBA | | | OF1 | | |
|---|---|---|---|---|---|---|---|---|---|---|---|---|
| | SW | HSW | GA | SW | HSW | GA | SW | HSW | GA | SW | HSW | GA |
| R1 | 0.732 | 0.763 | 0.732 | 0.845 | 0.852 | 0.846 | 0.936 | 0.936 | 0.938 | 0.898 | 0.898 | 0.900 |
| R2 | 0.067 | 0.053 | 0.067 | 0.526 | 0.526 | 0.526 | 0.938 | 0.938 | 0.938 | 0.901 | 0.900 | 0.900 |
| R3 | 0.122 | 0.098 | 0.098 | 0.560 | 0.560 | 0.559 | 0.937 | 0.937 | 0.937 | 0.900 | 0.899 | 0.899 |
| R4 | 0.221 | 0.233 | 0.233 | 0.601 | 0.603 | 0.603 | 0.936 | 0.938 | 0.936 | 0.898 | 0.900 | 0.898 |
| R5 | 0.984 | 0.984 | 0.984 | 0.782 | 0.784 | 0.784 | 0.938 | 0.938 | 0.938 | 0.900 | 0.901 | 0.901 |

Similarly we showed the amount of improvement in Table 12 and 14 respectively. From these two tables, we can observe:

- The amount of improvement in Table 14 is in general higher than that in Tables 12 and 13, confirming the advantage of using LLMs to generate synthetic data.

- Using synthetic data resulted in noticeable improvement on the recall measures of classes R2, R3, and R4.

- There was no single obvious superior strategy in all three tables. This confirmed that the knowledge-based approach is needed to enable the selection of more effective strategies for individual classes and objective metrics.

Table 10: Amazon Reviews Subset AEDA-Based Results

| | CR | | | CBA | | | OBA | | | OF1 | | |
|-------|-------|-------|-------|-------|-------|-------|-------|-------|-------|-------|-------|-------|
| Class | SW | HSW | GA | SW | HSW | GA | SW | HSW | GA | SW | HSW | GA |
| R1 | 0.739 | 0.788 | 0.651 | 0.852 | 0.853 | 0.813 | 0.938 | 0.938 | 0.937 | 0.901 | 0.901 | 0.899 |
| R2 | 0.105 | 0.067 | 0.158 | 0.578 | 0.578 | 0.552 | 0.938 | 0.938 | 0.937 | 0.900 | 0.900 | 0.900 |
| R3 | 0.122 | 0.188 | 0.098 | 0.560 | 0.549 | 0.548 | 0.938 | 0.937 | 0.938 | 0.900 | 0.900 | 0.901 |
| R4 | 0.233 | 0.348 | 0.197 | 0.615 | 0.614 | 0.597 | 0.937 | 0.938 | 0.938 | 0.900 | 0.901 | 0.900 |
| R5 | 0.984 | 0.987 | 0.975 | 0.811 | 0.812 | 0.825 | 0.938 | 0.938 | 0.934 | 0.901 | 0.901 | 0.895 |

Table 11: Amazon Reviews Subset LLM-Based Results

| | CR | | | CBA | | | OBA | | | OF1 | | |
|-------|-------|-------|-------|-------|-------|-------|-------|-------|-------|-------|-------|-------|
| Class | SW | HSW | GA | SW | HSW | GA | SW | HSW | GA | SW | HSW | GA |
| R1 | 0.745 | 0.797 | 0.745 | 0.855 | 0.866 | 0.854 | 0.938 | 0.938 | 0.938 | 0.901 | 0.902 | 0.901 |
| R2 | 0.158 | 0.158 | 0.158 | 0.579 | 0.579 | 0.579 | 0.940 | 0.939 | 0.938 | 0.904 | 0.902 | 0.901 |
| R3 | 0.146 | 0.188 | 0.146 | 0.573 | 0.570 | 0.571 | 0.938 | 0.938 | 0.938 | 0.901 | 0.902 | 0.901 |
| R4 | 0.348 | 0.348 | 0.326 | 0.615 | 0.614 | 0.608 | 0.938 | 0.938 | 0.938 | 0.901 | 0.901 | 0.901 |
| R5 | 0.989 | 0.997 | 0.995 | 0.828 | 0.822 | 0.825 | 0.938 | 0.939 | 0.940 | 0.902 | 0.903 | 0.905 |

Table 12: Systematic experimentation with a subset of the Amazon Review dataset and synthetic data generated by the EDA tool. (a) The results of systematic testing of 60 combinations of 3 strategies, 4 objective metrics, and 5 classes in the ticketing dataset. The bars in cells depict the improvement in the range $(0\%, 50\%]$, while red texts indicate worsened performance. For the six dark green cells on the left, the original recall measures were zero or near zero. Hence the improvement is well over 50%. The $\Delta$CR% values are shown as white text over dark green background. ###### indicates infinity. (b) The results can be summarized as a knowledge map showing the best strategy or strategies for each metric-class combination. For each map region, the strategies within 0.03% difference from the best strategy are also selected.

(a) the results of systematic testing

(b) a summary of the best strategy-metric combinations

| Class | ΔCR % SW | HSW | GA | ΔCBA % SW | HSW | GA | ΔOBA % SW | HSW | GA | ΔOF1 % SW | HSW | GA |
|---|---|---|---|---|---|---|---|---|---|---|---|---|
| R1 | 3.81 | 8.23 | 3.81 | 1.38 | 2.25 | 1.48 | 0.27 | 0.27 | 0.40 | 0.45 | 0.45 | 0.67 |
| R2 | ###### | ###### | ###### | 5.37 | 5.37 | 5.37 | 0.44 | 0.40 | 0.40 | 0.73 | 0.67 | 0.67 |
| R3 | 150.0 | 100.0 | 100.0 | 6.98 | 6.89 | 6.64 | 0.37 | 0.33 | 0.33 | 0.62 | 0.56 | 0.56 |
| R4 | 26.67 | 33.33 | 33.33 | 2.54 | 2.80 | 2.76 | 0.23 | 0.40 | 0.27 | 0.39 | 0.67 | 0.45 |
| R5 | 0.66 | 0.72 | 0.66 | 3.10 | 3.40 | 3.43 | 0.40 | 0.44 | 0.44 | 0.67 | 0.73 | 0.73 |

| Class | Measured Performance Metric CR | CBA | OBA | OF1 |
|---|---|---|---|---|
| R1 | HSW | HSW | GA | GA |
| R2 | SW/GA | SW/HSW/GA | SW | SW |
| R3 | SW | SW | SW | SW |
| R4 | HSW/GA | HSW | HSW | HSW |
| R5 | HSW | GA | HSW/GA | HSW/GA |

Table 13: Systematic experimentation with a subset of the Amazon Review dataset and synthetic data generated by the AEDA tool. (a) The results of systematic testing of 60 combinations of 3 strategies, 4 objective metrics, and 5 classes in the ticketing dataset. The bars in cells depict the improvement in the range $(0\%, 50\%]$, while red texts indicate worsened performance. For the six dark green cells on the left, the original recall measures were zero or near zero. Hence the improvement is well over 50%. The $\Delta$CR% values are shown as white text over dark green background. ###### indicates infinity. (b) The results can be summarized as a knowledge map showing the best strategy or strategies for each metric-class combination. For each map region, the strategies within 0.03% difference from the best strategy are also selected.

(a) the results of systematic testing

(b) a summary of the best strategy-metric combinations

| Class | ΔCR % SW | HSW | GA | ΔCBA % SW | HSW | GA | ΔOBA % SW | HSW | GA | ΔOF1 % SW | HSW | GA |
|---|---|---|---|---|---|---|---|---|---|---|---|---|
| R1 | 4.76 | 11.84 | -7.62 | 2.11 | 2.27 | -2.47 | 0.47 | 0.44 | 0.30 | 0.78 | 0.73 | 0.50 |
| R2 | ###### | ###### | ###### | 10.64 | 15.91 | 15.86 | 0.40 | 0.40 | 0.37 | 0.67 | 0.67 | 0.62 |
| R3 | 150.0 | 284.4 | 100.0 | 6.93 | 4.75 | 4.61 | 0.40 | 0.37 | 0.44 | 0.67 | 0.62 | 0.73 |
| R4 | 33.33 | 99.7 | 13.33 | 4.82 | 4.65 | 1.80 | 0.37 | 0.44 | 0.40 | 0.62 | 0.73 | 0.67 |
| R5 | 0.72 | 1.05 | -0.24 | 7.01 | 7.10 | 8.87 | 0.44 | 0.44 | 0.07 | 0.73 | 0.73 | 0.11 |

| Class | Measured Performance Metric CR | CBA | OBA | OF1 |
|---|---|---|---|---|
| R1 | HSW | HSW | SW | SW |
| R2 | GA | HSW | SW/HSW | SW/HSW |
| R3 | HSW | SW | HSW | HSW |
| R4 | HSW | SW | GA | GA |
| R5 | GA | GA | SW/HSW | SW/HSW |

# B  Impact on Other Performance Metrics

As mentioned in Section 4, with the ticketing data, there are 180 combinations of three strategies, four objective metrics, and 15 classes where examples were selected from their training data as the prompts for generating synthetic data. Given each of these combinations, in addition to the objective metric on the specific class, we also measure other performance metrics, including two overall metrics (overall balanced accuracy (OBA) and overall F1-score (OF1)) as well as two class-based metrics (recall (CR) and balanced accuracy (CBA)) on all classes. Table 3 showed the cross-impact results of 15 of the 180 combinations, i.e., SW strategy, $\phi^{\text{cr}}$ (class-based recall), and 15 classes. In this appendix, we provide the the cross-impact results for the other $15 \times 11$ combinations in Tables 15 $\sim$ 25.

In addition, from these tables, we can also observe many interesting phenomena, for example:

- Most targeted classes, i.e., the yellow cells in these tables benefited from the improvement process with positive values (black numbers) in these cell. Red numbers in yellow cells are in general uncommon.

- The recall measure of T12 is often improved noticeable even when it was not the target class, e.g., in Tables 3, 17, 18, 20, 21, 22, 23, 24, and 25, where the green bars in the column $\Delta$CR% T12 are noticeable.

Table 14: Systematic experimentation with a subset of the Amazon Review dataset and synthetic data generated by the GPT-3.5 API. (a) The results of systematic testing of 60 combinations of 3 strategies, 4 objective metrics, and 5 classes in the ticketing dataset. The bars in cells depict the improvement in the range $(0\%, 50\%)$, while red texts indicate worsened performance. For the nine dark green cells on the left, the original recall measures were 0 or near zero. Hence the improvement is well over 50%. The $\Delta$CR% values are shown as white text over dark green background. ###### indicates infinity. (b) The results can be summarized as a knowledge map showing the best strategy or strategies for each metric-class combination. For each map region, the strategies within 0.03% difference from the best strategy are also selected.

(a) the results of systematic testing

| Class | ΔCR % | | | ΔCBA % | | | ΔOBA % | | | ΔOF1 % | | |
|---|---|---|---|---|---|---|---|---|---|---|---|---|
| | SW | HSW | GA | SW | HSW | GA | SW | HSW | GA | SW | HSW | GA |
| R1 | 5.71 | 13.04 | 5.71 | 2.58 | 3.85 | 2.38 | 0.47 | 0.50 | 0.47 | 0.78 | 0.84 | 0.78 |
| R2 | ###### | ###### | ###### | 15.86 | 15.91 | 15.86 | 0.64 | 0.54 | 0.47 | 1.06 | 0.89 | 0.78 |
| R3 | 200.0 | 284.4 | 200.0 | 9.31 | 8.82 | 8.92 | 0.44 | 0.50 | 0.47 | 0.73 | 0.84 | 0.78 |
| R4 | 99.7 | 99.7 | 86.8 | 4.82 | 4.65 | 3.70 | 0.47 | 0.47 | 0.44 | 0.78 | 0.78 | 0.73 |
| R5 | 1.20 | 2.04 | 1.81 | 9.21 | 8.44 | 8.85 | 0.50 | 0.60 | 0.70 | 0.84 | 1.01 | 1.17 |

(b) a summary of the best strategy-metric combinations

| Class | Measured Performance Metric | | | |
|---|---|---|---|---|
| | CR | CBA | OBA | OF1 |
| R1 | HSW | HSW | HSW | HSW |
| R2 | SW/HSW/GA | HSW | SW | SW |
| R3 | HSW | SW | HSW | HSW |
| R4 | SW/HSW | SW | SW/HSW | SW/HSW |
| R5 | HSW | SW | GA | GA |

Table 15: Similar to Table 3, when a model is improved with the HSW strategy and objective metric $\phi^{\text{cr}}$ (class-based recall) for classes $T1, T2, \ldots, T15$ in the ticketing dataset, the direct improvement of the target classes can be seen in the yellow cells on the left part of the table. Meanwhile, there are positive and negative impact on other classes and other performance metrics. The green bars in cells depict the improvement in the range $(0\%, 50\%)$, while red texts indicate negative impact. The last row shows the baseline performance.

Optimization: HSW-CR-Class[row]; Performance Metric: ΔOF1%, ΔCR%

| Class | ΔOF1 % | T1 | T2 | T3 | T4 | T5 | T6 | T7 | T8 | T9 | T10 | T11 | T12 | T13 | T14 | T15 |
|---|---|---|---|---|---|---|---|---|---|---|---|---|---|---|---|---|
| | % | ΔCR % | | | | | | | | | | | | | | |
| T1 | -0.4 | 0.4 | -0.4 | 0.6 | 1.3 | -0.9 | -2.8 | 1.6 | -5.1 | 0.0 | 1.1 | 0.0 | 9.7 | -20.0 | -6.8 | -28.3 |
| T2 | 0.5 | -0.2 | 0.8 | 0.4 | 0.4 | 0.0 | 1.6 | 1.6 | 2.5 | -2.4 | 2.8 | 0.0 | 3.2 | 0.0 | 0.0 | -1.9 |
| T3 | -0.1 | -0.2 | -0.8 | 2.0 | 1.3 | -1.2 | 2.0 | -2.9 | 2.5 | 0.5 | 0.0 | 0.0 | 3.2 | 0.0 | 1.4 | -3.8 |
| T4 | 0.1 | -0.2 | -0.5 | 0.0 | 9.8 | -0.7 | -2.8 | 1.6 | 1.9 | -2.9 | 5.5 | 0.0 | 9.7 | -13.3 | -2.7 | -7.5 |
| T5 | 0.3 | -0.5 | 0.4 | 0.4 | 0.9 | 4.3 | 0.0 | 1.3 | 0.0 | -2.9 | 0.6 | 0.0 | 6.5 | -6.7 | 1.4 | -3.8 |
| T6 | 0.1 | -0.4 | 0.1 | -0.2 | 1.8 | 1.7 | 6.5 | 0.3 | -1.9 | -2.4 | 0.6 | 0.0 | -3.2 | 0.0 | -1.4 | -9.4 |
| T7 | 0.7 | -0.2 | 0.7 | 0.2 | 2.7 | 1.7 | 1.2 | 5.8 | 3.2 | -3.4 | 3.9 | 0.0 | 6.5 | -6.7 | -1.4 | -5.7 |
| T8 | -0.2 | -0.2 | -0.6 | -0.7 | 1.3 | -0.9 | -1.2 | -0.6 | 11.5 | 0.5 | 1.7 | 0.0 | -6.5 | -6.7 | -4.1 | -7.5 |
| T9 | 0.4 | -0.2 | 0.4 | 0.1 | 1.3 | 1.9 | -2.0 | 1.9 | -2.5 | 5.4 | 4.4 | 0.0 | 9.7 | 0.0 | -6.8 | -5.7 |
| T10 | 0.1 | -0.1 | -0.1 | 0.4 | -1.3 | 0.0 | 0.8 | 0.0 | -2.5 | 0.5 | 11.0 | 0.0 | -6.5 | 0.0 | -6.8 | -7.5 |
| T11 | 0.6 | -0.3 | 0.1 | 0.9 | 2.7 | 1.9 | 0.4 | 2.6 | 2.5 | -2.9 | 9.4 | 1.1 | 3.2 | -6.7 | 2.7 | -9.4 |
| T12 | 0.3 | -0.2 | 0.0 | 0.4 | 3.6 | 0.9 | -2.4 | 1.3 | -0.6 | 0.0 | 3.9 | 0.0 | 32.3 | -6.7 | -1.4 | -7.5 |
| T13 | 0.0 | -0.3 | -0.2 | 0.0 | 0.9 | 0.7 | -3.2 | 0.3 | 3.2 | 0.5 | 2.2 | 0.0 | -3.2 | 33.3 | 0.0 | -1.9 |
| T14 | -1.0 | -0.2 | -1.1 | -0.5 | 2.2 | -1.2 | -3.6 | 0.0 | -6.4 | -4.4 | -5.0 | 0.0 | -12.9 | -6.7 | 16.2 | -11.3 |
| T15 | 0.2 | -0.3 | -0.3 | 0.5 | 1.8 | 0.0 | -4.4 | 2.6 | 1.9 | -1.0 | 3.3 | 0.0 | 9.7 | -13.3 | 1.4 | 20.8 |
| M0 | 0.86 | 0.98 | 0.94 | 0.91 | 0.80 | 0.79 | 0.66 | 0.78 | 0.67 | 0.74 | 0.54 | 0.95 | 0.48 | 0.33 | 0.55 | 0.45 |

Optimization: HSW-CR-Class[row]; Performance Metric: ΔOBA%, ΔCBA%

| Class | ΔOBA % | T1 | T2 | T3 | T4 | T5 | T6 | T7 | T8 | T9 | T10 | T11 | T12 | T13 | T14 | T15 |
|---|---|---|---|---|---|---|---|---|---|---|---|---|---|---|---|---|
| | % | ΔCBA % | | | | | | | | | | | | | | |
| T1 | -0.2 | -0.2 | -0.1 | 0.2 | 0.5 | -0.5 | -1.1 | 0.7 | -1.9 | 0.1 | 0.3 | 0.0 | 3.1 | -5.0 | -2.4 | -8.9 |
| T2 | 0.2 | -0.1 | 0.4 | 0.2 | 0.2 | 0.0 | 0.7 | 0.8 | 1.1 | -1.0 | 0.9 | 0.0 | 1.1 | 0.0 | 0.1 | -0.6 |
| T3 | 0.0 | -0.1 | -0.2 | 0.6 | 0.6 | -0.5 | 0.8 | -1.3 | 1.2 | 0.3 | 0.0 | 0.0 | 1.0 | 0.0 | 0.5 | -1.2 |
| T4 | 0.0 | -0.1 | 0.0 | 0.0 | 4.1 | -0.3 | -1.1 | 0.7 | 0.8 | -1.2 | 1.9 | 0.0 | 3.2 | -3.3 | -0.9 | -2.4 |
| T5 | 0.2 | -0.2 | 0.3 | 0.1 | 0.4 | 1.6 | 0.2 | 0.7 | 0.1 | -1.2 | 0.1 | 0.0 | 2.1 | -1.7 | 0.5 | -1.2 |
| T6 | 0.1 | -0.2 | 0.2 | -0.1 | 0.8 | 0.8 | 2.4 | 0.2 | -0.7 | -1.0 | 0.2 | 0.0 | -1.1 | 0.0 | -0.5 | -3.0 |
| T7 | 0.4 | -0.1 | 0.5 | 0.1 | 1.2 | 0.8 | 0.6 | 2.6 | 1.3 | -1.4 | 1.4 | 0.0 | 2.1 | -1.7 | -0.5 | -1.8 |
| T8 | -0.1 | -0.1 | -0.3 | -0.4 | 0.6 | -0.3 | -0.5 | -0.3 | 4.3 | 0.3 | 0.6 | 0.0 | -2.1 | -1.7 | -1.4 | -2.3 |
| T9 | 0.2 | -0.1 | 0.2 | 0.1 | 0.6 | 0.9 | -0.7 | 0.9 | -1.0 | 2.2 | 1.7 | 0.0 | 3.2 | 0.0 | -2.4 | -1.8 |
| T10 | 0.0 | 0.0 | 0.2 | 0.2 | -0.6 | -0.1 | 0.5 | 0.1 | -0.9 | 0.2 | 3.2 | 0.0 | -2.1 | 0.0 | -2.4 | -2.3 |
| T11 | 0.3 | -0.2 | 0.4 | 0.4 | 1.2 | 0.9 | 0.3 | 1.2 | 1.1 | -1.1 | 3.2 | 0.5 | 1.0 | -1.7 | 0.9 | -3.0 |
| T12 | 0.1 | -0.1 | 0.2 | 0.1 | 1.5 | 0.4 | -0.8 | 0.6 | -0.1 | 0.0 | 1.3 | 0.0 | 10.4 | -1.7 | -0.4 | -2.4 |
| T13 | 0.0 | -0.1 | -0.1 | 0.0 | 0.4 | 0.3 | -1.3 | 0.2 | 1.2 | 0.2 | 0.8 | 0.0 | -1.0 | 8.2 | 0.0 | -0.6 |
| T14 | -0.5 | -0.1 | -0.4 | -0.3 | 1.0 | -0.5 | -1.5 | 0.0 | -2.5 | -1.9 | -1.8 | 0.0 | -4.2 | -1.7 | 5.1 | -3.5 |
| T15 | 0.1 | -0.1 | 0.1 | 0.2 | 0.8 | -0.1 | -1.7 | 1.1 | 0.8 | -0.3 | 1.1 | 0.0 | 3.1 | -3.3 | 0.5 | 6.1 |
| M0 | 0.92 | 0.99 | 0.95 | 0.95 | 0.90 | 0.89 | 0.82 | 0.88 | 0.83 | 0.86 | 0.76 | 0.97 | 0.74 | 0.67 | 0.77 | 0.73 |

**Most importantly, the data in such tables provides the ensemble algorithm in Section 5 with the knowledge map.**

Table 16: Similar to Table 3 and Table 15, this table shows the impact of the GA strategy and objective metric $\phi^{\mathrm{cr}}$ (class-based recall) for classes $T1, T2, \ldots, T15$ in the ticketing dataset.

| | Optimization: GA-CR-Class[row]; Performance Metric: ΔOF1%, ΔCR% | | | | | | | | | | | | | | | | Optimization: GA-CR-Class[row]; Performance Metric: ΔOBA%, ΔCBA% | | | | | | | | | | | | | | | |
| | ΔOF1 | ΔCR % | | | | | | | | | | | | | | | ΔOBA | ΔCBA % | | | | | | | | | | | | | | |
| Class | % | T1 | T2 | T3 | T4 | T5 | T6 | T7 | T8 | T9 | T10 | T11 | T12 | T13 | T14 | T15 | % | T1 | T2 | T3 | T4 | T5 | T6 | T7 | T8 | T9 | T10 | T11 | T12 | T13 | T14 | T15 |
| T1 | 0.2 | 0.4 | -0.6 | 0.8 | 1.3 | -0.7 | -0.4 | 1.3 | -2.5 | 0.5 | 10.5 | -1.1 | 3.2 | 0.0 | -4.1 | -13.2 | 0.1 | -0.1 | 0.0 | 0.4 | 0.5 | -0.3 | -0.1 | 0.6 | -1.0 | 0.3 | 3.7 | 0.0 | 1.0 | 0.0 | -1.5 | -4.1 |
| T2 | 0.0 | 0.4 | 0.8 | -0.5 | -0.9 | 0.9 | -2.8 | -0.6 | -5.7 | 1.0 | 5.0 | 0.0 | 3.2 | -13.3 | -2.7 | -17.0 | 0.0 | 0.2 | -0.1 | -0.2 | -0.4 | 0.4 | -0.9 | -0.2 | -2.3 | 0.6 | 1.8 | 0.0 | 1.0 | -3.3 | -1.0 | -5.3 |
| T3 | 0.3 | -0.2 | 0.0 | 2.0 | 1.3 | 0.5 | 0.8 | 1.0 | -0.6 | -1.5 | 3.9 | 0.0 | -6.5 | -6.7 | -5.4 | -5.7 | 0.1 | -0.1 | -0.1 | 0.7 | 0.6 | 0.2 | 0.3 | 0.5 | -0.2 | -0.5 | 1.6 | 0.0 | -2.1 | -1.7 | -1.9 | -1.7 |
| T4 | 0.4 | 0.0 | 0.0 | 1.1 | 8.4 | -0.7 | 0.8 | 1.0 | -2.5 | 0.5 | 3.3 | 0.0 | -3.2 | 0.0 | -2.7 | -3.8 | 0.2 | 0.1 | 0.2 | 0.5 | 3.6 | -0.3 | 0.2 | 0.5 | -0.9 | 0.2 | 1.3 | 0.0 | -1.1 | 0.0 | -0.9 | -1.2 |
| T5 | 0.2 | -0.3 | -0.3 | 0.9 | 0.4 | 2.8 | 1.2 | 1.0 | -2.5 | 2.4 | 3.3 | 0.0 | 3.2 | -20.0 | -1.4 | -9.4 | 0.1 | -0.1 | 0.3 | 0.6 | 0.2 | 0.7 | 0.6 | 0.4 | -0.9 | 1.1 | 1.2 | 0.0 | 1.0 | -5.0 | -0.5 | -3.0 |
| T6 | -0.1 | 0.0 | 0.0 | 0.2 | 0.0 | 0.9 | 4.8 | 1.0 | -2.5 | -3.4 | 0.0 | 0.0 | -3.2 | -13.3 | -4.1 | -15.1 | 0.0 | 0.0 | 0.2 | 0.1 | 0.0 | 0.3 | 1.2 | 0.5 | -0.8 | -1.4 | 0.2 | 0.0 | -1.0 | -3.3 | -1.4 | -4.7 |
| T7 | 0.0 | 0.0 | 0.0 | 0.4 | 0.0 | 0.5 | 2.4 | -0.3 | 1.9 | -2.0 | 1.7 | 0.0 | -3.2 | 0.0 | -6.8 | -5.7 | 0.0 | 0.0 | 0.1 | 0.2 | 0.1 | 0.1 | 0.9 | -0.3 | 0.8 | -0.8 | 0.6 | 0.0 | -1.1 | 0.0 | -2.4 | -1.8 |
| T8 | -0.4 | -0.2 | -0.5 | 0.1 | -0.9 | 1.2 | -3.2 | 0.6 | 5.1 | -2.0 | -4.4 | 0.0 | 3.2 | -6.7 | 0.0 | -9.4 | -0.2 | -0.1 | -0.2 | 0.1 | -0.4 | 0.5 | -1.3 | 0.3 | 1.7 | -0.8 | -1.5 | 0.0 | 1.0 | -1.7 | 0.0 | -3.0 |
| T9 | -0.1 | -0.2 | 0.1 | 0.2 | -0.4 | -0.2 | 0.0 | 1.3 | -3.2 | 2.9 | -2.2 | 0.0 | 6.5 | 0.0 | -1.4 | -9.4 | 0.0 | -0.1 | 0.2 | 0.1 | -0.2 | -0.1 | 0.0 | 0.6 | -1.2 | 0.9 | -0.7 | 0.0 | 2.1 | 0.0 | -0.4 | -3.0 |
| T10 | 0.1 | -0.5 | -0.2 | 0.2 | -0.4 | 1.2 | -0.4 | 1.0 | -3.2 | 0.5 | 12.1 | -1.1 | 3.2 | -6.7 | -1.4 | -11.3 | 0.0 | -0.2 | 0.2 | 0.1 | -0.2 | 0.5 | 0.0 | 0.5 | -1.2 | 0.3 | 3.7 | -0.5 | 1.1 | -1.6 | -0.4 | -3.5 |
| T11 | 0.2 | -0.1 | 0.1 | 0.5 | 0.0 | -1.9 | 3.2 | 2.6 | 1.9 | -2.4 | 3.3 | 1.1 | -3.2 | 0.0 | 0.0 | -11.3 | 0.1 | 0.0 | 0.1 | 0.2 | 0.0 | -0.9 | 1.2 | 1.1 | 0.8 | -0.9 | 1.1 | 0.5 | -1.0 | 0.0 | 0.0 | -3.6 |
| T12 | 0.1 | -0.4 | -0.2 | 0.2 | 1.8 | 1.7 | -0.4 | 1.0 | 0.0 | -2.0 | 1.7 | 0.0 | 32.3 | 6.7 | -1.4 | -5.7 | 0.1 | -0.2 | 0.1 | 0.1 | 0.8 | 0.7 | -0.2 | 0.4 | 0.1 | -0.9 | 0.6 | 0.0 | 10.4 | 1.7 | -0.5 | -1.8 |
| T13 | 0.1 | -0.6 | 0.1 | 0.2 | 1.3 | -1.4 | -2.0 | 5.2 | 1.9 | -2.4 | 1.1 | 0.0 | 3.2 | 33.3 | -1.4 | -3.8 | 0.0 | -0.3 | 0.3 | 0.1 | 0.5 | -0.6 | -0.8 | 2.2 | 0.7 | -1.0 | 0.4 | 0.0 | 1.0 | 8.2 | -0.4 | -1.2 |
| T14 | 0.0 | -0.2 | 0.1 | 0.1 | 0.4 | -0.9 | -0.8 | 0.6 | -2.5 | 0.0 | -1.7 | 0.0 | 0.0 | -13.3 | 16.2 | -5.7 | 0.0 | -0.1 | 0.2 | 0.1 | 0.2 | -0.3 | -0.2 | 0.3 | -1.0 | 0.1 | -0.5 | 0.0 | 0.0 | -3.3 | 5.1 | -1.8 |
| T15 | 0.1 | -0.4 | 0.0 | 0.4 | 1.3 | 0.2 | -1.6 | 1.0 | -3.8 | -1.5 | 2.8 | 0.0 | 3.2 | -6.7 | 0.0 | 24.5 | 0.1 | -0.1 | 0.2 | 0.1 | 0.6 | 0.1 | -0.5 | 0.4 | -1.5 | -0.6 | 0.9 | 0.0 | 1.1 | -1.7 | 0.0 | 7.3 |
| M0 | 0.86 | 0.98 | 0.94 | 0.91 | 0.80 | 0.79 | 0.66 | 0.78 | 0.67 | 0.74 | 0.54 | 0.95 | 0.48 | 0.33 | 0.55 | 0.45 | 0.92 | 0.99 | 0.95 | 0.95 | 0.90 | 0.89 | 0.82 | 0.88 | 0.83 | 0.86 | 0.76 | 0.97 | 0.74 | 0.67 | 0.77 | 0.73 |

Table 17: Similar to Table 3 and Tables 15∼16, this table shows the impact of the SW strategy and objective metric $\phi^{\mathrm{cba}}$ (class-based balanced accuracy) for classes $T1, T2, \ldots, T15$ in the ticketing dataset.

| | Optimization: SW-CBA-Class[row]; Performance Metric: ΔOF1%, ΔCR% | | | | | | | | | | | | | | | | Optimization: SW-CBA-Class[row]; Performance Metric: ΔOBA%, ΔCBA% | | | | | | | | | | | | | | | |
| | ΔOF1 | ΔCR % | | | | | | | | | | | | | | | ΔOBA | ΔCBA % | | | | | | | | | | | | | | |
| Class | % | T1 | T2 | T3 | T4 | T5 | T6 | T7 | T8 | T9 | T10 | T11 | T12 | T13 | T14 | T15 | % | T1 | T2 | T3 | T4 | T5 | T6 | T7 | T8 | T9 | T10 | T11 | T12 | T13 | T14 | T15 |
| T1 | -0.1 | -0.1 | 0.1 | 0.4 | 1.3 | -0.5 | -1.2 | 1.3 | 0.0 | -1.0 | -1.7 | 0.0 | 0.0 | -6.7 | -2.7 | -3.8 | 0.0 | 0.0 | 0.1 | 0.2 | 0.6 | -0.3 | -0.5 | 0.6 | 0.1 | -0.4 | -0.6 | 0.0 | 0.0 | -1.7 | -1.0 | -1.2 |
| T2 | 0.6 | -0.2 | 0.7 | -0.1 | 1.8 | 0.5 | 0.8 | 3.2 | 0.0 | 2.0 | 5.5 | 0.0 | 16.1 | -13.3 | -1.4 | -7.5 | 0.3 | -0.1 | 0.5 | -0.1 | 0.8 | 0.2 | 0.3 | 1.5 | 0.1 | 0.9 | 2.0 | 0.0 | 5.2 | -3.3 | -0.5 | -2.4 |
| T3 | -0.4 | 0.1 | -0.5 | 1.9 | 0.0 | -2.6 | -1.6 | -1.6 | 0.0 | -0.5 | -1.1 | 0.0 | 0.0 | -6.7 | -5.4 | -1.2 | -0.2 | 0.0 | -0.1 | 0.6 | 0.0 | -1.3 | -0.8 | -0.6 | 0.1 | -0.2 | -0.4 | 0.0 | 0.0 | -1.7 | -1.9 | -1.2 |
| T4 | 0.1 | -0.4 | -0.4 | 0.0 | 8.0 | -0.9 | 1.2 | 2.9 | 1.3 | -2.0 | 2.8 | 0.0 | -3.2 | 6.7 | -1.4 | -9.4 | 0.1 | -0.2 | -0.1 | 0.0 | 3.4 | -0.4 | 0.5 | 1.3 | 0.6 | -0.8 | 0.9 | 0.0 | -1.0 | 1.7 | -0.5 | -3.0 |
| T5 | 0.6 | -0.2 | 0.3 | 0.2 | 0.0 | 5.7 | 5.2 | 0.0 | 3.8 | 0.0 | 0.6 | 0.0 | -3.2 | 0.0 | -2.7 | -7.5 | 0.3 | -0.1 | 0.4 | 0.1 | 0.0 | 2.1 | 2.3 | 0.1 | 1.6 | 0.1 | 0.3 | 0.0 | -1.1 | 0.0 | -1.0 | -2.4 |
| T6 | 0.3 | -0.4 | 0.0 | -0.2 | 1.8 | 0.9 | 5.6 | 1.0 | 0.6 | -1.0 | 1.1 | 0.0 | 12.9 | -6.7 | 1.4 | -5.7 | 0.1 | -0.2 | 0.1 | -0.1 | 0.8 | 0.4 | 2.3 | 0.4 | 0.3 | -0.4 | 0.5 | 0.0 | 4.2 | -1.7 | 0.5 | -1.8 |
| T7 | -0.3 | -0.2 | -0.4 | -0.6 | 0.4 | -0.2 | -2.4 | 5.8 | 0.6 | -0.5 | -2.8 | 0.0 | -6.5 | 0.0 | -5.4 | -7.5 | 0.0 | -0.2 | -0.3 | 0.2 | -0.2 | -1.1 | -0.4 | 2.6 | 0.4 | 0.3 | -0.2 | 0.0 | -1.0 | 0.0 | -1.9 | -2.3 |
| T8 | -0.1 | -0.1 | -0.7 | 0.1 | 0.4 | -1.9 | -1.2 | 1.0 | 10.2 | 0.5 | 2.2 | 0.0 | 0.0 | 6.7 | -4.1 | -5.7 | -0.1 | 0.0 | -0.2 | 0.1 | 0.2 | -0.8 | -0.4 | 0.4 | 3.9 | 0.1 | 0.8 | 0.0 | 0.0 | 1.7 | -1.4 | -1.8 |
| T9 | 0.2 | -0.5 | 0.4 | 0.4 | 1.8 | 0.5 | -0.4 | 1.6 | -1.3 | 4.4 | 2.2 | 0.0 | -3.2 | -6.7 | -4.1 | -5.7 | 0.1 | -0.2 | 0.3 | 0.1 | 0.8 | 0.2 | -0.1 | 0.8 | -0.5 | 1.7 | 0.9 | 0.0 | -1.0 | -1.7 | -1.5 | -1.8 |
| T10 | -0.1 | -0.4 | -0.5 | 0.5 | -0.4 | -0.9 | -0.8 | 1.9 | -3.2 | -2.9 | 12.2 | 0.0 | -6.5 | 6.7 | 1.4 | -5.7 | 0.0 | -0.1 | 0.1 | 0.2 | -0.2 | -0.3 | -0.2 | 0.8 | -1.2 | -1.1 | 3.7 | 0.0 | -2.1 | 1.6 | 0.5 | -1.8 |
| T11 | -0.1 | -0.2 | 0.0 | 0.5 | 0.4 | -0.9 | -3.2 | 1.3 | 0.6 | -1.0 | 2.2 | 1.1 | -3.2 | -6.7 | 1.4 | -3.8 | 0.0 | -0.1 | 0.1 | 0.2 | 0.2 | -0.4 | -1.3 | 0.5 | 0.3 | -0.4 | 0.8 | 0.6 | -1.1 | -1.7 | 0.5 | -1.2 |
| T12 | 0.4 | -0.2 | 0.5 | 0.2 | 0.4 | 0.7 | -1.2 | 1.6 | 4.5 | -2.4 | 4.4 | 0.0 | 29.0 | 0.0 | -4.1 | -7.5 | 0.2 | -0.1 | 0.4 | 0.1 | 0.2 | 0.3 | -0.4 | 0.7 | 1.8 | -1.0 | 1.6 | 0.0 | 9.4 | 0.0 | -1.5 | -2.4 |
| T13 | 0.2 | -0.5 | 0.0 | 0.0 | 1.8 | 0.2 | 0.8 | 3.9 | 0.0 | -2.0 | 4.4 | 0.0 | -6.5 | 26.7 | -1.4 | -9.4 | 0.1 | -0.2 | 0.1 | 0.0 | 0.8 | 0.1 | 0.4 | 1.7 | 0.0 | -0.9 | 1.6 | 0.0 | -2.1 | 6.6 | -0.5 | -3.0 |
| T14 | -0.1 | -0.2 | -0.1 | 0.0 | -0.4 | 1.2 | -1.6 | 0.0 | -1.3 | -1.0 | -1.1 | 0.0 | 3.2 | 0.0 | 13.5 | -5.7 | -0.1 | -0.1 | 0.2 | 0.0 | -0.2 | 0.5 | -0.5 | 0.0 | -0.4 | -0.4 | -0.4 | 0.0 | 1.1 | 0.0 | 4.2 | -1.8 |
| T15 | 0.1 | 0.0 | 0.0 | -0.4 | 1.8 | 0.2 | -1.2 | -1.0 | 1.9 | -2.0 | 3.9 | 0.0 | -3.2 | 0.0 | -2.7 | 20.8 | 0.1 | 0.0 | 0.1 | -0.2 | 0.8 | 0.1 | -0.4 | -0.3 | 0.8 | -0.8 | 1.3 | 0.0 | -1.0 | 0.0 | -1.0 | 6.3 |
| M0 | 0.86 | 0.98 | 0.94 | 0.91 | 0.80 | 0.79 | 0.66 | 0.78 | 0.67 | 0.74 | 0.54 | 0.95 | 0.48 | 0.33 | 0.55 | 0.45 | 0.92 | 0.99 | 0.95 | 0.95 | 0.90 | 0.89 | 0.82 | 0.88 | 0.83 | 0.86 | 0.76 | 0.97 | 0.74 | 0.67 | 0.77 | 0.73 |

Table 18: Similar to Table 3 and Tables 15∼17, this table shows the impact of the HSW strategy and objective metric $\phi^{\mathrm{cba}}$ (class-based balanced accuracy) for classes $T1, T2, \ldots, T15$ in the ticketing dataset.

| | Optimization: HSW-CBA-Class[row]; Performance Metric: ΔOF1%, ΔCR% | | | | | | | | | | | | | | | | Optimization: HSW-CBA-Class[row]; Performance Metric: ΔOBA%, ΔCBA% | | | | | | | | | | | | | | | |
| | ΔOF1 | ΔCR % | | | | | | | | | | | | | | | ΔOBA | ΔCBA % | | | | | | | | | | | | | | |
| Class | % | T1 | T2 | T3 | T4 | T5 | T6 | T7 | T8 | T9 | T10 | T11 | T12 | T13 | T14 | T15 | % | T1 | T2 | T3 | T4 | T5 | T6 | T7 | T8 | T9 | T10 | T11 | T12 | T13 | T14 | T15 |
| T1 | -0.1 | -0.1 | 0.1 | 0.4 | 1.3 | -0.5 | -1.2 | 1.3 | 0.0 | -1.0 | -1.7 | 0.0 | 0.0 | -6.7 | -2.7 | -3.8 | 0.0 | 0.0 | 0.1 | 0.2 | 0.6 | -0.3 | -0.5 | 0.6 | 0.1 | -0.4 | -0.6 | 0.0 | 0.0 | -1.7 | -1.0 | -1.2 |
| T2 | 0.7 | 0.1 | 0.6 | 0.2 | 2.7 | 0.9 | 0.4 | 2.9 | 3.8 | -1.5 | 5.0 | 0.0 | 12.9 | -13.3 | 1.4 | -3.8 | 0.4 | 0.0 | 0.5 | 0.1 | 1.1 | 0.4 | 0.3 | 1.4 | 1.5 | -0.4 | 1.7 | 0.0 | 4.2 | -3.3 | 0.5 | -1.2 |
| T3 | 0.3 | -0.2 | -0.2 | 1.4 | -0.4 | 0.0 | 1.6 | 2.3 | 1.9 | -0.5 | 1.7 | 0.0 | 3.2 | 0.0 | 0.0 | -1.2 | 0.1 | -0.1 | 0.0 | 0.6 | -0.2 | 0.1 | 0.6 | 1.1 | 0.7 | -0.2 | 0.6 | 0.0 | 1.0 | 0.0 | 0.0 | -1.2 |
| T4 | 0.1 | -0.2 | -0.5 | 0.0 | 9.8 | -0.7 | -2.8 | 1.6 | 1.9 | -2.9 | 5.5 | 0.0 | 9.7 | -13.3 | -2.7 | -7.5 | 0.1 | -0.1 | 0.0 | 0.0 | 4.1 | -0.3 | -1.1 | 0.7 | 0.8 | -1.2 | 1.9 | 0.0 | 3.2 | -3.3 | -0.9 | -2.4 |
| T5 | 0.3 | -0.5 | 0.4 | 0.4 | 0.9 | 4.3 | 0.0 | 1.3 | 0.0 | -2.9 | 0.6 | 0.0 | 6.5 | -6.7 | 1.4 | -3.8 | 0.2 | -0.2 | 0.3 | 0.1 | 0.4 | 1.6 | 0.2 | 0.7 | 0.1 | -1.2 | 0.1 | 0.0 | 2.1 | -1.7 | 0.5 | -1.2 |
| T6 | -0.3 | -0.1 | -0.6 | -0.4 | -0.4 | -1.4 | 4.8 | 0.6 | -1.3 | -1.5 | -0.6 | 0.0 | 0.0 | -6.7 | -1.4 | -2.4 | -0.2 | 0.0 | -0.3 | -0.2 | -0.2 | -0.6 | 1.8 | 0.3 | -0.5 | -0.6 | -0.2 | 0.0 | 0.0 | -1.7 | -0.5 | -2.4 |
| T7 | 0.7 | -0.2 | 0.7 | 0.2 | 2.7 | 1.7 | 1.2 | 5.8 | 3.2 | -3.4 | 3.9 | 0.0 | 6.5 | -6.7 | -1.4 | -5.7 | 0.4 | -0.1 | 0.5 | 0.1 | 1.2 | 0.8 | 0.6 | 2.6 | 1.3 | -1.4 | 1.4 | 0.0 | 2.1 | -1.7 | -0.5 | -1.8 |
| T8 | -0.2 | -0.2 | -0.6 | -0.7 | 1.3 | -0.9 | -1.2 | -0.6 | 11.5 | 0.5 | 1.7 | 0.0 | -6.5 | -6.7 | -4.1 | -7.5 | -0.1 | -0.1 | -0.3 | -0.4 | 0.6 | -0.3 | -0.5 | -0.3 | 4.3 | 0.3 | 0.6 | 0.0 | -2.1 | -1.7 | -1.4 | -2.3 |
| T9 | 0.4 | -0.2 | 0.4 | 0.1 | 1.3 | 1.9 | -2.0 | 1.9 | -2.5 | 5.4 | 4.4 | 0.0 | 9.7 | 0.0 | -6.8 | -5.7 | 0.2 | -0.1 | 0.2 | 0.1 | 0.6 | 0.9 | -0.7 | 0.9 | -1.0 | 2.2 | 1.7 | 0.0 | 3.2 | 0.0 | -0.4 | -1.8 |
| T10 | 0.1 | -0.1 | -0.1 | 0.4 | -1.3 | 0.0 | 0.8 | 0.0 | -2.5 | 0.5 | 11.0 | 0.0 | -6.5 | 0.0 | -6.8 | -7.5 | 0.0 | 0.0 | 0.2 | 0.2 | -0.6 | -0.1 | 0.5 | 0.1 | -0.9 | 0.2 | 3.2 | 0.0 | -2.1 | 0.0 | -2.4 | -2.3 |
| T11 | 0.6 | -0.3 | 0.1 | 0.9 | 2.7 | 1.9 | 0.4 | 2.6 | 2.5 | -2.9 | 9.4 | 1.1 | 3.2 | -6.7 | 2.7 | -9.4 | 0.3 | -0.2 | 0.4 | 0.4 | 1.2 | 0.9 | 0.3 | 1.2 | 1.1 | -1.1 | 3.2 | 0.5 | 1.0 | -1.7 | 0.9 | -3.0 |
| T12 | 0.3 | -0.2 | 0.0 | 3.6 | 0.9 | -2.4 | 1.3 | -0.6 | 0.0 | 3.9 | 0.0 | 0.0 | 32.3 | -1.4 | -0.4 | -7.5 | 0.1 | -0.1 | 0.2 | 0.1 | 1.5 | 0.4 | -0.8 | 0.6 | 0.1 | 0.0 | 1.3 | 0.0 | 10.4 | -1.7 | -0.4 | -2.4 |
| T13 | 0.0 | -0.3 | -0.2 | 0.0 | 0.9 | 0.7 | -3.2 | 0.3 | 3.2 | 0.5 | 2.2 | 0.0 | -3.2 | 33.3 | 0.0 | -1.9 | 0.0 | -0.1 | -0.1 | 0.0 | 0.4 | 0.3 | -1.3 | 0.2 | 1.2 | 0.2 | 0.8 | 0.0 | -1.0 | 8.2 | 0.0 | -0.6 |
| T14 | -1.0 | -0.2 | -1.1 | -0.5 | 2.2 | -1.2 | -3.6 | 0.0 | -6.4 | -4.4 | -5.0 | 0.0 | -12.9 | -6.7 | 16.2 | -11.3 | -0.5 | -0.1 | -0.4 | -0.3 | 1.0 | -0.5 | -1.5 | 0.0 | -2.5 | -1.9 | -1.8 | 0.0 | -4.2 | -1.7 | 5.1 | -3.5 |
| T15 | 0.2 | -0.3 | -0.3 | 0.5 | 1.8 | 0.0 | -4.4 | 2.6 | 1.9 | -1.0 | 3.3 | 0.0 | 9.7 | -13.3 | 1.4 | 20.8 | 0.1 | -0.1 | 0.1 | 0.2 | 0.8 | -0.1 | -1.7 | 1.1 | 0.8 | -0.3 | 1.1 | 0.0 | 3.1 | -3.3 | 0.5 | 6.1 |
| M0 | 0.86 | 0.98 | 0.94 | 0.91 | 0.80 | 0.79 | 0.66 | 0.78 | 0.67 | 0.74 | 0.54 | 0.95 | 0.48 | 0.33 | 0.55 | 0.45 | 0.92 | 0.99 | 0.95 | 0.95 | 0.90 | 0.89 | 0.82 | 0.88 | 0.83 | 0.86 | 0.76 | 0.97 | 0.74 | 0.67 | 0.77 | 0.73 |

Table 19: Similar to Table 3 and Tables 15∼18, this table shows the impact of the GA strategy and objective metric $\phi^{\mathrm{cba}}$ (class-based balanced accuracy) for classes $T1, T2, \ldots, T15$ in the ticketing dataset.

| | Optimization: GA-CBA-Class[row]; Performance Metric: ΔOF1%, ΔCR% | | | | | | | | | | | | | | | | Optimization: GA-CBA-Class[row]; Performance Metric: ΔOBA%, ΔCBA % | | | | | | | | | | | | | | | |
|---|---|---|---|---|---|---|---|---|---|---|---|---|---|---|---|---|---|---|---|---|---|---|---|---|---|---|---|---|---|---|---|---|
| | ΔOF1 | ΔCR% | | | | | | | | | | | | | | | ΔOBA | ΔCBA% | | | | | | | | | | | | | | |
| Class | % | T1 | T2 | T3 | T4 | T5 | T6 | T7 | T8 | T9 | T10 | T11 | T12 | T13 | T14 | T15 | % | T1 | T2 | T3 | T4 | T5 | T6 | T7 | T8 | T9 | T10 | T11 | T12 | T13 | T14 | T15 |
| T1 | 0.3 | 0.2 | -0.5 | 0.9 | 0.0 | -0.5 | 0.0 | 2.3 | 1.9 | 2.9 | 3.9 | 0.0 | 12.9 | -20.0 | -4.1 | -1.9 | 0.1 | -0.2 | 0.0 | 0.4 | 0.0 | -0.1 | 0.1 | 1.0 | 0.8 | 1.3 | 1.3 | 0.0 | 4.2 | -5.0 | -1.4 | -0.6 |
| T2 | 0.0 | 0.4 | 0.8 | -0.5 | -0.9 | 0.9 | -2.8 | -0.6 | -5.7 | 1.0 | 5.0 | 0.0 | 3.2 | -13.3 | -2.7 | -17.0 | 0.0 | 0.2 | -0.1 | -0.2 | -0.4 | 0.4 | -0.9 | -0.2 | -2.3 | 0.6 | 1.8 | 0.0 | 1.0 | -3.3 | -1.0 | -5.3 |
| T3 | 0.3 | -0.2 | 0.0 | 2.0 | 1.3 | 0.5 | 0.8 | 1.0 | -0.6 | -1.5 | 3.9 | 0.0 | -6.5 | -6.7 | -5.4 | -5.7 | 0.1 | -0.1 | -0.1 | 0.7 | 0.6 | 0.2 | 0.3 | 0.5 | -0.2 | -0.5 | 1.6 | 0.0 | -2.1 | -1.7 | -1.9 | -1.7 |
| T4 | 0.4 | 0.0 | 0.0 | 1.1 | 8.4 | -0.7 | 0.8 | 1.0 | -2.5 | 0.5 | 3.3 | 0.0 | -3.2 | 0.0 | -2.7 | -3.8 | 0.2 | 0.1 | 0.2 | 0.5 | 3.6 | -0.3 | 0.2 | 0.5 | -0.9 | 0.2 | 1.3 | 0.0 | -1.1 | 0.0 | -0.9 | -1.2 |
| T5 | 0.2 | -0.3 | -0.3 | 0.9 | 0.4 | 2.8 | 1.2 | 1.0 | -2.5 | 2.4 | 3.3 | 0.0 | 3.2 | -20.0 | -1.4 | -9.4 | -0.1 | -0.1 | 0.3 | 0.6 | 0.2 | 0.7 | 0.6 | 0.4 | -0.9 | 1.1 | 1.2 | 0.0 | 1.0 | -5.0 | -0.5 | -3.0 |
| T6 | -0.2 | -0.1 | -0.4 | 0.5 | -0.4 | 0.5 | 4.0 | 0.3 | -3.8 | -2.4 | -3.3 | -1.1 | 19.4 | -13.3 | -2.7 | -7.5 | -0.1 | 0.0 | 0.0 | 0.2 | -0.2 | 0.2 | 0.9 | 0.1 | -1.4 | -0.9 | -1.1 | -0.5 | 6.3 | -3.3 | -0.9 | -2.4 |
| T7 | 0.0 | 0.0 | 0.0 | 0.4 | 0.0 | 0.5 | 2.4 | -0.3 | 1.9 | -2.0 | 1.7 | 0.0 | -3.2 | 0.0 | 0.0 | -6.8 | -0.2 | 0.0 | 0.1 | 0.2 | 0.1 | 0.1 | 0.9 | -0.3 | 0.8 | -0.8 | 0.6 | 0.0 | -1.1 | 0.0 | -2.4 | -1.8 |
| T8 | -0.4 | -0.2 | -0.5 | 0.1 | -0.9 | 1.2 | -3.2 | 0.6 | 5.1 | -2.0 | -4.4 | 0.0 | 3.2 | -6.7 | 0.0 | -9.4 | -0.2 | -0.1 | -0.2 | 0.1 | -0.4 | 0.5 | -1.3 | 0.3 | 1.7 | -0.8 | -1.5 | 0.0 | 1.0 | -1.7 | 0.0 | -3.0 |
| T9 | -0.1 | -0.2 | 0.1 | 0.2 | -0.4 | -0.2 | 0.0 | 1.3 | -3.2 | 2.9 | -2.2 | 0.0 | 6.5 | 0.0 | -1.4 | -9.4 | 0.0 | -0.1 | 0.2 | 0.1 | -0.2 | -0.1 | 0.0 | 0.6 | -1.2 | 0.9 | -0.7 | 0.0 | 2.1 | 0.0 | -0.4 | -3.0 |
| T10 | 0.6 | -0.2 | -0.3 | 0.6 | -0.4 | 4.0 | 2.0 | 1.6 | -3.8 | 3.9 | 9.4 | 0.0 | 3.2 | 0.0 | 2.7 | -7.5 | 0.3 | -0.1 | 0.2 | 0.3 | -0.2 | 1.8 | 0.9 | 0.8 | -1.4 | 1.8 | 2.7 | 0.0 | 1.1 | 0.0 | 1.0 | -2.3 |
| T11 | 0.2 | -0.1 | 0.1 | 0.5 | 0.0 | -1.9 | 3.2 | 2.6 | 1.9 | -2.4 | 3.3 | 1.1 | -3.2 | 0.0 | 0.0 | -11.3 | 0.1 | 0.0 | 0.1 | 0.2 | 0.0 | -0.9 | 1.2 | 1.1 | 0.8 | -0.9 | 1.1 | 0.5 | -1.0 | 0.0 | 0.0 | -3.6 |
| T12 | 0.1 | -0.4 | -0.2 | 0.2 | 1.8 | 1.7 | -0.4 | 1.0 | 0.0 | -2.0 | 1.7 | 0.0 | 32.3 | 6.7 | -1.4 | -5.7 | 0.1 | -0.2 | 0.1 | 0.1 | 0.8 | 0.7 | -0.2 | 0.4 | 0.1 | -0.9 | 0.6 | 0.0 | 10.4 | 1.7 | -0.5 | -1.8 |
| T13 | 0.1 | -0.6 | 0.1 | 0.2 | 1.3 | -1.4 | -2.0 | 5.2 | 1.9 | -2.4 | 1.1 | 0.0 | 3.2 | 33.3 | -1.4 | -3.8 | 0.0 | -0.3 | 0.3 | 0.1 | 0.5 | -0.6 | -0.8 | 2.2 | 0.7 | -1.0 | 0.4 | 0.0 | 1.0 | 8.2 | -0.4 | -1.2 |
| T14 | 0.0 | -0.2 | 0.1 | 0.1 | 0.4 | -0.9 | -0.8 | 0.6 | -2.5 | 0.0 | -1.7 | 0.0 | 0.0 | -13.3 | 16.2 | -5.7 | -0.1 | -0.1 | 0.2 | 0.1 | 0.2 | -0.3 | -0.2 | 0.3 | -1.0 | 0.1 | -0.5 | 0.0 | 0.0 | -3.3 | 5.1 | -1.8 |
| T15 | 0.1 | -0.4 | 0.0 | 0.4 | 1.3 | 0.2 | -1.6 | 1.0 | -3.8 | -1.5 | 2.8 | 0.0 | 3.2 | -6.7 | 0.0 | 24.5 | 0.1 | -0.1 | 0.2 | 0.1 | 0.6 | 0.1 | -0.5 | 0.4 | -1.5 | -0.6 | 0.9 | 0.0 | 1.1 | -1.7 | 0.0 | 7.3 |
| M0 | 0.86 | 0.98 | 0.94 | 0.91 | 0.80 | 0.79 | 0.66 | 0.78 | 0.67 | 0.74 | 0.54 | 0.95 | 0.48 | 0.33 | 0.55 | 0.45 | 0.92 | 0.99 | 0.95 | 0.95 | 0.90 | 0.89 | 0.82 | 0.88 | 0.83 | 0.86 | 0.76 | 0.97 | 0.74 | 0.67 | 0.77 | 0.73 |

Table 20: Similar to Table 3 and Tables 15∼19, this table shows the impact of the SW strategy and objective metric $\phi^{\mathrm{oba}}$ (overall balanced accuracy) for classes $T1, T2, \ldots, T15$ in the ticketing dataset.

| | Optimization: SW-OBA-Class[row]; Performance Metric: ΔOF1%, ΔCR% | | | | | | | | | | | | | | | | Optimization: SW-OBA-Class[row]; Performance Metric: ΔOBA%, ΔCBA % | | | | | | | | | | | | | | | |
|---|---|---|---|---|---|---|---|---|---|---|---|---|---|---|---|---|---|---|---|---|---|---|---|---|---|---|---|---|---|---|---|---|
| | ΔOF1 | ΔCR% | | | | | | | | | | | | | | | ΔOBA | ΔCBA% | | | | | | | | | | | | | | |
| Class | % | T1 | T2 | T3 | T4 | T5 | T6 | T7 | T8 | T9 | T10 | T11 | T12 | T13 | T14 | T15 | % | T1 | T2 | T3 | T4 | T5 | T6 | T7 | T8 | T9 | T10 | T11 | T12 | T13 | T14 | T15 |
| T1 | 0.6 | -0.1 | 0.6 | 0.2 | 2.7 | 0.2 | 0.0 | 1.3 | 4.5 | -1.0 | 6.1 | 0.0 | 16.1 | 0.0 | 0.0 | -7.5 | 0.3 | -0.1 | 0.5 | 0.1 | 1.2 | 0.1 | 0.1 | 0.6 | 1.9 | -0.4 | 2.1 | 0.0 | 5.3 | 0.0 | 0.0 | -2.4 |
| T2 | 0.7 | 0.1 | 0.0 | 0.0 | 4.9 | 2.4 | 2.0 | 2.3 | -0.6 | 0.0 | 3.3 | 0.0 | 9.7 | 0.0 | 2.7 | -1.9 | 0.3 | 0.1 | 0.1 | 0.0 | 2.2 | 1.1 | 0.8 | 1.1 | -0.2 | 0.1 | 1.1 | 0.0 | 3.2 | 0.0 | 1.0 | -0.6 |
| T3 | 0.7 | -0.1 | 0.1 | 0.9 | 3.6 | 2.1 | -0.4 | 3.6 | 1.9 | 0.0 | 5.5 | 0.0 | 3.2 | -6.7 | -2.7 | -3.8 | 0.3 | 0.0 | 0.3 | 0.4 | 1.6 | 0.9 | 0.0 | 1.6 | 0.8 | 0.0 | 1.9 | 0.0 | 1.1 | -1.7 | -1.0 | -1.2 |
| T4 | 0.6 | -0.1 | 0.3 | 0.1 | 7.6 | -0.5 | 1.6 | 3.2 | 2.5 | -1.5 | 3.9 | 0.0 | 6.5 | -6.7 | 0.0 | -3.8 | 0.3 | 0.0 | 0.3 | 0.0 | 3.2 | -0.1 | 0.7 | 1.5 | 1.1 | -0.5 | 1.3 | 0.0 | 2.1 | -1.7 | 0.0 | -1.2 |
| T5 | 0.7 | -0.4 | 0.6 | 0.1 | 3.6 | 3.3 | -1.2 | 3.2 | 1.3 | 0.0 | 6.6 | 0.0 | 3.2 | -13.3 | -1.4 | -3.8 | 0.3 | -0.1 | 0.4 | 0.1 | 1.6 | 1.4 | -0.4 | 1.5 | 0.6 | 0.0 | 2.4 | 0.0 | 1.1 | -3.3 | -0.5 | -1.2 |
| T6 | 0.8 | 0.0 | 0.7 | 0.6 | 1.8 | 1.9 | 2.4 | 3.2 | 1.3 | -2.0 | 2.8 | 0.0 | 9.7 | -13.3 | 0.0 | -1.9 | 0.4 | 0.0 | 0.5 | 0.3 | 0.8 | 0.9 | 1.0 | 1.5 | 0.5 | -0.7 | 1.0 | 0.0 | 3.1 | -3.3 | 0.0 | -0.6 |
| T7 | 0.8 | -0.1 | 0.6 | 0.0 | 1.3 | 0.9 | 0.0 | 5.8 | 3.8 | 0.5 | 3.3 | 0.0 | 16.1 | 0.0 | -1.4 | -1.9 | 0.4 | 0.0 | 0.5 | 0.0 | 0.6 | 0.5 | 0.1 | 2.5 | 1.6 | 0.3 | 1.2 | 0.0 | 5.3 | 0.0 | -0.5 | -0.6 |
| T8 | 0.7 | -0.2 | 0.7 | 0.1 | 0.4 | 0.9 | -2.0 | 3.6 | 2.5 | 1.5 | 7.7 | 0.0 | 0.0 | -13.3 | 2.7 | 0.0 | 0.3 | -0.1 | 0.5 | 0.1 | 0.2 | 0.4 | -0.7 | 1.6 | 1.0 | 0.6 | 2.8 | 0.0 | 0.0 | -3.3 | 1.0 | 0.0 |
| T9 | 0.8 | -0.1 | 0.7 | 0.7 | 2.2 | 2.1 | 3.6 | 1.0 | 3.8 | -2.0 | 3.3 | 0.0 | 3.2 | 0.0 | 4.1 | -7.5 | 0.4 | 0.0 | 0.6 | 0.4 | 1.0 | 0.9 | 1.6 | 0.5 | 1.6 | -1.0 | 1.1 | 0.0 | 1.1 | 0.0 | 1.5 | -2.4 |
| T10 | 0.7 | 0.0 | 0.4 | -0.1 | 3.1 | 1.7 | 3.2 | 1.3 | 0.6 | 0.5 | 5.5 | 0.0 | 16.1 | 0.0 | 0.0 | -1.9 | 0.4 | 0.0 | 0.5 | -0.1 | 1.4 | 0.8 | 1.5 | 0.6 | 0.3 | 0.3 | 1.7 | 0.0 | 5.3 | 0.0 | 0.0 | -0.6 |
| T11 | 0.7 | -0.1 | 0.7 | 0.6 | 3.1 | 0.0 | 2.4 | 2.3 | 1.9 | -1.0 | 3.3 | 0.0 | 12.9 | 0.0 | -2.7 | -5.7 | 0.3 | 0.0 | 0.5 | 0.3 | 1.4 | 0.0 | 0.9 | 1.1 | 0.8 | -0.4 | 1.2 | 0.0 | 4.2 | 0.0 | -1.0 | -1.8 |
| T12 | 0.8 | -0.1 | 0.6 | 0.5 | 4.0 | 1.9 | -0.4 | 3.6 | 1.3 | 0.0 | 1.1 | 0.0 | 25.8 | -6.7 | 1.4 | -0.6 | 0.4 | 0.0 | 0.5 | 0.2 | 1.8 | 0.8 | 0.0 | 1.6 | 0.6 | 0.1 | 0.4 | 0.0 | 8.4 | -1.7 | 0.4 | -0.6 |
| T13 | 0.7 | -0.4 | 0.4 | 0.2 | 3.1 | 1.4 | -1.2 | 3.6 | 1.3 | 1.5 | 3.3 | 0.0 | 9.7 | 20.0 | 1.4 | 1.9 | 0.3 | -0.2 | 0.4 | 0.1 | 1.3 | 0.7 | -0.3 | 1.6 | 0.6 | 0.6 | 1.2 | 0.0 | 3.2 | 4.9 | 0.4 | 0.6 |
| T14 | 0.5 | -0.2 | 0.2 | 0.7 | 3.6 | 2.1 | 0.8 | 1.0 | 1.9 | -2.0 | 3.9 | 0.0 | 3.2 | -6.7 | 5.4 | -7.5 | 0.3 | -0.1 | 0.2 | 0.3 | 1.6 | 1.0 | 0.4 | 0.5 | 0.8 | -0.8 | 1.4 | 0.0 | 1.1 | -1.7 | 1.8 | -2.4 |
| T15 | 0.7 | -0.1 | 0.7 | 0.7 | 1.8 | 0.0 | 0.8 | 1.6 | 0.6 | -0.5 | 6.1 | 0.0 | 19.4 | -6.7 | 0.0 | 3.8 | 0.4 | 0.0 | 0.5 | 0.4 | 0.8 | 0.0 | 0.4 | 0.8 | 0.3 | -0.2 | 2.2 | 0.0 | 6.3 | -1.7 | 0.0 | 1.1 |
| M0 | 0.86 | 0.98 | 0.94 | 0.91 | 0.80 | 0.79 | 0.66 | 0.78 | 0.67 | 0.74 | 0.54 | 0.95 | 0.48 | 0.33 | 0.55 | 0.45 | 0.92 | 0.99 | 0.95 | 0.95 | 0.90 | 0.89 | 0.82 | 0.88 | 0.83 | 0.86 | 0.76 | 0.97 | 0.74 | 0.67 | 0.77 | 0.73 |

Table 21: Similar to Table 3 and Tables 15∼20, this table shows the impact of the HSW strategy and objective metric $\phi^{\mathrm{oba}}$ (overall balanced accuracy) for classes $T1, T2, \ldots, T15$ in the ticketing dataset.

| | Optimization: HSW-OBA-Class[row]; Performance Metric: ΔOF1%, ΔCR% | | | | | | | | | | | | | | | | Optimization: HSW-OBA-Class[row]; Performance Metric: ΔOBA%, ΔCBA % | | | | | | | | | | | | | | | |
|---|---|---|---|---|---|---|---|---|---|---|---|---|---|---|---|---|---|---|---|---|---|---|---|---|---|---|---|---|---|---|---|---|
| | ΔOF1 | ΔCR% | | | | | | | | | | | | | | | ΔOBA | ΔCBA% | | | | | | | | | | | | | | |
| Class | % | T1 | T2 | T3 | T4 | T5 | T6 | T7 | T8 | T9 | T10 | T11 | T12 | T13 | T14 | T15 | % | T1 | T2 | T3 | T4 | T5 | T6 | T7 | T8 | T9 | T10 | T11 | T12 | T13 | T14 | T15 |
| T1 | 0.6 | -0.3 | 0.6 | 0.4 | 2.2 | 1.7 | 0.4 | 1.9 | 3.8 | -3.4 | 3.9 | 0.0 | 19.4 | -6.7 | 2.7 | -1.9 | 0.3 | -0.1 | 0.5 | 0.2 | 1.0 | 0.7 | 0.3 | 0.9 | 1.5 | -1.4 | 1.4 | 0.0 | 6.3 | -1.7 | 0.9 | -0.6 |
| T2 | 0.7 | 0.1 | 0.6 | 0.2 | 2.7 | 0.9 | 0.4 | 2.9 | 3.8 | -1.5 | 5.0 | 0.0 | 12.9 | -13.3 | 1.4 | -3.8 | 0.4 | 0.0 | 0.5 | 0.1 | 1.1 | 0.4 | 0.3 | 1.4 | 1.5 | -0.4 | 1.7 | 0.0 | 4.2 | -3.3 | 0.5 | -1.2 |
| T3 | 0.6 | -0.1 | 0.3 | 0.4 | 3.1 | 2.6 | 2.0 | 1.6 | -1.9 | 1.0 | 2.8 | 0.0 | 6.5 | 0.0 | 0.0 | -1.8 | 0.3 | 0.0 | 0.3 | 0.1 | 1.3 | 1.1 | 0.8 | 0.8 | -0.7 | 0.6 | 1.1 | 0.0 | 2.1 | 0.0 | 0.0 | -1.8 |
| T4 | 0.7 | -0.4 | 0.4 | -0.1 | 4.4 | 1.9 | -0.4 | 4.9 | 0.6 | 2.0 | 6.1 | 0.0 | 12.9 | -6.7 | -1.4 | -5.7 | 0.4 | -0.1 | 0.5 | 0.0 | 1.9 | 0.9 | 0.0 | 2.2 | 0.2 | 0.9 | 2.2 | 0.0 | 4.2 | -1.7 | -0.5 | -1.8 |
| T5 | 0.7 | -0.1 | 0.8 | 0.2 | 0.4 | 1.9 | 0.8 | 3.2 | 0.6 | 0.5 | 6.1 | 0.0 | 3.2 | -6.7 | -2.7 | 1.9 | 0.4 | 0.0 | 0.6 | 0.1 | 0.2 | 0.8 | 0.4 | 1.5 | 0.3 | 0.2 | 2.3 | 0.0 | 1.1 | -1.7 | -0.9 | 0.6 |
| T6 | 0.7 | -0.4 | 0.6 | 0.2 | 0.9 | -0.5 | 5.6 | 4.2 | 1.9 | 0.5 | 3.9 | 0.0 | 9.7 | 6.7 | -5.4 | -3.8 | 0.3 | -0.1 | 0.4 | 0.2 | 0.4 | -0.2 | 2.3 | 1.8 | 0.8 | 0.2 | 1.4 | 0.0 | 3.2 | 1.7 | -1.9 | -1.2 |
| T7 | 0.7 | -0.2 | 0.7 | 0.2 | 2.7 | 1.7 | 1.2 | 5.8 | 3.2 | -3.4 | 3.9 | 0.0 | 6.5 | -6.7 | -1.4 | -5.7 | 0.4 | -0.1 | 0.5 | 0.1 | 1.2 | 0.8 | 0.6 | 2.6 | 1.3 | -1.4 | 1.4 | 0.0 | 2.1 | -1.7 | -0.5 | -1.8 |
| T8 | 0.6 | 0.1 | 0.3 | 0.7 | 2.2 | 1.7 | -0.4 | 1.9 | 5.7 | 1.5 | 2.8 | 0.0 | 3.2 | -6.7 | 0.0 | -7.5 | 0.3 | 0.0 | 0.5 | 0.4 | 1.0 | 0.8 | -0.1 | 0.9 | 2.3 | 0.7 | 0.8 | 0.0 | 1.1 | -1.7 | 0.0 | -2.3 |
| T9 | 0.6 | -0.3 | 0.8 | 0.6 | 2.2 | 0.7 | -1.2 | 3.9 | 4.5 | -1.0 | 3.3 | 0.0 | 3.2 | -6.7 | 0.0 | -9.4 | 0.3 | -0.1 | 0.7 | 0.3 | 1.0 | 0.3 | -0.4 | 1.8 | 1.8 | -0.4 | 1.2 | 0.0 | 1.1 | -1.7 | -0.1 | -2.9 |
| T10 | 1.0 | -0.1 | 0.5 | 1.1 | 4.0 | 2.1 | 0.4 | 3.9 | 3.8 | -1.5 | 3.3 | 0.0 | 9.7 | 0.0 | 1.4 | 1.9 | 0.5 | -0.1 | 0.5 | 0.5 | 1.8 | 1.0 | 0.3 | 1.8 | 1.6 | -0.6 | 1.2 | 0.0 | 3.1 | 0.0 | 0.5 | 0.6 |
| T11 | 0.7 | -0.1 | 0.5 | 1.1 | 2.2 | -0.2 | -1.2 | 2.3 | 4.5 | -1.0 | 8.3 | 0.0 | 3.2 | 0.0 | 0.0 | -5.7 | 0.3 | -0.1 | 0.5 | 0.5 | 1.0 | -0.1 | -0.4 | 1.1 | 1.8 | -0.4 | 2.9 | 0.0 | 1.1 | 0.0 | 0.0 | -1.8 |
| T12 | 0.7 | -0.2 | 0.6 | 0.1 | 3.6 | 0.5 | -1.2 | 4.5 | 0.5 | 7.7 | 0.0 | 0.0 | 19.4 | 0.0 | -1.4 | -7.5 | 0.3 | -0.1 | 0.5 | 0.1 | 1.6 | 0.2 | -0.3 | 1.9 | 0.2 | 2.8 | 0.0 | 0.0 | 6.3 | 0.0 | -0.4 | -2.4 |
| T13 | 0.8 | -0.2 | 0.4 | 0.7 | 2.7 | 1.7 | 1.6 | 3.6 | 3.2 | -2.0 | 4.4 | 0.0 | 9.7 | 20.0 | -1.4 | 0.0 | 0.4 | 0.0 | 0.4 | 0.4 | 1.2 | 0.7 | 0.8 | 1.6 | 1.3 | -0.8 | 1.6 | 0.0 | 3.2 | 4.8 | -0.5 | 0.0 |
| T14 | 0.6 | -0.4 | 0.2 | 0.5 | 3.1 | 1.7 | -2.4 | 3.9 | 3.2 | 0.5 | 5.0 | 0.0 | 16.1 | 0.0 | 0.0 | 0.0 | 0.3 | -0.2 | 0.3 | 0.2 | 1.4 | 0.6 | -0.9 | 1.7 | 1.4 | 0.2 | 1.8 | 0.0 | 5.3 | 0.0 | 0.0 | 0.0 |
| T15 | 0.7 | -0.1 | 0.9 | 0.4 | 1.3 | -0.9 | -2.0 | 2.6 | 1.9 | 2.0 | 4.4 | 0.0 | 16.1 | -6.7 | 0.0 | 11.3 | 0.4 | 0.0 | 0.7 | 0.2 | 0.6 | -0.4 | -0.6 | 1.2 | 0.8 | 0.9 | 1.6 | 0.0 | 5.2 | -1.7 | 0.0 | 3.3 |
| M0 | 0.86 | 0.98 | 0.94 | 0.91 | 0.80 | 0.79 | 0.66 | 0.78 | 0.67 | 0.74 | 0.54 | 0.95 | 0.48 | 0.33 | 0.55 | 0.45 | 0.92 | 0.99 | 0.95 | 0.95 | 0.90 | 0.89 | 0.82 | 0.88 | 0.83 | 0.86 | 0.76 | 0.97 | 0.74 | 0.67 | 0.77 | 0.73 |

Table 22: Similar to Table 3 and Tables 15∼21, this table shows the impact of the GA strategy and objective metric $\phi^{\mathrm{oba}}$ (overall balanced accuracy) for classes $T1, T2, \ldots, T15$ in the ticketing dataset.

| | Optimization: GA-OBA-Class[row]; Performance Metric: ΔOF1%, ΔCR % | | | | | | | | | | | | | | | | Optimization: GA-OBA-Class[row]; Performance Metric: ΔOBA%, ΔCBA % | | | | | | | | | | | | | | | |
| Class | ΔOF1 % | T1 | T2 | T3 | T4 | T5 | T6 | T7 | T8 | T9 | T10 | T11 | T12 | T13 | T14 | T15 | ΔOBA % | T1 | T2 | T3 | T4 | T5 | T6 | T7 | T8 | T9 | T10 | T11 | T12 | T13 | T14 | T15 |
|---|---|---|---|---|---|---|---|---|---|---|---|---|---|---|---|---|---|---|---|---|---|---|---|---|---|---|---|---|---|---|---|---|
| T1 | 0.3 | 0.2 | -0.5 | 0.9 | 0.0 | -0.5 | 0.0 | 2.3 | 1.9 | 2.9 | 3.9 | 0.0 | 12.9 | -20.0 | -4.1 | -1.9 | 0.1 | -0.2 | 0.0 | 0.4 | 0.0 | -0.1 | 0.1 | 1.0 | 0.8 | 1.3 | 1.3 | 0.0 | 4.2 | -5.0 | -1.4 | -0.6 |
| T2 | 0.2 | 0.1 | 0.7 | -0.4 | 0.9 | 1.2 | -3.2 | 2.3 | -4.5 | -2.4 | 4.4 | 0.0 | 9.7 | -13.3 | -2.7 | -1.9 | 0.1 | 0.1 | -0.2 | -0.1 | 0.4 | 0.5 | -1.1 | 1.0 | -1.8 | -0.8 | 1.6 | 0.0 | 3.1 | -3.3 | -1.0 | -0.6 |
| T3 | 0.6 | 0.1 | 0.4 | 1.2 | 3.6 | 1.2 | 0.4 | 1.9 | -2.5 | -0.5 | 7.2 | 0.0 | 3.2 | -6.7 | -5.4 | -9.4 | 0.3 | 0.1 | 0.3 | 0.4 | 1.5 | 0.5 | 0.2 | 0.9 | -0.9 | -0.1 | 2.7 | 0.0 | 1.0 | -1.6 | -1.9 | -2.9 |
| T4 | 0.8 | 0.0 | 0.6 | 0.2 | 6.2 | 1.4 | 2.4 | 0.3 | -0.6 | 2.0 | 5.0 | 0.0 | 6.5 | -6.7 | -1.4 | -3.8 | 0.4 | 0.0 | 0.5 | 0.2 | 2.6 | 0.7 | 1.0 | 0.2 | -0.1 | 0.8 | 1.9 | 0.0 | 2.1 | -1.7 | -0.5 | -1.2 |
| T5 | 0.2 | -0.3 | -0.3 | 0.9 | 0.4 | 2.8 | 1.2 | 1.0 | -2.5 | 2.4 | 3.3 | 0.0 | 3.2 | -20.0 | -1.4 | -9.4 | 0.1 | -0.1 | 0.3 | 0.6 | 0.2 | 0.7 | 0.6 | 0.4 | -0.9 | 1.1 | 1.2 | 0.0 | 1.0 | -5.0 | -0.5 | -3.0 |
| T6 | 0.3 | 0.3 | 0.0 | 0.6 | 2.2 | 2.6 | 2.0 | 1.6 | -1.3 | -2.0 | 0.0 | 0.0 | 12.9 | -13.3 | -2.7 | -15.1 | 0.2 | 0.1 | 0.2 | 0.3 | 1.0 | 1.1 | 0.2 | 0.9 | -0.4 | -0.8 | 0.2 | 0.0 | 4.2 | -3.3 | -0.9 | -4.7 |
| T7 | 0.3 | 0.0 | 0.3 | 0.1 | 2.7 | 0.9 | 2.0 | -4.5 | 0.6 | 2.4 | 1.7 | 0.0 | 9.7 | -13.3 | -4.1 | 1.9 | 0.1 | 0.0 | 0.5 | 0.1 | 1.2 | 0.4 | 0.8 | -2.2 | 0.3 | 1.1 | 0.5 | 0.0 | 3.1 | -3.3 | -1.5 | 0.5 |
| T8 | 0.4 | 0.2 | -0.1 | 0.6 | 1.8 | 1.7 | -2.4 | 2.3 | 1.3 | 2.0 | 4.4 | 0.0 | 3.2 | -6.7 | 0.0 | -7.5 | 0.2 | 0.1 | 0.2 | 0.3 | 0.8 | 0.8 | -0.8 | 1.1 | 0.3 | 0.9 | 1.6 | 0.0 | 1.0 | -1.7 | 0.0 | -2.4 |
| T9 | 0.5 | -0.1 | 0.4 | 0.8 | -0.4 | 1.4 | 0.0 | 1.6 | 0.0 | -1.5 | 2.2 | 0.0 | 9.7 | 0.0 | 4.1 | 1.9 | 0.2 | 0.0 | 0.4 | 0.5 | -0.2 | 0.6 | 0.1 | 0.9 | 0.0 | -0.8 | 0.8 | 0.0 | 3.1 | 0.0 | 1.4 | 0.5 |
| T10 | 0.6 | -0.2 | -0.3 | 0.6 | -0.4 | 4.0 | 2.0 | 1.6 | -3.8 | 3.9 | 9.4 | 0.0 | 3.2 | 0.0 | 2.7 | -7.5 | 0.3 | -0.1 | 0.2 | 0.3 | -0.2 | 1.8 | 0.9 | 0.8 | -1.4 | 1.8 | 2.7 | 0.0 | 1.1 | 0.0 | 1.0 | -2.3 |
| T11 | 0.9 | -0.1 | 0.7 | 0.1 | 1.8 | 2.8 | 3.6 | 2.3 | 0.0 | 2.4 | 6.6 | 0.0 | 0.0 | 0.0 | -1.4 | -9.4 | 0.4 | 0.0 | 0.4 | 0.1 | 0.8 | 1.2 | 1.5 | 1.2 | 0.1 | 1.1 | 2.4 | 0.0 | 0.0 | 0.0 | -0.5 | -2.9 |
| T12 | 0.7 | 0.0 | 0.1 | -0.4 | 4.4 | 0.0 | 0.0 | 4.9 | 1.9 | 1.0 | 5.5 | 0.0 | 22.6 | 0.0 | 2.7 | -1.9 | 0.3 | 0.0 | 0.1 | -0.1 | 1.9 | 0.1 | 0.1 | 2.2 | 0.8 | 0.4 | 2.0 | 0.0 | 7.3 | 0.0 | 1.0 | -0.6 |
| T13 | 0.7 | -0.1 | 0.6 | 0.6 | 3.1 | 1.2 | -0.8 | 2.3 | 1.3 | 2.0 | 1.7 | 0.0 | 9.7 | 20.0 | 4.1 | -7.5 | 0.3 | 0.0 | 0.5 | 0.3 | 1.4 | 0.5 | -0.2 | 1.1 | 0.5 | 0.9 | 0.6 | 0.0 | 3.2 | 4.9 | 1.5 | -2.4 |
| T14 | 0.4 | 0.0 | 0.8 | 0.8 | 1.3 | -1.4 | 0.4 | 2.3 | -1.9 | -2.4 | 1.7 | 0.0 | 6.5 | -6.7 | 8.1 | -3.8 | 0.2 | 0.0 | 0.5 | 0.4 | 0.6 | -0.5 | 0.3 | 1.1 | -0.7 | -1.0 | 0.6 | 0.0 | 2.1 | -1.7 | 2.4 | -1.2 |
| T15 | 0.7 | 0.1 | 0.0 | 0.5 | 2.7 | 1.9 | 0.8 | 0.6 | 5.1 | -1.5 | 5.0 | 0.0 | 6.5 | 0.0 | 0.0 | 18.9 | 0.4 | 0.1 | 0.3 | 0.2 | 1.2 | 0.8 | 0.5 | 0.4 | 2.1 | -0.5 | 1.8 | 0.0 | 2.1 | 0.0 | 0.0 | 5.6 |
| M0 | 0.86 | 0.98 | 0.94 | 0.91 | 0.80 | 0.79 | 0.66 | 0.78 | 0.67 | 0.74 | 0.54 | 0.95 | 0.48 | 0.33 | 0.55 | 0.45 | 0.92 | 0.99 | 0.95 | 0.95 | 0.90 | 0.89 | 0.82 | 0.88 | 0.83 | 0.86 | 0.76 | 0.97 | 0.74 | 0.67 | 0.77 | 0.73 |

Table 23: Similar to Table 3 and Tables 15∼22, this table shows the impact of the SW strategy and objective metric $\phi^{\mathrm{of1}}$ (overall F1-score) for classes $T1, T2, \ldots, T15$ in the ticketing dataset.

| | Optimization: SW-OF1-Class[row]; Performance Metric: ΔOF1%, ΔCR % | | | | | | | | | | | | | | | | Optimization: SW-OF1-Class[row]; Performance Metric: ΔOBA%, ΔCBA % | | | | | | | | | | | | | | | |
| Class | ΔOF1 % | T1 | T2 | T3 | T4 | T5 | T6 | T7 | T8 | T9 | T10 | T11 | T12 | T13 | T14 | T15 | ΔOBA % | T1 | T2 | T3 | T4 | T5 | T6 | T7 | T8 | T9 | T10 | T11 | T12 | T13 | T14 | T15 |
|---|---|---|---|---|---|---|---|---|---|---|---|---|---|---|---|---|---|---|---|---|---|---|---|---|---|---|---|---|---|---|---|---|
| T1 | 0.6 | -0.1 | 0.6 | 0.2 | 2.7 | 0.2 | 0.0 | 1.3 | 4.5 | -1.0 | 6.1 | 0.0 | 16.1 | 0.0 | 0.0 | -7.5 | 0.3 | -0.1 | 0.5 | 0.1 | 1.2 | 0.1 | 0.1 | 0.6 | 1.9 | -0.4 | 2.1 | 0.0 | 5.3 | 0.0 | 0.0 | -2.4 |
| T2 | 0.7 | 0.1 | 0.0 | 0.0 | 4.9 | 2.4 | 2.0 | 2.3 | -0.6 | 0.0 | 3.3 | 0.0 | 9.7 | 0.0 | 2.7 | -1.9 | 0.3 | 0.1 | 0.1 | 0.0 | 2.2 | 1.1 | 0.8 | 1.1 | -0.2 | 0.1 | 1.1 | 0.0 | 3.2 | 0.0 | 1.0 | -0.6 |
| T3 | 0.7 | -0.1 | 0.1 | 0.9 | 3.6 | 2.1 | -0.4 | 3.6 | 1.9 | 0.0 | 5.0 | 0.0 | 3.2 | -6.7 | -2.7 | -3.8 | 0.3 | 0.0 | 0.3 | 0.4 | 1.6 | 0.9 | 0.0 | 1.6 | 0.8 | 0.0 | 1.9 | 0.0 | 1.1 | -1.7 | -1.0 | -1.2 |
| T4 | 0.6 | -0.1 | 0.3 | 0.1 | 7.6 | -0.5 | 1.6 | 3.2 | 2.5 | -1.5 | 3.9 | 0.0 | 6.5 | -6.7 | 0.0 | -3.8 | 0.3 | 0.0 | 0.3 | 0.0 | 3.2 | -0.1 | 0.7 | 1.5 | 1.1 | -0.5 | 1.3 | 0.0 | 2.1 | -1.7 | 0.0 | -1.2 |
| T5 | 0.7 | -0.4 | 0.6 | 0.1 | 3.6 | 3.3 | -1.2 | 3.2 | 1.3 | 0.0 | 6.6 | 0.0 | 3.2 | -13.3 | -1.4 | -3.8 | 0.3 | -0.1 | 0.4 | 0.1 | 1.6 | 1.4 | -0.4 | 1.5 | 0.6 | 0.0 | 2.4 | 0.0 | 1.1 | -3.3 | -0.5 | -1.2 |
| T6 | 0.8 | 0.0 | 0.7 | 0.6 | 1.8 | 1.9 | 2.4 | 3.2 | 1.3 | -2.0 | 2.8 | 0.0 | 9.7 | -13.3 | 0.0 | -0.6 | 0.4 | 0.0 | 0.0 | 0.5 | 0.3 | 0.8 | 0.9 | 1.0 | 1.5 | -0.7 | 1.0 | 0.0 | 3.1 | -3.3 | 0.0 | -0.6 |
| T7 | 0.8 | -0.1 | 0.6 | 0.0 | 1.3 | 0.9 | 0.0 | 5.8 | 3.8 | 0.5 | 3.3 | 0.0 | 16.1 | 0.0 | -1.4 | -1.9 | 0.4 | 0.0 | 0.5 | 0.0 | 0.6 | 0.5 | 0.1 | 2.5 | 1.6 | 0.3 | 1.2 | 0.0 | 5.3 | 0.0 | -0.5 | -0.6 |
| T8 | 0.7 | -0.2 | 0.7 | 0.1 | 0.4 | 0.9 | -2.0 | 3.6 | 2.5 | 1.5 | 7.7 | 0.0 | 0.0 | -13.3 | 2.7 | 0.0 | 0.3 | -0.1 | 0.5 | 0.1 | 0.2 | 0.4 | -0.7 | 1.6 | 1.0 | 0.6 | 2.8 | 0.0 | 0.0 | -3.3 | 1.0 | 0.0 |
| T9 | 0.8 | -0.1 | 0.7 | 0.7 | 2.2 | 2.1 | 3.6 | 1.0 | 3.8 | -2.0 | 3.3 | 0.0 | 3.2 | 0.0 | 4.1 | -7.5 | 0.4 | 0.0 | 0.6 | 0.4 | 1.0 | 0.9 | 1.6 | 0.5 | 1.6 | -1.0 | 1.1 | 0.0 | 1.1 | 0.0 | 1.5 | -2.4 |
| T10 | 0.7 | 0.0 | 0.4 | -0.1 | 3.1 | 1.7 | 3.2 | 1.3 | 0.6 | 0.5 | 5.5 | 0.0 | 16.1 | 0.0 | 0.0 | -1.9 | 0.4 | 0.0 | 0.5 | -0.1 | 1.4 | 0.8 | 1.5 | 0.6 | 0.3 | 0.3 | 1.7 | 0.0 | 5.3 | 0.0 | 0.0 | -0.6 |
| T11 | 0.7 | -0.1 | 0.7 | 0.6 | 3.1 | 0.0 | 2.4 | 2.3 | 1.9 | -1.0 | 3.3 | 0.0 | 12.9 | 0.0 | -2.7 | -5.7 | 0.3 | 0.0 | 0.5 | 0.3 | 1.4 | 0.0 | 0.9 | 1.1 | 0.8 | -0.4 | 1.2 | 0.0 | 4.2 | 0.0 | -1.0 | -1.8 |
| T12 | 0.8 | -0.1 | 0.6 | 0.5 | 4.0 | 1.9 | -0.4 | 3.6 | 1.3 | 0.0 | 1.1 | 0.0 | 25.8 | -6.7 | 1.4 | -1.9 | 0.4 | 0.0 | 0.5 | 0.2 | 1.8 | 0.8 | 0.0 | 1.6 | 0.6 | 0.1 | 0.4 | 0.0 | 8.4 | -1.7 | 0.4 | -0.6 |
| T13 | 0.7 | -0.4 | 0.4 | 0.2 | 3.1 | 1.4 | -1.2 | 3.6 | 1.3 | 1.5 | 3.3 | 0.0 | 9.7 | 20.0 | 1.4 | 1.9 | 0.3 | -0.2 | 0.4 | 0.1 | 1.3 | 0.7 | -0.3 | 1.6 | 0.6 | 0.6 | 1.2 | 0.0 | 3.2 | 4.9 | 0.4 | 0.6 |
| T14 | 0.5 | -0.2 | 0.2 | 0.7 | 3.6 | 2.1 | 0.8 | 1.0 | 1.9 | -2.0 | 3.9 | 0.0 | 3.2 | -6.7 | 5.4 | -7.5 | 0.3 | -0.1 | 0.2 | 0.3 | 1.6 | 1.0 | 0.4 | 0.5 | 0.8 | -0.8 | 1.4 | 0.0 | 1.1 | -1.7 | 1.8 | -2.4 |
| T15 | 0.7 | -0.1 | 0.7 | 0.7 | 1.8 | 0.0 | 0.8 | 1.6 | 0.6 | -0.5 | 6.1 | 0.0 | 19.4 | -6.7 | 0.0 | 3.8 | 0.4 | 0.0 | 0.5 | 0.4 | 0.8 | 0.0 | 0.4 | 0.8 | 0.3 | -0.2 | 2.2 | 0.0 | 6.3 | -1.7 | 0.0 | 1.1 |
| M0 | 0.86 | 0.98 | 0.94 | 0.91 | 0.80 | 0.79 | 0.66 | 0.78 | 0.67 | 0.74 | 0.54 | 0.95 | 0.48 | 0.33 | 0.55 | 0.45 | 0.92 | 0.99 | 0.95 | 0.95 | 0.90 | 0.89 | 0.82 | 0.88 | 0.83 | 0.86 | 0.76 | 0.97 | 0.74 | 0.67 | 0.77 | 0.73 |

Table 24: Similar to Table 3 and Tables 15∼23, this table shows the impact of the HSW strategy and objective metric $\phi^{\mathrm{of1}}$ (overall F1-score) for classes $T1, T2, \ldots, T15$ in the ticketing dataset.

| | Optimization: HSW-OF1-Class[row]; Performance Metric: ΔOF1%, ΔCR % | | | | | | | | | | | | | | | | Optimization: HSW-OF1-Class[row]; Performance Metric: ΔOBA%, ΔCBA % | | | | | | | | | | | | | | | |
| Class | ΔOF1 % | T1 | T2 | T3 | T4 | T5 | T6 | T7 | T8 | T9 | T10 | T11 | T12 | T13 | T14 | T15 | ΔOBA % | T1 | T2 | T3 | T4 | T5 | T6 | T7 | T8 | T9 | T10 | T11 | T12 | T13 | T14 | T15 |
|---|---|---|---|---|---|---|---|---|---|---|---|---|---|---|---|---|---|---|---|---|---|---|---|---|---|---|---|---|---|---|---|---|
| T1 | 0.6 | -0.3 | 0.6 | 0.4 | 2.2 | 1.7 | 0.4 | 1.9 | 3.8 | -3.4 | 3.9 | 0.0 | 19.4 | -6.7 | 2.7 | -1.9 | 0.3 | -0.1 | 0.5 | 0.2 | 1.0 | 0.7 | 0.3 | 0.9 | 1.5 | -1.4 | 1.4 | 0.0 | 6.3 | -1.7 | 0.9 | -0.6 |
| T2 | 0.6 | -0.1 | 0.4 | 0.1 | 3.1 | 0.9 | 3.2 | 1.3 | -0.6 | -2.0 | 7.2 | 0.0 | 9.7 | -6.7 | -1.4 | 0.0 | 0.3 | -0.1 | 0.2 | 0.0 | 1.4 | 0.5 | 1.4 | 0.7 | -0.2 | -0.8 | 2.5 | 0.0 | 3.2 | -1.7 | -0.5 | 0.0 |
| T3 | 0.6 | -0.1 | 0.3 | 0.4 | 3.1 | 2.6 | 2.0 | 1.6 | -1.9 | 1.0 | 2.8 | 0.0 | 6.5 | 0.0 | 0.0 | -1.8 | 0.3 | 0.0 | 0.3 | 0.1 | 1.3 | 1.1 | 0.8 | 0.8 | -0.7 | 0.6 | 1.1 | 0.0 | 2.1 | 0.0 | 0.0 | -1.8 |
| T4 | 0.7 | -0.4 | 0.4 | -0.1 | 4.4 | 1.9 | -0.4 | 4.9 | 0.6 | 2.0 | 6.1 | 0.0 | 12.9 | -6.7 | -1.4 | -5.7 | 0.4 | -0.1 | 0.5 | 0.0 | 1.9 | 0.9 | 0.0 | 2.2 | 0.2 | 0.9 | 2.2 | 0.0 | 4.2 | -1.7 | -0.5 | -1.8 |
| T5 | 0.7 | -0.1 | 0.8 | 0.2 | 0.4 | 1.9 | 0.8 | 3.2 | 0.6 | 0.5 | 6.1 | 0.0 | 3.2 | -6.7 | -2.7 | 1.9 | 0.4 | 0.0 | 0.6 | 0.1 | 0.2 | 0.8 | 0.4 | 1.5 | 0.3 | 0.2 | 2.3 | 0.0 | 1.1 | -1.7 | -0.9 | 0.6 |
| T6 | 0.7 | | 0.6 | 0.2 | 0.9 | -0.5 | 5.6 | 1.9 | 0.5 | 3.9 | 0.0 | 0.0 | 9.7 | -6.7 | -5.4 | -1.2 | 0.3 | | -0.1 | 0.4 | 0.2 | 0.4 | -0.2 | 2.3 | 1.8 | 0.2 | 1.4 | 0.0 | 3.2 | 1.7 | -1.9 | -1.2 |
| T7 | 0.7 | -0.2 | 0.7 | 0.2 | 2.7 | 1.7 | 1.2 | 5.8 | 3.2 | -3.4 | 3.9 | 0.0 | 6.5 | -6.7 | -1.4 | -5.7 | 0.4 | -0.1 | 0.5 | 0.1 | 1.2 | 0.8 | 0.6 | 2.6 | 1.3 | -1.4 | 1.4 | 0.0 | 2.1 | -1.7 | -0.5 | -1.8 |
| T8 | 0.6 | 0.1 | 0.3 | 0.7 | 2.2 | 1.7 | -0.4 | 1.9 | 5.7 | 1.5 | 2.8 | 0.0 | 3.2 | -6.7 | 0.0 | -7.5 | 0.3 | 0.0 | 0.5 | 0.4 | 1.0 | 0.8 | -0.1 | 0.9 | 2.3 | 0.7 | 0.8 | 0.0 | 1.1 | -1.7 | 0.0 | -2.3 |
| T9 | 0.6 | -0.3 | 0.8 | 0.6 | 2.2 | 0.7 | -1.2 | 3.9 | 4.5 | -1.0 | 3.3 | 0.0 | 3.2 | -6.7 | 0.0 | -9.4 | 0.3 | -0.1 | 0.7 | 0.3 | 1.0 | 0.3 | -0.4 | 1.8 | 1.8 | -0.4 | 1.2 | 0.0 | 1.1 | -1.7 | -0.1 | -2.9 |
| T10 | 1.0 | -0.1 | 0.5 | 1.1 | 4.0 | 2.1 | 0.4 | 3.9 | 3.8 | -1.5 | 3.3 | 0.0 | 9.7 | 0.0 | 1.4 | 1.9 | 0.5 | -0.1 | 0.5 | 0.5 | 1.8 | 1.0 | 0.3 | 1.8 | 1.6 | -0.6 | 1.2 | 0.0 | 3.1 | 0.0 | 0.5 | 0.6 |
| T11 | 0.7 | -0.1 | 0.5 | 1.1 | 2.2 | -0.2 | -1.2 | 2.3 | 4.5 | -1.0 | 8.3 | 0.0 | 3.2 | 0.0 | 0.0 | -5.7 | 0.3 | -0.1 | 0.5 | 0.5 | 1.0 | -0.1 | -0.4 | 1.1 | 1.8 | -0.4 | 2.9 | 0.0 | 1.1 | 0.0 | 0.0 | -1.8 |
| T12 | 0.7 | -0.2 | 0.6 | 0.1 | 3.6 | 0.5 | -1.2 | 2.3 | 4.5 | 0.5 | 7.7 | 0.0 | 19.4 | 0.0 | -1.4 | -1.5 | 0.3 | -0.1 | 0.5 | 0.1 | 1.6 | 0.2 | -0.3 | 1.0 | 1.9 | 0.2 | 2.8 | 0.0 | 6.3 | 0.0 | -0.5 | -2.4 |
| T13 | 0.8 | -0.2 | 0.4 | 0.7 | 2.7 | 1.7 | 1.6 | 3.6 | 3.2 | -2.0 | 4.4 | 0.0 | 9.7 | 20.0 | -1.4 | 0.0 | 0.4 | 0.0 | 0.4 | 0.4 | 1.2 | 0.7 | 0.8 | 1.6 | 1.3 | -0.8 | 1.6 | 0.0 | 3.2 | 4.8 | -0.5 | 0.0 |
| T14 | 0.6 | -0.4 | 0.2 | 0.5 | 3.1 | 1.7 | -2.4 | 3.9 | 3.2 | 0.5 | 5.0 | 0.0 | 16.1 | 0.0 | 0.0 | 0.0 | 0.3 | -0.2 | 0.3 | 0.2 | 1.4 | 0.6 | -0.9 | 1.7 | 1.4 | 0.2 | 1.8 | 0.0 | 5.3 | 0.0 | 0.0 | 0.0 |
| T15 | 0.7 | -0.1 | 0.9 | 0.4 | 1.3 | -0.9 | -2.0 | 2.6 | 1.9 | 2.0 | 4.4 | 0.0 | 16.1 | -6.7 | 0.0 | 11.3 | 0.4 | 0.0 | 0.7 | 0.2 | 0.6 | -0.4 | -0.6 | 1.2 | 0.8 | 0.9 | 1.6 | 0.0 | 5.2 | -1.7 | 0.0 | 3.3 |
| M0 | 0.86 | 0.98 | 0.94 | 0.91 | 0.80 | 0.79 | 0.66 | 0.78 | 0.67 | 0.74 | 0.54 | 0.95 | 0.48 | 0.33 | 0.55 | 0.45 | 0.92 | 0.99 | 0.95 | 0.95 | 0.90 | 0.89 | 0.82 | 0.88 | 0.83 | 0.86 | 0.76 | 0.97 | 0.74 | 0.67 | 0.77 | 0.73 |

Table 25: Similar to Table 3 and Tables 15∼24, this table shows the impact of the GA strategy and objective metric $\phi^{\text{of1}}$ (overall F1-score) for classes $T1, T2, \ldots, T15$ in the ticketing dataset.

| | Optimization: GA-OF1-Class[row]; Performance Metric: ΔOF1%, ΔCR % | | | | | | | | | | | | | | | | Optimization: GA-OF1-Class[row]; Performance Metric: ΔOBA%, ΔCBA % | | | | | | | | | | | | | | | |
|---|---|---|---|---|---|---|---|---|---|---|---|---|---|---|---|---|---|---|---|---|---|---|---|---|---|---|---|---|---|---|---|---|
| | ΔOF1 | ΔCR % | | | | | | | | | | | | | | | ΔOBA | ΔCBA % | | | | | | | | | | | | | | |
| Class | % | T1 | T2 | T3 | T4 | T5 | T6 | T7 | T8 | T9 | T10 | T11 | T12 | T13 | T14 | T15 | % | T1 | T2 | T3 | T4 | T5 | T6 | T7 | T8 | T9 | T10 | T11 | T12 | T13 | T14 | T15 |
| T1 | 0.1 | -0.1 | -0.6 | 1.3 | 0.9 | -0.5 | -1.6 | 0.3 | 0.6 | 2.4 | 7.7 | 0.0 | 6.5 | -20.0 | -4.1 | -3.8 | 0.1 | -0.3 | -0.1 | 0.6 | 0.4 | -0.1 | -0.6 | 0.1 | 0.3 | 1.1 | 2.6 | 0.0 | 2.1 | -5.0 | -1.4 | -1.2 |
| T2 | 0.2 | -0.3 | 0.5 | 0.1 | 1.8 | 0.2 | -2.4 | 2.9 | -2.5 | 1.5 | 0.0 | 0.0 | 9.7 | -33.3 | 1.4 | -1.9 | 0.1 | -0.1 | -0.3 | 0.1 | 0.8 | 0.1 | -0.8 | 1.3 | -0.9 | 0.7 | 0.1 | 0.0 | 3.2 | -8.3 | 0.5 | -0.6 |
| T3 | 0.6 | -0.1 | -0.2 | 1.5 | 2.2 | 2.4 | 4.8 | 1.0 | 0.6 | -2.0 | 8.8 | 0.0 | 6.5 | 0.0 | -8.1 | -9.4 | 0.3 | 0.0 | 0.0 | 0.5 | 1.0 | 1.1 | 2.0 | 0.5 | 0.3 | -0.7 | 3.2 | 0.0 | 2.1 | 0.0 | -2.8 | -2.9 |
| T4 | 0.8 | -0.1 | 0.1 | 0.8 | 4.4 | 0.9 | 2.8 | 2.3 | 3.2 | 1.5 | 6.1 | 0.0 | 0.0 | 6.7 | -1.4 | -5.7 | 0.4 | 0.0 | 0.3 | 0.4 | 1.8 | 0.5 | 1.2 | 1.1 | 1.4 | 0.7 | 2.2 | 0.0 | 0.0 | 1.7 | -0.5 | -1.8 |
| T5 | 0.2 | -0.1 | 0.2 | 0.8 | -1.3 | -1.2 | 1.6 | 1.6 | -5.7 | 3.4 | 3.9 | 0.0 | 6.5 | -13.3 | -4.1 | -5.7 | 0.1 | 0.0 | 0.4 | 0.4 | -0.6 | -1.1 | 0.7 | 0.9 | -2.2 | 1.5 | 1.4 | 0.0 | 2.1 | -3.3 | -1.4 | -1.7 |
| T6 | 0.1 | 0.0 | 0.1 | 0.9 | 0.9 | 1.9 | 3.2 | 2.3 | -4.5 | -3.9 | 2.8 | -1.1 | 3.2 | -13.3 | -4.1 | -20.8 | 0.1 | 0.0 | 0.4 | 0.5 | 0.4 | 0.8 | 0.6 | 1.1 | -1.8 | -1.6 | 1.1 | -0.5 | 1.1 | -3.3 | -1.4 | -6.5 |
| T7 | 0.2 | 0.1 | 0.5 | 0.9 | 3.6 | 1.2 | 2.8 | -1.6 | -6.4 | 0.5 | 0.0 | 0.0 | -6.5 | -13.3 | -2.7 | -7.5 | 0.1 | 0.0 | 0.5 | 0.5 | 1.6 | 0.6 | 1.2 | -0.9 | -2.5 | 0.3 | 0.0 | 0.0 | -2.1 | -3.3 | -0.9 | -2.4 |
| T8 | 0.4 | 0.0 | 0.3 | 0.7 | 1.8 | 1.2 | -3.2 | 2.6 | 0.6 | 1.5 | 3.9 | 0.0 | 3.2 | 0.0 | -4.1 | -3.8 | 0.2 | 0.0 | 0.4 | 0.3 | 0.8 | 0.5 | -1.2 | 1.2 | -0.1 | 0.7 | 1.4 | 0.0 | 1.0 | 0.0 | -1.5 | -1.2 |
| T9 | 0.6 | -0.2 | 0.7 | 0.4 | 1.8 | 3.1 | 0.4 | 2.3 | 2.5 | -2.9 | -2.2 | 0.0 | 12.9 | 0.0 | 4.1 | -3.8 | 0.3 | -0.1 | 0.4 | 0.2 | 0.8 | 1.4 | 0.3 | 1.1 | 1.1 | -1.3 | -0.7 | 0.0 | 4.2 | 0.0 | 1.4 | -1.2 |
| T10 | 0.4 | -0.2 | 0.0 | -0.1 | 0.0 | 2.4 | 2.8 | 2.9 | -3.8 | 2.9 | 4.4 | 0.0 | 6.5 | 0.0 | -1.4 | -5.7 | 0.2 | -0.1 | 0.2 | 0.0 | 0.0 | 1.0 | 1.3 | 1.3 | -1.4 | 1.3 | 1.1 | 0.0 | 2.1 | 0.0 | -0.4 | -1.7 |
| T11 | 0.8 | -0.2 | 0.7 | 0.7 | 2.7 | 2.4 | -0.4 | 3.9 | 3.2 | -1.5 | 3.9 | 0.0 | 3.2 | -6.7 | 1.4 | -5.7 | 0.4 | -0.1 | 0.5 | 0.4 | 1.2 | 1.1 | -0.1 | 1.8 | 1.3 | -0.5 | 1.4 | 0.0 | 1.1 | -1.7 | 0.5 | -1.8 |
| T12 | 0.8 | -0.1 | 0.3 | 0.8 | 1.3 | -0.2 | 1.2 | 3.2 | 5.1 | -1.0 | 7.2 | 0.0 | 22.6 | -6.7 | 1.4 | 0.0 | 0.4 | 0.0 | 0.4 | 0.4 | 0.6 | -0.1 | 0.6 | 1.5 | 2.1 | -0.3 | 2.6 | 0.0 | 7.3 | -1.6 | 0.4 | 0.0 |
| T13 | 0.7 | -0.2 | 0.7 | 0.2 | 2.7 | 0.9 | 2.4 | 3.9 | 3.2 | 0.5 | 2.8 | 0.0 | 9.7 | 13.3 | -9.5 | 0.0 | 0.4 | 0.0 | 0.6 | 0.1 | 1.1 | 0.5 | 1.1 | 1.7 | 1.3 | 0.2 | 1.0 | 0.0 | 3.2 | 3.2 | -3.3 | 0.0 |
| T14 | 0.3 | 0.0 | 0.6 | -0.2 | -0.4 | 0.0 | 0.4 | 1.0 | -1.9 | 0.5 | 1.7 | 0.0 | 0.0 | -6.7 | 10.8 | -3.8 | 0.1 | 0.0 | 0.5 | -0.1 | -0.2 | 0.1 | 0.2 | 0.5 | -0.7 | 0.3 | 0.7 | 0.0 | 0.0 | -1.7 | 3.3 | -1.2 |
| T15 | 0.0 | -0.1 | -0.4 | 0.1 | -1.3 | 1.4 | -3.6 | 1.9 | -1.9 | 0.5 | 4.4 | 0.0 | 3.2 | -13.3 | -4.1 | 15.1 | 0.0 | 0.0 | 0.0 | 0.0 | -0.6 | 0.6 | -1.4 | 0.9 | -0.7 | 0.3 | 1.6 | 0.0 | 1.1 | -3.3 | -1.4 | 4.3 |
| M0 | 0.86 | 0.98 | 0.94 | 0.91 | 0.80 | 0.79 | 0.66 | 0.78 | 0.67 | 0.74 | 0.54 | 0.95 | 0.48 | 0.33 | 0.55 | 0.45 | 0.92 | 0.99 | 0.95 | 0.95 | 0.90 | 0.89 | 0.82 | 0.88 | 0.83 | 0.86 | 0.76 | 0.97 | 0.74 | 0.67 | 0.77 | 0.73 |

## C   Pilot Experiments: Parameter for ML Training

Before developing the AutoGeTS framework, we aimed to improve the original CatBoost model, $M0$, through parameter tuning. A grid search was conducted with the following parameter ranges:

- learning_rate: [0.01, 0.05, 0.1, 0.2, 0.5]

- depth: [4, 6, 8, 10]

- l2_leaf_reg: [1, 3, 5, 10]

Five-fold cross-validation was employed, with overall classification accuracy as the primary criterion for evaluating the model performance.

### C.1   Benchmark M0 Parameters Experiments

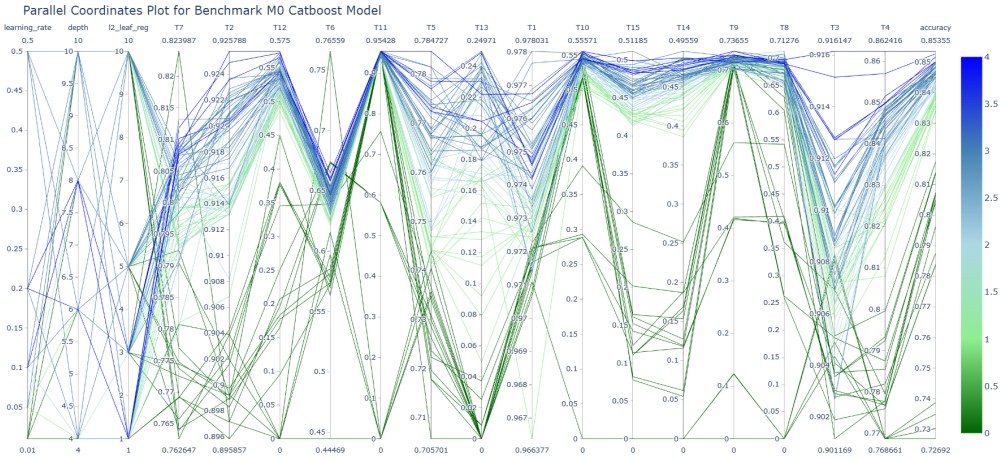

Figure 7: M0 Parameter Experiments Overview.

Figure 7 presents an overview of all experimented M0 CatBoost model parameters in a parallel coordinate plot. Each line represents a unique parameter set and its corresponding classification performance. The first three coordinates depict the experimented parameters: learning rate, tree depth, and L2 leaf regularization. The subsequent 15 coordinates (T1 to T15) represent the recall for each of the 15 classes, while the final coordinate shows the overall classification accuracy, which is the primary performance metric.

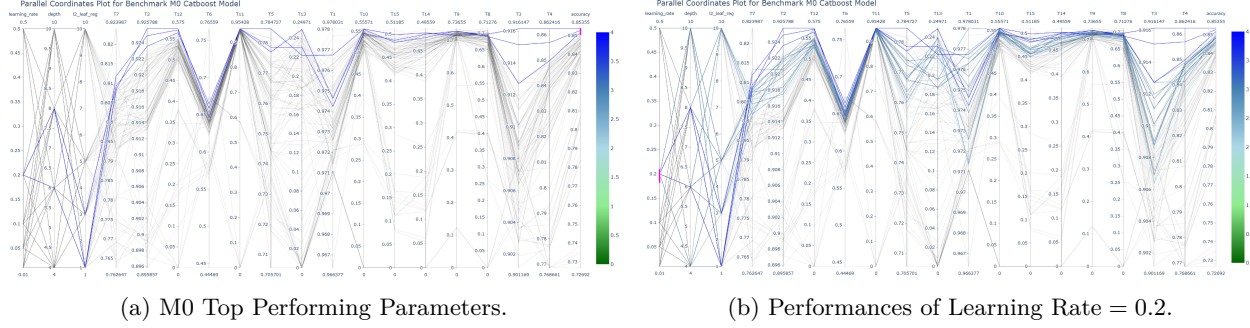

(a) M0 Top Performing Parameters.

(b) Performances of Learning Rate = 0.2.

Figure 8: M0 Top Performances and Learning Rate = 0.2.

Figure 8a highlights the top-performing parameter sets. These sets consistently use a learning rate of 0.2, tree depths of 6 or 8, and L2 leaf regularization values of 1 or 3. Based on these observations, we further

examine the performance of learning_rate = 0.2 and determine the optimal values for depth and L2 leaf regularization.

Figure 8b highlights parameter sets with learning_rate = 0.2. All highlighted sets demonstrate good accuracy, including the three best accuracy scores, confirming 0.2 as the optimal learning rate among the experimented values.

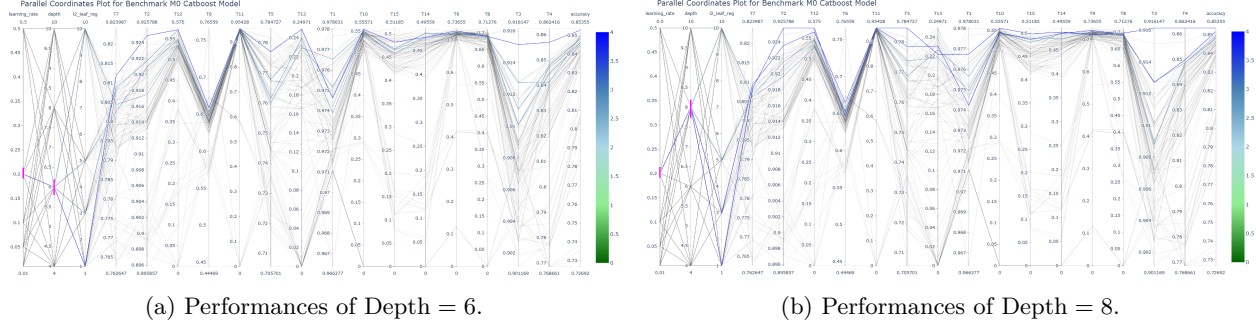

(a) Performances of Depth = 6.  (b) Performances of Depth = 8.

Figure 9: M0 Best Experimented Depth Value.

Figures 9a and 9b compare model performances between depth = 6 and depth = 8 with learning rate set to 0.2. While both depth values yield good accuracy, depth = 8 shows slightly superior results overall.

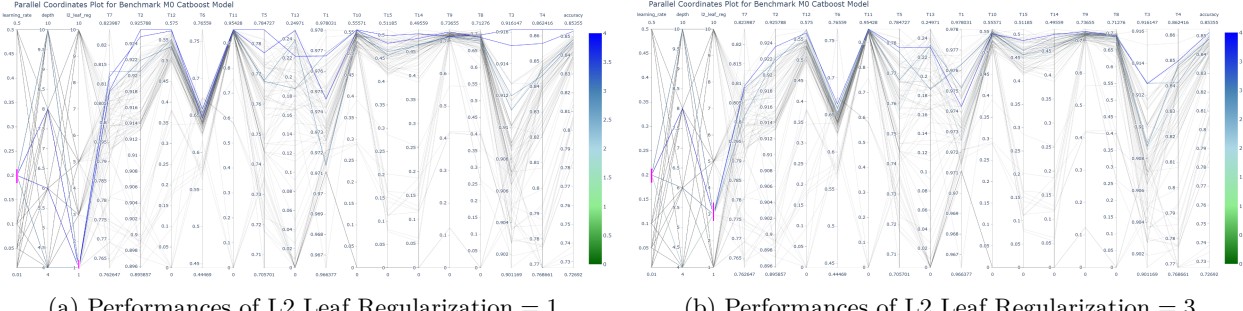

(a) Performances of L2 Leaf Regularization = 1.  (b) Performances of L2 Leaf Regularization = 3.

Figure 10: M0 Best Experimented L2 Leaf Regularization Value.

Figures 10a and 10b compare model performances between L2_leaf_regularization = 1 and L2_leaf_regularization = 3 with learning rate set to 0.2. Both values produce good performances, but L2_leaf_regularization = 1 demonstrates marginally better results.

Based on these analyses, the optimal parameter set for the M0 benchmark CatBoost model is:

- learning_rate = 0.2
- depth = 8
- L2_leaf_regularization = 1

This parameter set is used for all experiments involving the CatBoost model.

## C.2 Further Analysis on M0

Despite identifying a relatively optimal parameter set, analysis of class-specific performance revealed significant shortcomings. As shown in Table 1, several classes, particularly small or underrepresented ones (T12, T13, T14, and T15), exhibited unacceptable performance levels. These classes had balanced accuracies below 0.8 and recall rates around or below 0.5, potentially causing severe delays in messages and error reports processing.

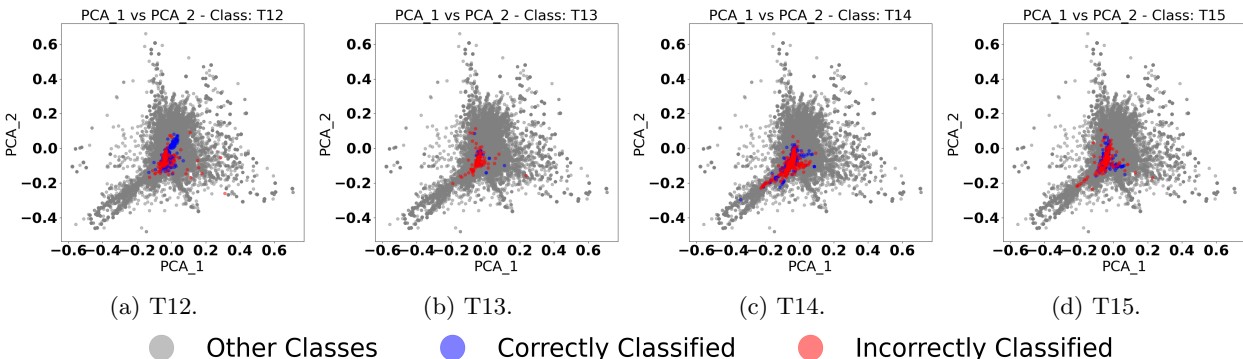

Figure 11: M0 PCA Plots for Small Classes T12, T13, T14, T15.

To this end, we further realised that a serious class imbalance problem exists in the dataset with these small classes ranging only from 0.5% to 2% of the whole dataset, and 1.6% to 6.7% of the largest class. Moreover, data scarcity exists in these small classes, as illustrated in figure 11 that the red and blue dots distribute loosely across the plot. We therefore decided to investigate the use of synthetic data to improve this text classification model.

# D Pilot Experiments: Parameters for Example Search

In developing the AutoGeTS framework described in Section 3, we found that parameters for both the LLM's synthetic sample generation and the three example selection strategies significantly influenced AutoGeTS performance. To determine optimal parameter sets and understand their impact, we conducted extensive experiments on each component.

Given that our objectives for each retrained model were to maximize both overall performance (overall accuracy was used) and class-specific performance (class-based recall was used) for the chosen class, we employed the Hypervolume (HV) indicator (Zitzler & Thiele (1999); Jiang et al. (2014)) to evaluate performance. This indicator allows us to compare results across different parameter configurations by considering both overall accuracy and class-based recall simultaneously.

We implemented 5-fold cross-validation throughout our experiments. In addition to the HV indicator, we tracked the best overall accuracy and best class-based recall across all five folds as supplementary performance metrics.

## D.1 Synthetic Data Generation Parameter Experiments

The synthetic data generation process utilizes GPT-3.5 through its API interface. In preliminary experiments, we also evaluated Llama 3 as an alternative language model for synthetic sample generation, which yielded comparable results.

For each original text sample, we invoke a new chat session and employ a zero-shot approach without providing additional context. The input prompt template for generating synthetic samples follows this format:

```
'Generate ' + str(num) + ' lines of the data similar to this format data: + '
    meta_data['text'].values[i] ' + 'put & at the end of each line'
```

where 'num' is the number of generated samples, meta_data is the input data, and 'i' is the data index number.

Upon receiving the LLM's response, we implement an automated pipeline for cleaning, parsing, and separating the output into 'num' synthetic samples based on the formatting parameters specified in the prompt template. A verification function inspects each synthetic sample for format quality assurance, checking includes:

- Empty samples or null responses

- Extraneous empty lines or spaces

- Correct placement of separation symbols ('&')

If 'm' samples fail these quality checks, the generation process is automatically repeated for the same input text, adjusting the prompt to request only the remaining 'm' samples (i.e., 'Generate ' + str(m) + ' lines...').

We investigated the impact of varying 'num' on AutoGeTS performance through a series of experiments.

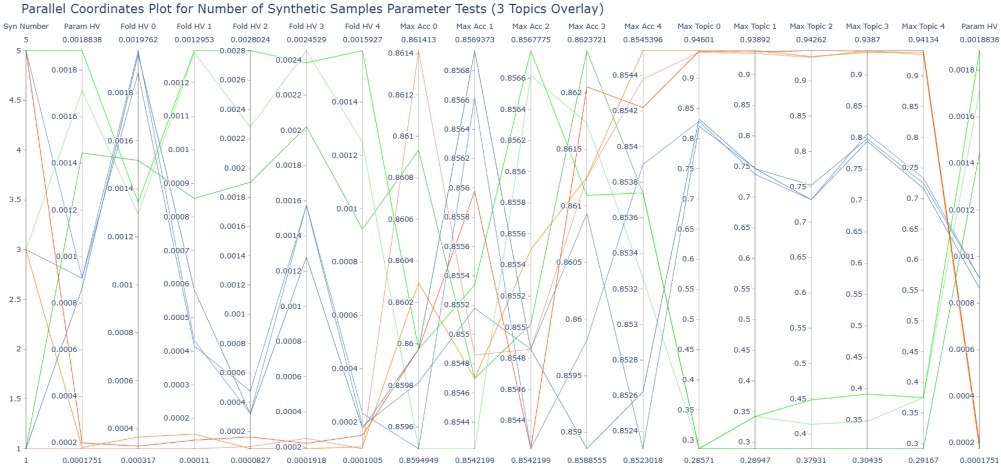

Figure 12: Synthetic Data Generation Parameter Experiments Overview (F0_HV: Fold 0 Hypervolume, MaxO: Max Overall Accuracy Improvement, MaxT: Max Topic Recall Improvement).

Figure 12 presents a parallel coordinate plot of synthetic text generation parameter experiments, focusing on classes T2 (orange, largest class), T9 (blue, median size class), and T13 (green, smallest class). The primary parameter under investigation, Syn Number, represents the number of synthetic text samples generated for each selected original text data point. The primary criterion, HV, appears as both the second-left and rightmost coordinates in the plot.

The results indicate that Syn Number = 5 consistently yielded the best Hypervolume for all three classes among the tested values. Consequently, we adopted the generation of five synthetic samples per selected original data point for all subsequent experiments.

### D.1.1 PCA Projection of Synthetic Samples

To verify the effectiveness of generated synthetic samples in addressing class imbalance and data scarcity, we projected these data using the same fitted vectorizer and PCA model used for the original data.

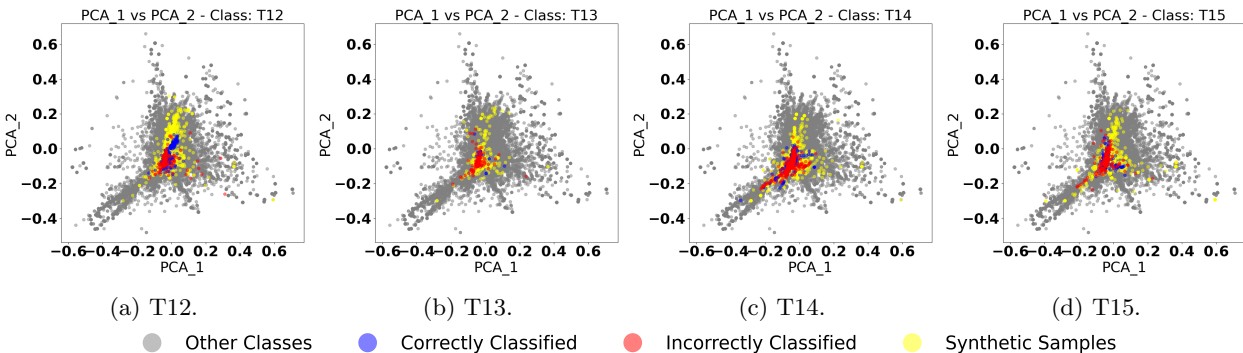

(a) T12.  (b) T13.  (c) T14.  (d) T15.

⬤ Other Classes   ⬤ Correctly Classified   ⬤ Incorrectly Classified   ⬤ Synthetic Samples

Figure 13: M0 PCA Plots with Synthetic Samples for Small Classes T12, T13, T14, T15.

Figure 13 presents updated PCA plots for small classes T12, T13, T14, and T15. The inclusion of synthetic samples significantly increased the number of colored data points. Moreover, these additional points appear more densely clustered, suggesting improvement on the previous data scarcity issues.

To illustrate the synthetic data generated, we present an example of an original data example and its corresponding LLM-generated synthetic samples. The following listing shows the original Spanish text

followed by five synthetic samples, demonstrating how the LLM maintains the context and structure while introducing variations in content:

Listing 1: Original data example and LLM-generated synthetic samples

```
Original Data Example:
phone_nmb
En el ticket REQ0026231 se le solicita acceso a unidades de red pero sigue sin
    poder acceder y figura como resuelto, revisar por favor, la usuaria lleva 2
    meses con este problema

Synthetic Samples:
1. El cliente reporta que su cuenta de correo electronico ha sido bloqueada, por
    favor revisar el caso REQ0027456.

2. Se solicita la instalacion de un software especifico en el equipo del usuario,
    el ticket es REQ0028745.

3. El usuario indica que no puede imprimir desde su equipo, se necesita revision
    del caso REQ0029367.

4. La usuaria reporta problemas con su conexion a internet, el ticket es
    REQ0030172.

5. Se requiere la asignacion de permisos adicionales en el sistema para el usuario
    , el caso es REQ0031298.
```

As evident from Listing 1, the synthetic samples maintain the overall structure of a ticketing system entry while diversifying the reported issues.

### D.1.2 Synthetic Samples Performances without Examples Selection

To evaluate the potential of our generated synthetic samples in improving the text classification model $M0$, we appended all generated synthetic samples to the training set and retrained the CatBoost model $M0$ for small classes T12, T13, T14, and T15.

Tables 26 and 27 present the results of this experiment. We observed that class-based performance could be improved when synthetic samples for that class were appended, demonstrating the potential of synthetic data. However, only T12 showed improvement in overall performance. Notably, T13 failed to improve even its class-specific performance.

These mixed results suggest that indiscriminate use of all generated synthetic samples may not consistently yield improvements. This observation suggested us that a selective approach to choosing text examples for synthetic data generation is necessary.

### D.1.3 Synthetic Samples Performances with Random Examples Selection

To establish a baseline for evaluating selection strategies, we also implemented a random selection approach. For each target class, this baseline process randomly samples a random number (between 1 and the size of the class pool) of examples with replacement, where selected examples subsequently go through the synthetic data generation process described at the beginning of Appendix D.1. We evaluated this baseline using 1 GPU hour fixed-time experiments to improve M0, maintaining consistency with the experimental settings described in Section 4.

The optimization process ran multiple iterations for 1 GPU hour, averaging 299 iterations per class, with 4487 sets of random selected examples tested in total. Figure 14 presents the performance distribution of models retrained using each example set. The overall balanced accuracy (OBA) improvement ranged from -0.9% to 0.2%, with an average of -0.3% and only 3.9% of retraining leading to positive OBA improvement.

Table 26: Performance of Retrained Models with T12 and T13 Synthetic Samples

| Class | Δ Balanced Accuracy | | Δ Recall | | Δ F1-Score | |
|---|---|---|---|---|---|---|
| | T12 | T13 | T12 | T13 | T12 | T13 |
| T1 | ▼0.0014 | ▼0.0011 | ▼0.0034 | ▼0.0023 | ▼0.0006 | ▼0.0009 |
| T2 | ▲0.0036 | ▼0.0033 | ▲0.0044 | ▼0.0058 | ▲0.0053 | ▼0.0040 |
| T3 | ▲0.0015 | ▼0.0003 | ▲0.0032 | ▲0.0011 | ▲0.0013 | ▼0.0054 |
| T4 | ▲0.0104 | ▼0.0001 | ▲0.0214 | 0.0000 | ▲0.0064 | ▼0.0016 |
| T5 | ▼0.0015 | ▼0.0018 | ▼0.0038 | ▼0.0038 | ▲0.0022 | ▼0.0015 |
| T6 | ▼0.0021 | ▲0.0036 | ▼0.0027 | ▲0.0080 | ▼0.0102 | ▼0.0003 |
| T7 | ▲0.0023 | ▼0.0039 | ▲0.0025 | ▼0.0076 | ▲0.0161 | ▼0.0065 |
| T8 | ▲0.0046 | ▼0.0104 | ▲0.0085 | ▼0.0212 | ▲0.0144 | ▼0.0101 |
| T9 | ▼0.0011 | ▼0.0066 | ▼0.0036 | ▼0.0144 | ▲0.0093 | ▲0.0014 |
| T10 | ▲0.0094 | ▲0.0043 | ▲0.0179 | ▲0.0090 | ▲0.0192 | ▲0.0041 |
| T11 | ▲0.0001 | 0.0000 | 0.0000 | 0.0000 | ▲0.0053 | 0.0000 |
| T12 | ▲0.0376 | ▼0.0234 | ▲0.0781 | ▼0.0469 | ▼0.0497 | ▼0.0364 |
| T13 | ▼0.0333 | ▼0.0127 | ▼0.0667 | ▼0.0222 | ▼0.0753 | ▼0.1506 |
| T14 | ▲0.0110 | ▲0.0040 | ▲0.0222 | ▲0.0074 | ▲0.0144 | ▲0.0184 |
| T15 | ▼0.0045 | ▼0.0298 | ▼0.0085 | ▼0.0598 | ▼0.0177 | ▼0.0543 |
| Overall | ▲0.0015 | ▼0.0023 | ▲0.0028 | ▼0.0043 | ▲0.0028 | ▼0.0043 |

Table 27: Performance of Retrained Models with T14 and T15 Synthetic Samples

| Class | Δ Balanced Accuracy | | Δ Recall | | Δ F1-Score | |
|---|---|---|---|---|---|---|
| | T14 | T15 | T14 | T15 | T14 | T15 |
| T1 | ▼0.0011 | ▼0.0004 | ▼0.0029 | ▼0.0017 | ▼0.0003 | ▲0.0005 |
| T2 | ▼0.0014 | ▼0.0039 | ▼0.0071 | ▼0.0097 | ▲0.0011 | ▼0.0030 |
| T3 | ▼0.0027 | ▼0.0027 | ▼0.0054 | ▼0.0043 | ▼0.0029 | ▼0.0063 |
| T4 | ▼0.0071 | ▲0.0069 | ▼0.0142 | ▲0.0142 | ▼0.0071 | ▲0.0037 |
| T5 | ▼0.0035 | ▼0.0044 | ▼0.0075 | ▼0.0094 | ▼0.0023 | ▼0.0027 |
| T6 | ▲0.0023 | ▼0.0027 | ▲0.0054 | ▼0.0054 | ▼0.0012 | ▼0.0042 |
| T7 | ▲0.0073 | ▲0.0023 | ▲0.0152 | ▲0.0051 | ▲0.0053 | ▼0.0007 |
| T8 | ▼0.0080 | ▼0.0104 | ▼0.0169 | ▼0.0212 | ▼0.0013 | ▼0.0101 |
| T9 | ▼0.0024 | ▼0.0031 | ▼0.0072 | ▼0.0072 | ▲0.0165 | ▲0.0047 |
| T10 | ▼0.0095 | ▲0.0016 | ▼0.0179 | ▲0.0030 | ▼0.0200 | ▲0.0039 |
| T11 | 0.0000 | 0.0000 | 0.0000 | 0.0000 | 0.0000 | 0.0000 |
| T12 | ▼0.0156 | ▲0.0001 | ▼0.0313 | 0.0000 | ▼0.0278 | ▲0.0122 |
| T13 | 0.0000 | ▼0.0111 | 0.0000 | ▼0.0222 | 0.0000 | ▼0.0243 |
| T14 | ▲0.0135 | ▼0.0259 | ▲0.0370 | ▼0.0519 | ▼0.1219 | ▼0.0388 |
| T15 | ▼0.0128 | ▲0.0219 | ▼0.0256 | ▲0.0513 | ▼0.0241 | ▼0.1079 |
| Overall | ▼0.0026 | ▼0.0026 | ▼0.0049 | ▼0.0049 | ▼0.0049 | ▼0.0049 |

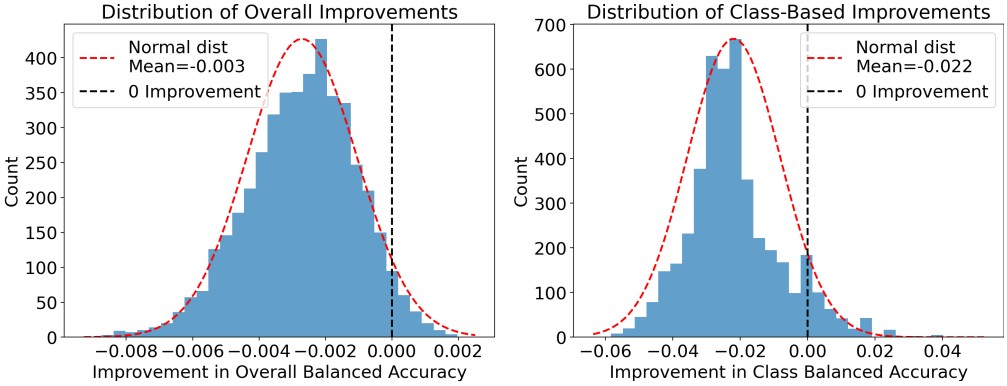

Figure 14: Random Selection Results Distribution on Improving OBA (left) and CBA (Right).

At the class level, the class balanced accuracy (CBA) improvement fluctuated between -5.8% and 4.7%, with an average of -2.2% and 9.1% of retraining achieving positive CBA improvement.

Table 28: Performance Comparison between Random Selection and Strategic Selections.

| Class Name | Random | | Sliding Window | | Hierarchical SW | | Genetic Algorithm | |
|---|---|---|---|---|---|---|---|---|
| | Overall | Class | Overall | Class | Overall | Class | Overall | Class |
| T2 | ▲0.0012 | ▼0.0155 | ▲0.0030 | ▲0.0050 | ▲0.0034 | ▲0.0048 | ▲0.0009 | ▼0.0010 |
| T1 | ▲0.0009 | ▼0.0202 | ▲0.0028 | ▲0.0005 | ▲0.0029 | ▲0.0005 | ▲0.0018 | ▼0.0018 |
| T3 | ▲0.0014 | ▼0.0148 | ▲0.0030 | ▲0.0058 | ▲0.0027 | ▲0.0062 | ▲0.0029 | ▲0.0069 |
| T5 | ▲0.0002 | ▼0.0136 | ▲0.0032 | ▲0.0189 | ▲0.0034 | ▲0.0140 | ▲0.0010 | ▲0.0059 |
| T7 | ▲0.0018 | ▼0.0091 | ▲0.0036 | ▲0.0228 | ▲0.0034 | ▲0.0226 | ▲0.0012 | ▼0.0026 |
| T6 | ▲0.0016 | ▼0.0124 | ▲0.0035 | ▲0.0190 | ▲0.0030 | ▲0.0196 | ▲0.0015 | ▲0.0073 |
| T10 | ▼0.0006 | ▲0.0051 | ▲0.0034 | ▲0.0281 | ▲0.0044 | ▲0.0247 | ▲0.0027 | ▲0.0208 |
| T9 | ▲0.0016 | ▲0.0049 | ▲0.0036 | ▲0.0147 | ▲0.0027 | ▲0.0191 | ▲0.0026 | ▲0.0077 |
| T4 | ▲0.0018 | ▲0.0118 | ▲0.0029 | ▲0.0304 | ▲0.0033 | ▲0.0369 | ▲0.0036 | ▲0.0323 |
| T8 | ▲0.0012 | ▲0.0146 | ▲0.0030 | ▲0.0321 | ▲0.0029 | ▲0.0358 | ▲0.0020 | ▲0.0142 |
| T14 | ▲0.0009 | ▲0.0029 | ▲0.0023 | ▲0.0326 | ▲0.0029 | ▲0.0395 | ▲0.0019 | ▲0.0396 |
| T15 | ▲0.0005 | ▲0.0067 | ▲0.0033 | ▲0.0456 | ▲0.0034 | ▲0.0446 | ▲0.0037 | ▲0.0533 |
| T11 | ▲0.0020 | ▼0.0248 | ▲0.0030 | ▲0.0054 | ▲0.0030 | ▲0.0053 | ▲0.0039 | ▲0.0053 |
| T12 | ▲0.0014 | ▲0.0472 | ▲0.0037 | ▲0.0699 | ▲0.0032 | ▲0.0772 | ▲0.0036 | ▲0.0775 |
| T13 | ▲0.0006 | ▲0.0237 | ▲0.0030 | ▲0.0443 | ▲0.0037 | ▲0.0548 | ▲0.0034 | ▲0.0548 |

The best results achieved through random selection are presented in Table 28. While overall performance improvements were observed, the random selection struggled particularly with improving class performance for larger classes.

One set of plots showing the overall balanced accuracy (OBA) improvements over time for the 15 classes is presented in Figure 15, where the random selection benchmark results are shown in gray dash-lines. We can observe that random selection frequently achieves only marginal improvements over the original model, as evidenced in classes T7, T13, T14, and T15. Moreover, for T10, the random selection approach performed worse than the original model.

These findings motivated us to develop the three example selection strategies and conduct parameter studies for each, which we present in the following sections.

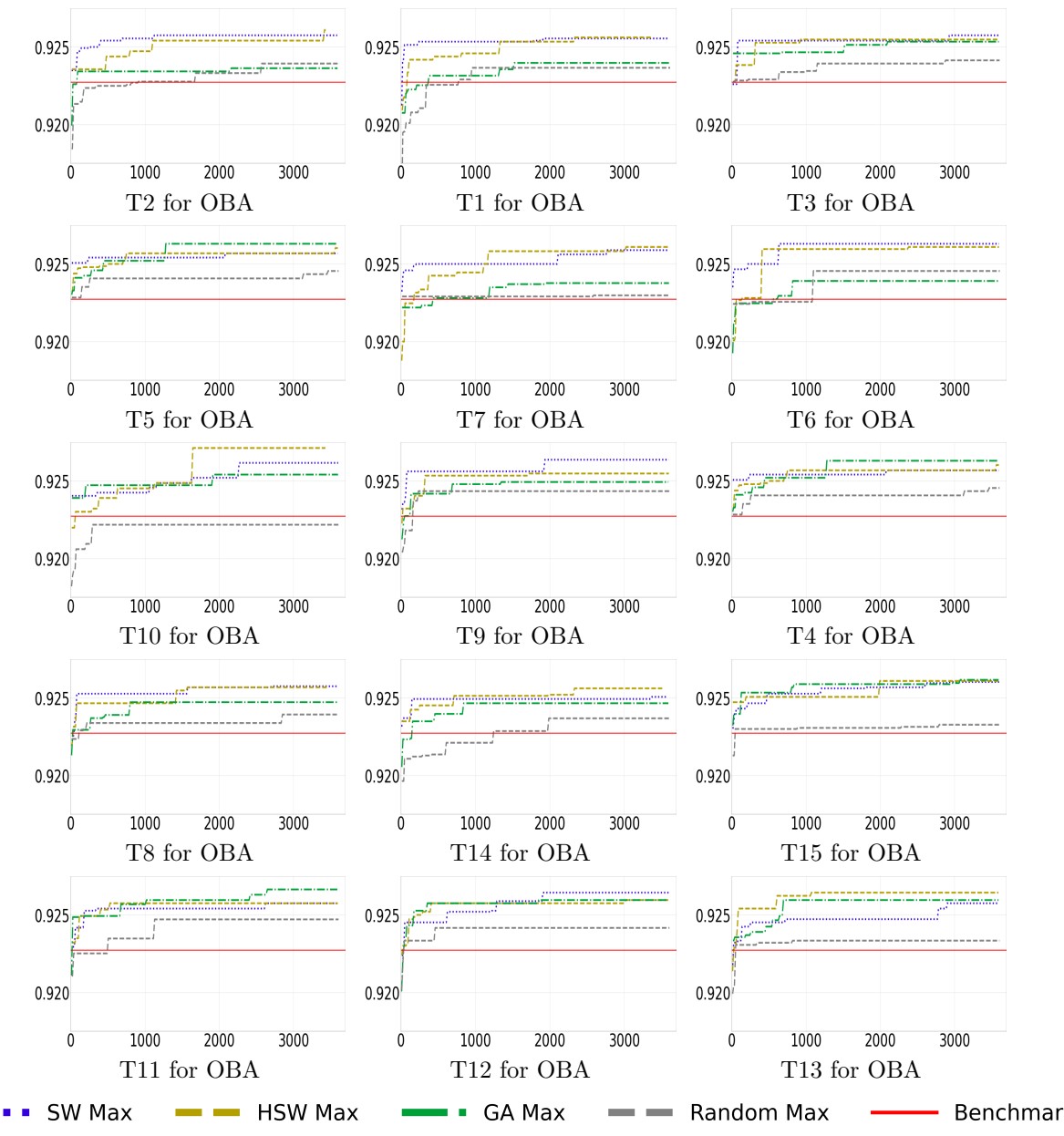

Figure 15: Fixed 1 Hour GPU time (x-axis, in seconds), Comparing on OBA Improvement (y-axis).

### D.2 Example Selection Strategies Parameter Experiments

### D.2.1 Sliding Window (SW) Parameters Experiments

We investigated the parameters of the Sliding Window strategy. Figure 16a presents an overview of the experimental results. The parameters under investigation are:

- Area Size (AS): the size of the area to which the sliding window is applied

- Num Seg (NS): the number of segments/bins per dimension

- Window Size (WS): the number of bins per window

These parameters are represented by the three leftmost coordinates in the plot. We used the Hypervolume Indicator as the primary comparison metric, supplemented by the maximum values of both objectives in each of the 5 cross-validation folds.

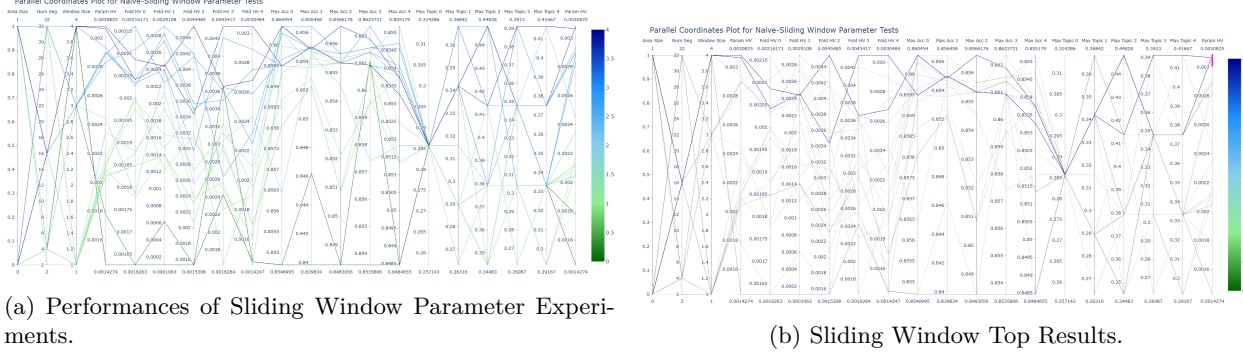

(a) Performances of Sliding Window Parameter Experiments.

(b) Sliding Window Top Results.

Figure 16: Sliding Window Parameter Experiments Overview and Top Results.

Figure 16b highlights the top-performing parameter sets. The two best configurations both used Number of Segments (Num Seg) = 16 and Window Size = 4. Subsequent analysis focuses on verifying the effectiveness of these values and determining the optimal Area Size.

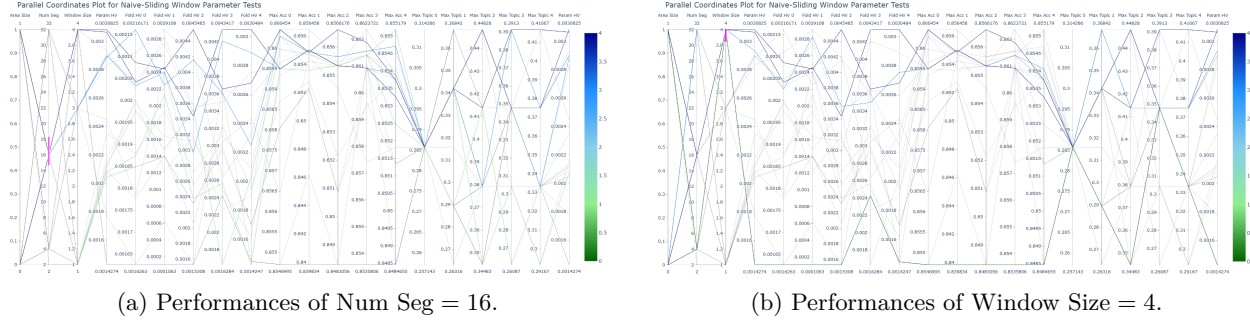

(a) Performances of Num Seg = 16.

(b) Performances of Window Size = 4.

Figure 17: Sliding Window Num Seg and Window Size Parameter Experiments.

Figure 17a shows that Num Seg = 16 yields both top and suboptimal results, with consistently good performance when Window Size > 1. Similarly, Figure 17b demonstrates that Window Size = 4 produces good results when Num Seg ≠ 4. These findings confirm that Num Seg = 16 and Window Size = 4 are optimal values among those tested.

Figures 18a and 18b compare the two Area Size values tested:

- FullSize: using the minimum and maximum values from the entire dataset for each dimension

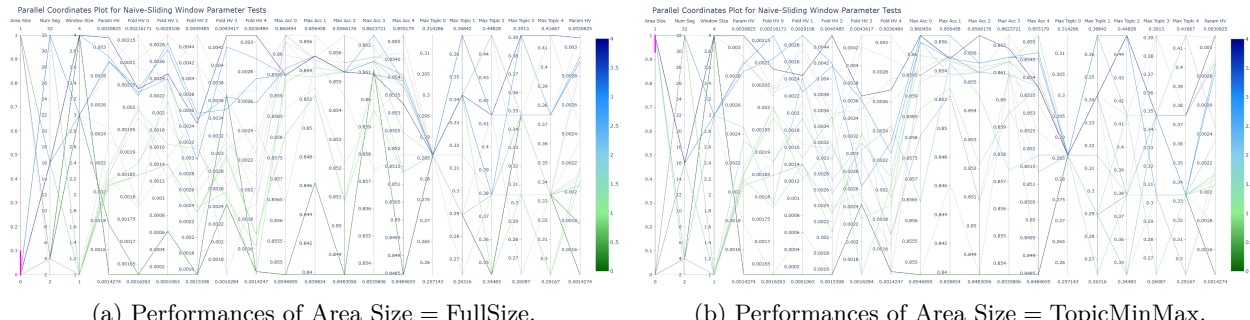

(a) Performances of Area Size = FullSize.  (b) Performances of Area Size = TopicMinMax.

Figure 18: Sliding Window Area Size Parameter Experiments.

- TopicMinMax: using the minimum and maximum values only from the selected topic/class

The results show no clear advantage for either option. Therefore, we selected the Area Size that produced the higher hypervolume given Num Seg = 16 and Window Size = 4. In conclusion, the experimentally determined optimal parameter set for the Sliding Window strategy is:

- Area Size = TopicMinMax

- Num Seg = 16

- Window Size = 4

### D.2.2  Hierarchical Sliding Window (HSW) Parameters Experiments

We investigated the parameters of the Hierarchical Sliding Window (HSW) strategy. Figure 19a presents an overview of the parameter experiments. The parameters under investigation are:

- Area Size (AS)

- Window Size (WS)

- Level 0 Num Seg (NS0)

- Level 1 Num Seg (NS1)

- Level 2 Num Seg (NS2, using a 3-level structure for HSW)

These parameters are represented by the five leftmost coordinates in the plot. We used the Hypervolume Indicator as the primary comparison metric, supplemented by the maximum values of both objectives in each of the 5 cross-validation folds.

Figure 19b highlights the top two parameter sets, both using Area Size = TopicMinMax and Window Size = Half of each level's Num Seg. Subsequent analysis focuses on verifying these parameter choices and determining optimal Num Seg values for each level.

Figures 20a and 20b compare FullSize and TopicMinMax Area Size values. While both show variable performance, TopicMinMax yields more top results, leading to its selection.

Figures 21a and 21b compare the Window Size of Half and 1. Window Size = Half clearly outperforms, leading to its selection.

Figures 22a and 22b compare Level 0 Num Seg values of 8 and 2, when Window Size = Half. Num Seg = 8 shows more consistent good performance, leading to its selection.

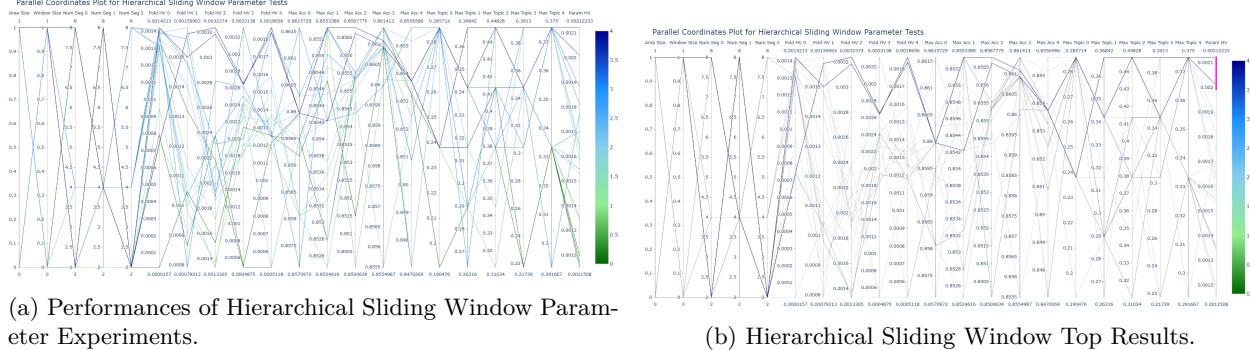

(a) Performances of Hierarchical Sliding Window Parameter Experiments.

(b) Hierarchical Sliding Window Top Results.

Figure 19: Hierarchical Sliding Window Parameter Experiments Overview and Top Results.

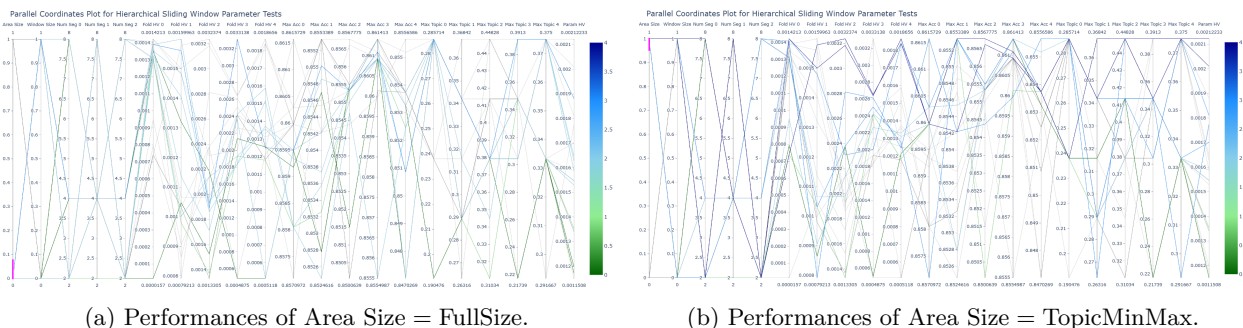

(a) Performances of Area Size = FullSize.

(b) Performances of Area Size = TopicMinMax.

Figure 20: Hierarchical Sliding Window Area Size Parameter Experiments.

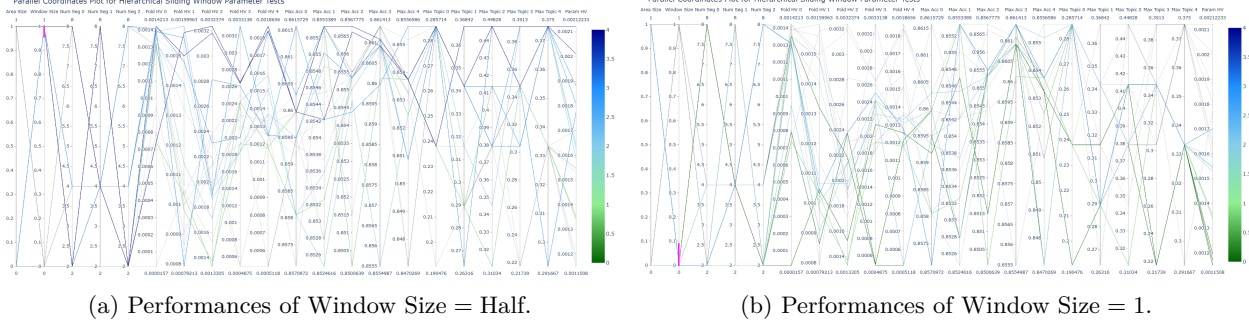

(a) Performances of Window Size = Half.

(b) Performances of Window Size = 1.

Figure 21: Hierarchical Sliding Window Window Size Parameter Experiments.

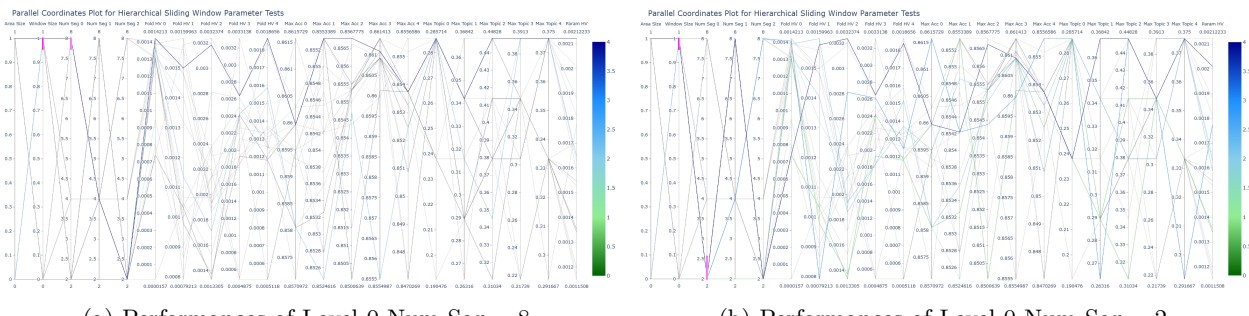

(a) Performances of Level 0 Num Seg = 8.

(b) Performances of Level 0 Num Seg = 2.

Figure 22: Hierarchical Sliding Window Level 0 Number of Segments Parameter Experiments.

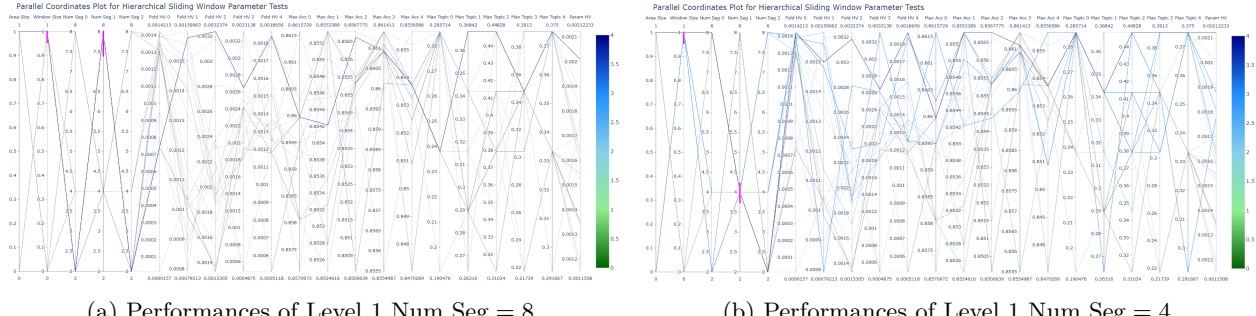

(a) Performances of Level 1 Num Seg = 8.

(b) Performances of Level 1 Num Seg = 4.

Figure 23: Hierarchical Sliding Window Level 1 Number of Segments Parameter Experiments.

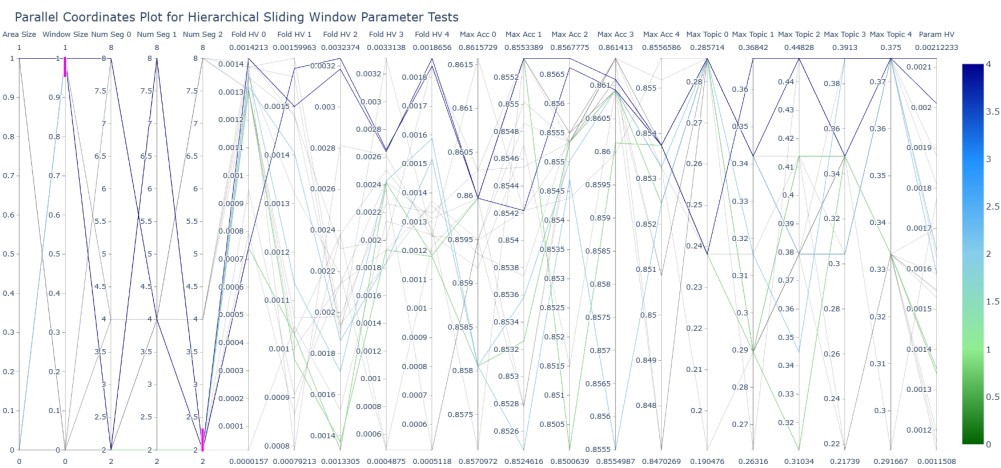

Figure 24: Performances of Level 2 Num Seg = 2.

Figures 23a and 23b compare Level 1 Num Seg values of 8 and 4. While both yield top results, Num Seg = 4 shows better overall performance, leading to its selection.

Figure 24 shows results for Level 2 Num Seg = 2. Performance is consistently good when Level 1 Num Seg ≠ 2, aligns with our previous parameter choices. Therefore, with Level 0 Num Seg = 8 and Level 1 Num Seg = 4, Level 2 Num Seg = 2 will provide top results.

In conclusion, the optimal parameter set for the Hierarchical Sliding Window strategy is:

- Area Size = TopicMinMax
- Window Size = Half of each Level's Num Seg
- Level 0 Num Seg = 8
- Level 1 Num Seg = 4
- Level 2 Num Seg = 2

### D.2.3 Genetic Algorithm (GA) Parameters Experiments

We investigated the parameters for the Genetic Algorithm (GA) strategy. Figure 25 presents an overview of the parameter experiments. The parameters under investigation are:

- Population Size and Selection Size (PSSize)
- Crossover Rate and Initial Mutation Rate (CMRate)

These parameters are represented by the four leftmost coordinates in the plot. We used the Hypervolume Indicator as the primary comparison metric, supplemented by the maximum values of both objectives in each of the 5 cross-validation folds.

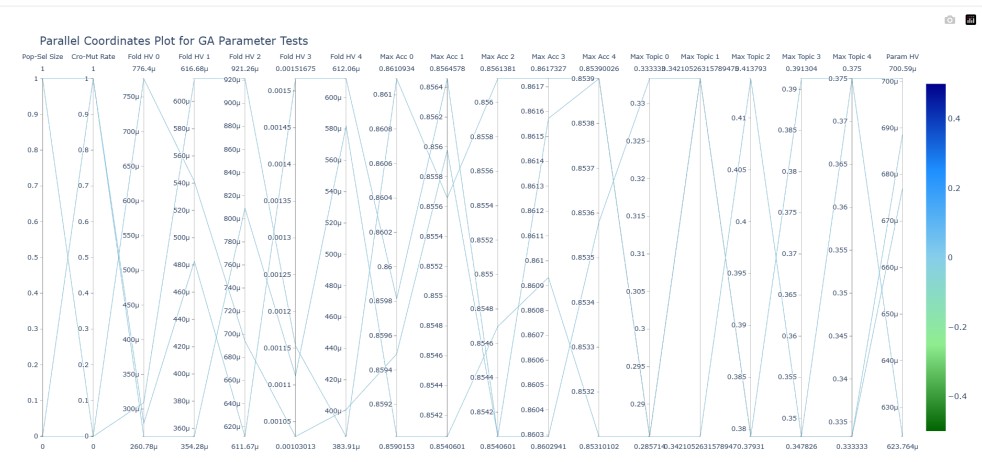

Figure 25: Genetic Algorithm Parameters Experiment Overview (PSSize: Population-Selection Size, CM-Rate: Crossover-Mutation Rate).

Figure 26a illustrates the performance when both population size and selection size are set to 20. These results consistently outperform the alternative configuration of population size 20 and selection size 10.

Figure 26b shows the performance with a crossover rate of 0.7 and an initial mutation rate of 0.3. This configuration demonstrates superior performance compared to the alternative rates of 0.9 and 0.1, respectively, when population and selection sizes are held constant.

Based on these experiments, we determined the optimal parameter set for the Genetic Algorithm strategy:

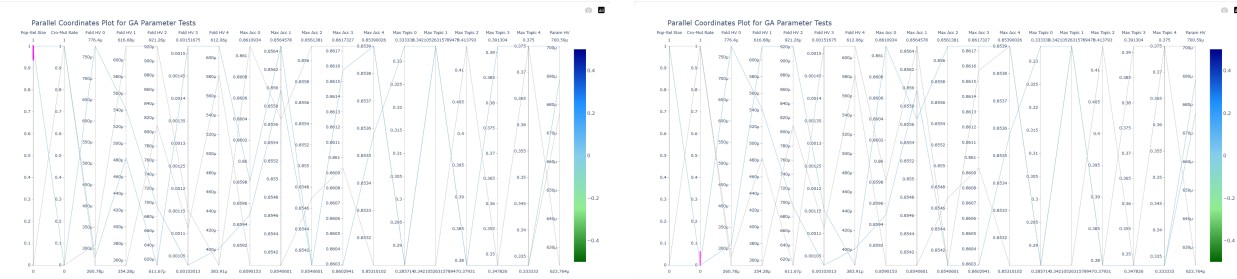

(a) Performances of Population and Selection Sizes of 20 and 20.

(b) Performances of Crossover and Initial Mutation rates of 0.7 and 0.3.

Figure 26: Genetic Algorithm Parameters Experiments.

- Population Size = 20

- Selection Size = 20

- Crossover Rate = 0.7

- Initial Mutation Rate = 0.3

