# OpenReview forum: "AutoGeTS: Knowledge-based Automated Generation of Text Synthetics for Improving Text Classification"
_TMLR — Rejected by TMLR_

### Review · Reviewer_2cCH · 2026-01-13

**Summary Of Contributions:**

The paper addresses an important challenge that arises when developing text classification models for real-world applications, namely the difficulty of collecting sufficient data for all text classes. To address this challenge the paper proposes a strategy of generating synthetic data to improve performance of a model (particularly for underrepresented classes). The method involves choosing a subset of the provided data via a search process to generate more effective synthetic data. The paper provides experimental results for three search algorithms, a single dataset, and a single base model (with additional analysis in the appendix). Based on these results, the procedure is extended to multi-stage and ensemble model improvement.

**Audience:**

No

**Audience Explanation:**

Although the paper addresses an interesting and important problem, the reported experimentation is not thorough and the analysis does not currently support the claims and conclusions.

In general, it is difficult to see how the proposed approach differs from a standard data augmentation approach. In general, one would propose an strategy for data augmentation and search over a hyperparameter space (using a selection of strategies) to determine the most effective mechanism for constructing the augmentations (synthetic data in this setting). The hyperparameters could be selected via a search procedure (e.g. Bayesian optimization, local search). The selection process would often involve splitting the data to evaluate a test metric on a validation dataset.

The approach in this paper seems like a special case of this process (synthetic data generation, search over the dataset to select examples - these can be viewed as the hyperparameters). There might be interesting aspects if there were a principled designed approach for the construction of the synthetic data and the search procedure for this specific application, but it appears to be heuristic and somewhat arbitrary.

To be of interest to the community, there would need to be (i) a principled design that clearly explains and motivates the choices for the data augmentation approach; and/or (ii) an extensive and thorough experimental study that includes a careful statistical analysis to verify hypotheses. Given that the paper contains relatively limited novel methodological content, one would expect extensive experimentation involving multiple base models and multiple datasets.

**Broader Impact Concerns:**

None.

**Claims And Evidence:**

No

**Claims Explanation:**

The paper claims to do the following:

C1) Improve the performance of a model by generating synthetic data without waiting for more real data to be collected and labelled

C2) formulate an automatic workflow for identifying input examples that result in more effective synthetic data

C3) specify three search strategies and conduct systematic machine learning experiments to construct an ensemble algorithm that chooses the search strategy based on the characteristics of the class

C4) demonstrate via experiment that the ensemble approach is more effective than each individual strategy.

The paper relies on another paper to motivate these search heuristics, but the other paper is cited as “Anonymous Authors” and “Anonymous Venue” in a strange attempt to preserve anonymity. Are reviewers not supposed to access the paper? A much better approach would just be to cite the paper properly and write this submitted paper so that there is no indication that the authors of that paper are the same as the authors of this paper.

There are several issues with the claims:

The search strategies are heuristic and there is no attempt to relate them to the many search procedures that have been proposed in the past (e.g., for a general overview of candidate methods, Kochenderfer and Wheeler, Algorithms for Optimization, 2025). Without this linkage, very little can be understood about the general performance of these search techniques or why they have been designed and selected. This leads to a very heavy reliance on the reported empirical results. There is then an important question as to why we should believe that the results and design decisions generalize beyond the few examined cases.

The main experiments are conducted for a single real-world dataset from a ticketing system with only one base model. These are the only results reported in the main body of the paper. Several design decisions are thus based on this one dataset and one model.

Concerning claim C1, there are no systematic studies to support the claim that the proposed approach consistently improves the performance of a model. This would require careful analysis of the variability of performance and testing a hypothesis that establishes a statistically significant difference. The analysis should also carefully account for multiple hypothesis testing, because there is a real danger that the design process has led to multiple models being evaluated and the best one selected for comparison to the original model. Beyond this, there is a concern about the generality of the claim if it is only established for three datasets and one base model.

The same problem exists for Claim C2. As it stands, the experimental analysis does not clearly establish that the proposed approach consistently identifies input examples that result in more effective data generation. This requires a careful pairwise comparison test to establish statistical significance.

Claim C3 is satisfied because the paper does propose search strategies and an ensembling approach. However, little motivation is provided for the selection or design of the search methods. These seem like arbitrary choices.

Claim C4 has the same issues as Claims C1 and C2: very few datasets, one base method, and no statistical testing of a hypothesis.

**Requested Changes:**

In my view, the paper requires major changes before it is ready for publication.

(1) The experiments need to be considerably more comprehensive with more datasets and more base models. Even if we consider three datasets to be borderline acceptable, there would need to be a careful and thorough statistical analysis of the results to support the claims that are made.

(2) The paper should make connections to existing data augmentation, validation, hyperparameter selection, and search approaches (clearly explaining what is different from a standard strategy). The identified search procedures under study should be carefully motivated and connected to existing search/optimization techniques in the literature.

---

> ### Author Response · Authors · 2026-01-30
> **Rebuttal by Author**
>
> **Q1 - “Citation problem on “Anonymous Authors” and “Anonymous Venue””**
>
> A1: We apologize for the confusion regarding the 'Anonymous Authors' citation. This references our prior work, 'iGAiVA' (Jin et al., arXiv:2409.15848), which introduced a visual analytics approach requiring human experts to manually select examples. AutoGeTS automates and extends that workflow by replacing human judgment with systematic search strategies. We will properly cite iGAiVA in the revision and clearly delineate the contributions of each paper.
>
> **Q2 - “search strategies are heuristic and there is no attempt to relate them to the many search procedures…”**
>
> A2: We thank the reviewer for this suggestion and the reference to Kochenderfer and Wheeler (2019). We agree that connecting our methods to established optimization frameworks strengthens the paper.
>
> Our strategies map to formal optimization concepts as follows:
>
> - Sliding Window (SW): Grid search with spatial locality constraints, exhaustively evaluating discrete regions in the projected feature space
> - Hierarchical Sliding Window (HSW): Coarse-to-fine search, which prunes unpromising regions at coarse resolution before refining
> - Genetic Algorithm (GA): Stochastic population-based optimization operating on binary selection vectors
>
> We chose these specific heuristics because text classification errors exhibit spatial clustering in embedding space. Misclassified examples from the same class tend to cluster together (Figure 3). Our search strategies exploit this structure, whereas generic black-box optimizers might ignore it. The knowledge map also mimics a meta-level learning component. It empirically captures which strategy-metric combinations work best for different class characteristics (e.g., GA outperforms SW for very small classes where spatial projections are too sparse for window-based search). This reduces computational cost in deployment by directing resources to effective configurations. We will add a 'Theoretical Connections' subsection linking our methods to this literature.
>
> **Q3 & Q4 - Statistical Significance (Stage 1 and Stage 2)**
>
> We thank the reviewer for emphasizing the need for statistical validation. We will address this concern for both stages.
>
> Clarification on model selection: Regarding the worry “real danger that the design process has led to multiple models being evaluated and the best one selected for comparison to the original model”, we thank the reviewer for pointing out that we did not explain clearly enough the systematic testing results, i.e., Tables 3, 15-25 for the Ticketing dataset; Tables 12(a), 13(a), 14(a) for TREC-6 and Amazon review datasets.
>
> We first clarify that reported improvements are not the result of selecting the best-performing model on the test set. Our workflow uses a strict split: Training (64%), Validation (16%), Test (20%). All model selection decisions, including which search strategy configuration to report, are made based on validation set performance only. The improvements in Tables 2-3 (Stage 1) and Figures 5-6 (Stage 2) are computed on the held-out test set, which never influences model selection. These improvements (e.g., around 30% of topic recall, 10% topic balanced accuracy) do not result from intentionally selecting the models with the best performance on the test set.
>
> We will add bootstrap resampling analysis (1000 iterations) to compute 95% confidence intervals for:
> - Stage 1: Per-class, per-metric improvements
> - Stage 2: Ensemble improvements in Figures 5-6
>
> We will also report results of paired hypothesis tests comparing AutoGeTS configurations against the baseline (no augmentation) and against uniform random selection (Li et al., 2023). These additions will appear in Sections 4 and 5, respectively.

---

> ### Author Response · Authors · 2026-01-30
> **Rebuttal by Author (continued)**
>
> **Q5 - “datasets”**
>
> A5: We acknowledge the reviewer's concern about generalization. We offer the following response:
>
> Current coverage: Beyond the main Ticketing dataset (15 imbalanced classes, industrial setting), we include experiments on TREC-6 and Amazon Reviews (Tables 12-14 in Appendix A). We also validate across three augmentation methods: GPT-3.5, EDA, and AEDA. This represents 3 datasets × 3 augmentation methods = 9 experimental configurations.
>
> Base model consideration: We used CatBoost because it is the production model in our industrial deployment. We agree that validating with other base models would strengthen generalization claims. In the revision, we will add experiments with an extra base model on the Ticketing dataset to verify that improvements are not specific to gradient-boosted trees.
>
> Scope clarification: Our primary contribution is the knowledge-based search framework, not a claim of universal superiority across all possible settings. The knowledge map approach is designed to be model-agnostic. It learns which search configurations work for which class characteristics, and this meta-knowledge should transfer across base models. The additional neural baseline experiments will test this hypothesis.
>
> **Q6 - “Differentiation from Standard Data Augmentation and Hyperparameter Search”**
>
> We appreciate this fundamental question, which we should have addressed more directly in the paper.
>
> Standard data augmentation treats all training examples equally or selects augmentation candidates uniformly at random (as in Li et al., 2023). Hyperparameter search in this context typically optimizes augmentation parameters (e.g., number of synthetic samples, generation temperature) but not which specific examples to augment.
>
> AutoGeTS differs in two ways:
> - Example-level selection: We optimize which training examples serve as seeds for synthetic generation. Our experiments show this matters substantially. Augmenting examples found with search strategies yields larger improvements than uniform selection (Table 28 in Appendix D1.3).
> - Class-conditional strategy selection: Different classes benefit from different search strategies (Table 2b). The knowledge map captures these patterns, enabling targeted resource allocation rather than applying a single strategy uniformly.
>
> We will add a subsection in Section 2 (Related Work) explicitly contrasting AutoGeTS with standard augmentation pipelines.

---

### Review · Reviewer_VWKC · 2026-01-14

**Summary Of Contributions:**

This work introduces a pipeline for enhancing classification performance on low-frequency classes using LLM-generated synthetic data. It targets a common setting of improving classifiers with scarce, imbalanced, and evolving data.

---
### Strengths
- The paper is well-motivated and easy to follow. The figures are also well-designed, but the main experimental figures are a bit dense.
- The selection of training examples method improves classification performance, which shows the motivation of the paper.

### Weaknesses
- The proposed strategies (SW, HSW, GA) are described as relying on the visualization of "negative testing results (red dots)" to guide the search for training examples. Is this not considered overfitting to a particular test set, thereby inflating performance? Moreover, why do the authors continue to use the 2D constrained space?
- Can the authors elaborate on how the synthetic examples are being generated? A few more details being added about the data generation itself would be appreciated. I do however appreciate the detail and rigor shown in Appendix C.
- How computationally expensive are the search algorithms (especially GA)?

**Audience:**

Yes

**Audience Explanation:**

Yes, this paper would be interesting to the TMLR's general audience, since this problem is practical and common - assisting with several tasks seen daily.

**Claims And Evidence:**

No

**Claims Explanation:**

The experiments do support the overall research questions defined in Section 3 and 4. However, as per my weaknesses defined, the major concern is that the selection of training examples seems to be overfitting to the test distribution, which could be inflating the performances in the experiments.

Regardless, there are extensive experiments in the Appendix, which I highly appreciate.

**Requested Changes:**

Several requested changes have been listed in the Weaknesses. I would support this paper if the above concerns are addressed sufficiently by the authors.

---

> ### Author Response · Authors · 2026-01-30
> **Rebuttal by Author**
>
> **Q1.1 - “overfitting”**
>
> A1.1: We thank the reviewer for identifying this ambiguity, which we recognize was unclear in our submission.
>
> We confirm there is no data leakage. The AutoGeTS workflow uses a strict three-way split: Training (64%), Validation (16%), and Test (20%). The "red dots" (misclassified examples) shown in Figure 3 come exclusively from the Validation set, not the Test set. The search strategies (SW, HSW, GA) use only validation performance to guide example selection. All final performance metrics reported in Tables 2-3 and Figures 5-6 are computed on the held-out Test set, which never influences the search process.
>
> We will add a clear data flow diagram and explicit description of this split in Section 4 to remove any ambiguity.
>
> **Q1.2 - “2D constrained space”**
>
> A1.2: We thank the reviewer for suggesting that we search in high-dimensional space. Indeed, the sliding window and hierarchical sliding window methods can operate in the high-dimensional space. We will conduct experiments on this in the following research. We employ the 2D projection (for SW and HSW methods) as a necessary approximation regarding the Curse of Dimensionality. Searching for dense clusters of errors in a high-dimensional space (e.g., a 768-dimensional embedding space) is much more computationally costly. The volume of the space grows exponentially, making "sliding windows" too sparse (a window would contain mostly empty space, with very few data dots).
>
> **Q2 - “How synthetic samples are generated”**
>
> A2: We thank the reviewer for pointing out that we did not explain the synthetic samples generation process clearly enough in the main paper. We presented this process in Appendix D.1, but we agree that the main paper should include a summary.
>
> In short, for each selected example, we call GPT-3.5 with a zero-shot prompt requesting similar text samples. Generated outputs pass through an automated quality check (filtering empty responses, malformed formatting). Failed samples trigger regeneration. We will add a concise description of this process to Section 3 in the revision.
>
> **Q3 - “Computational cost, especially GA”**
>
> A3: All results we report in the paper were obtained from 1 GPU hour fixed-time experiments. This makes sure the number of model retrainings is almost the same across search strategies. We have measured the training time and performance gains of our three selection strategies under a 1-GPU-hour limit. Our findings show that:
>
> 1. HSW and GA often achieve their best performance within the first third of the total 1GPU hour retraining time
> 2. SW sometimes requires more time in classes where it performs best
>
> Detailed plots of improvement trajectories are presented in Appendix D1.3 (Figure 15).
>
> Theoretically, the sliding window method (SW), which is an exhaustive grid search, has time complexity $O(n^d)$, where $d$ is the number of dimensions and $n$ is the number of discrete positions per dimension (assuming constant window size). For heterogeneous grids, this generalizes to $O\!\left(\prod_{i=1}^{d} n_i\right)$.
>
> HSW, which is a coarse-to-fine search, reduces exhaustive enumeration of $O(n^d)$ windows to $O(K\,r^d\,\log n)$ window evaluations under the assumption that only $K$ regions survive pruning per level and each refinement subdivides a region by a factor $r$ along each of the $d$ dimensions (so the per-level branching factor is $r^d$). In the worst case (no effective pruning), HSW can revert to $O(n^d)$ evaluations.
>
> The GA performs $P$ fitness evaluations per generation; excluding model retraining, the remaining search overhead (selection, crossover, mutation, and survivor selection) is at most $O(P\,L + P\log P)$ per generation, where $P$ is the population size and $L$ is the chromosome length. Therefore, over $G$ generations the total search cost is $O\!\left(G(P\,L + P\log P) + G\,P\,C_{\text{eval}}\right)$, where $C_{\text{eval}}$ denotes the per-individual non-retraining fitness computation.

---

### Review · Reviewer_cgs4 · 2026-01-19

**Summary Of Contributions:**

The paper studies the use of LLM-generated synthetic text to improve class-imbalanced text classification systems, particularly in industrial ticketing settings. It frames synthetic data generation as an optimization problem over example-selection strategies and evaluation metrics, and conducts systematic experiments comparing multiple selection methods and objectives across several datasets and classes. The results are aggregated into empirical “knowledge maps” that summarize which strategy–metric combinations tend to work best for different classes, with the main finding that no single strategy or metric consistently dominates across settings.

**Additional Comments:**

The paper presents a large-scale empirical study of LLM-based synthetic data generation for text classification under class imbalance. While the experimental coverage is broad and the results are reported transparently, the contributions are largely methodological and incremental. The work does not introduce new modeling ideas, theoretical insights, or principled mechanisms beyond systematic enumeration of existing strategies and metrics. As a result, I am not fully convinced that the paper meets the bar for acceptance.

**Audience:**

Yes

**Audience Explanation:**

Yes, but only a limited subset. Some TMLR readers interested in data-centric ML, applied NLP, and LLM-assisted augmentation may find the findings useful, but the paper is unlikely to attract broad interest across the general TMLR audience.

**Broader Impact Concerns:**

The paper may help practitioners experiment with synthetic data for existing datasets, but it does not meaningfully help people build datasets in the usual sense (collection, annotation, curation, schema design, quality control). Its focus is on model improvement via LLM-generated augmentation, not on dataset construction or release practices.

**Claims And Evidence:**

No

**Claims Explanation:**

The submission provides extensive and carefully reported empirical results showing that LLM-generated synthetic data can improve minority-class performance and that outcomes depend on the choice of example-selection strategy and optimization metric. However, broader claims about AutoGeTS as a general or knowledge-based optimization framework are not convincingly supported, as the evidence does not clearly isolate its contribution beyond systematic experimentation and standard synthetic data augmentation.

**Requested Changes:**

I do not have specific requested changes. While the paper is clearly written and the experiments are extensive, I am not convinced that the overall idea and contribution meet the bar for acceptance, and my concerns are primarily about the novelty and significance of the approach rather than correctable issues in the current submission.

---

> ### Author Response · Authors · 2026-01-30
> **Rebuttal by Author**
>
> Q - “Novelty”
>
> A: We thank the reviewer for the thoughtful assessment and for recognizing that our experiments are large-scale and carefully reported.
>
> We respectfully clarify that our novelty claim does not rest on the individual components (LLMs, GA, PCA), which are indeed well-established. Rather, our contribution lies in how these components are integrated into a principled, knowledge-based framework that solves a concrete industrial problem. Specifically:
>
> 1. Knowledge Map Construction: We systematically explore 180 strategy-metric-class combinations (Table 2) to build empirical "knowledge maps" that capture which search strategies work best under which conditions. This meta-level learning distinguishes our approach from ad-hoc augmentation.
> 2. Knowledge-Based Resource Allocation: The ensemble algorithm (Figure 4) uses these maps to distribute computational resources proportionally to expected performance gains. Such a mechanism is absent from prior work.
> 3. Generalization Beyond LLMs: We validate that the framework improves performance with EDA and AEDA (Tables 5-7, 12-14), demonstrating that our contribution is the search and selection methodology, not LLM-specific engineering.
>
> Differentiation from Prior Work: Unlike Jin et al. (2024, arXiv:2409.15848), which requires human experts to visually select examples, AutoGeTS fully automates this process. Unlike Li et al. (2023), which uses uniform random selection, our knowledge-based approach achieves substantially higher improvements (e.g., 32% vs. baseline recall for T12 in Table 2a).
>
> Practical Significance: We believe the practical impact merits consideration under TMLR's evaluation criteria. Our framework delivers:
> 1. Minority-class recall improvements of 20-33% for severely underrepresented classes (T12-T15, Table 2a)
> 2. 10%+ balanced accuracy gains for the smallest classes (Tables 18, 19)
> 3. Multi-objective improvements: +3% recall on medium classes while simultaneously achieving +30% on minority classes and +1% overall F1 (Figure 6)
>
> For industrial ticketing systems processing thousands of messages daily, a 30% reduction in misclassified minority-class tickets translates directly to reduced manual intervention costs and faster service delivery. We will add this practical interpretation to the revision.

---

### Decision · Action_Editor_piX9 · 2026-03-07

**Recommendation:** Reject

**Audience:**

Yes

**Audience Explanation:**

The paper studies a topic that is likely of interest to at least part of the TMLR audience. However, this potential interest is reduced by the current paper’s limited evidentiary support and unclear methodological contribution.

**Claims And Evidence:**

No

**Claims Explanation:**

The paper addresses a relevant problem, but the current version does not provide sufficiently strong evidence for several of its main claims. In particular, the empirical support is limited in scope, which makes it difficult to assess the generality of the conclusions, and the statistical analysis is not sufficient to establish the robustness of the reported improvements. In addition, the methodological contribution remains somewhat unclear in terms of its broader generality and principled justification